# Pre-trained Large Language Models Learn to Predict *Hidden* Markov Models In-context

**Yijia Dai**[*]  **Zhaolin Gao**  **Yahya Sattar**  **Sarah Dean**  **Jennifer J. Sun**

Cornell University

## Abstract

Hidden Markov Models (HMMs) are foundational tools for modeling sequential data with latent Markovian structure, yet fitting them to real-world data remains computationally challenging. In this work, we show that pre-trained large language models (LLMs) can effectively model data generated by HMMs via in-context learning (ICL)—their ability to infer patterns from examples within a prompt. On a diverse set of synthetic HMMs, LLMs achieve predictive accuracy approaching the theoretical optimum. We uncover novel scaling trends influenced by HMM properties, and offer theoretical conjectures for these empirical observations. We also provide practical guidelines for scientists on using ICL as a diagnostic tool for complex data. On real-world animal decision-making tasks, ICL achieves competitive performance with models designed by human experts. To our knowledge, this is the first demonstration that ICL can learn to predict HMM-generated sequences—an advance that deepens our understanding of in-context learning in LLMs and establishes its potential as a powerful tool for uncovering hidden structure in complex scientific data. Our code is available at https://github.com/DaiYijia02/icl-hmm.

## 1 Introduction

Many natural and artificial systems, from animal decision-making to ecological processes to climate patterns, generate observations governed by underlying, unobservable states that follow Markovian dynamics [3, 17, 34, 48, 57]. Hidden Markov Models (HMMs) [39] provide a powerful framework for studying such phenomena. However, accurately modeling these systems presents significant challenges. Parameter estimation and model fitting require complex algorithms like Baum-Welch [5], and Gibbs Sampling [16]. These methods are often computationally intensive and can be algorithmically unstable, demanding extensive domain expertise [8, 9]. For scientists across disciplines, these accessibility and computational bottlenecks limit the practical applications of HMM modeling tools.

Recently, large language models (LLMs) [1, 18] have reshaped the landscape of AI. Trained on vast amounts of sequential text data, they have achieved unprecedented performance across natural language processing tasks and exhibit remarkable in-context learning (ICL) capabilities—the ability to learn patterns and perform new tasks directly from examples provided in the input context, without explicit parameter updates [7]. While prior theoretical and empirical works [13, 33, 40] have demonstrated LLMs' capabilities as Bayesian learners and their ability to model fully observed Markov processes, their capacity to implicitly learn Hidden Markov Models—with latent states, complex transition dependencies, and observation emissions—remains largely unexplored. Understanding this capacity could illuminate the mechanisms underlying in-context learning HMMs, and reveal new ways to leverage LLMs for analyzing complex sequential phenomena in scientific contexts.

---

[*]Correspondence to yd73@cornell.edu.

39th Conference on Neural Information Processing Systems (NeurIPS 2025).

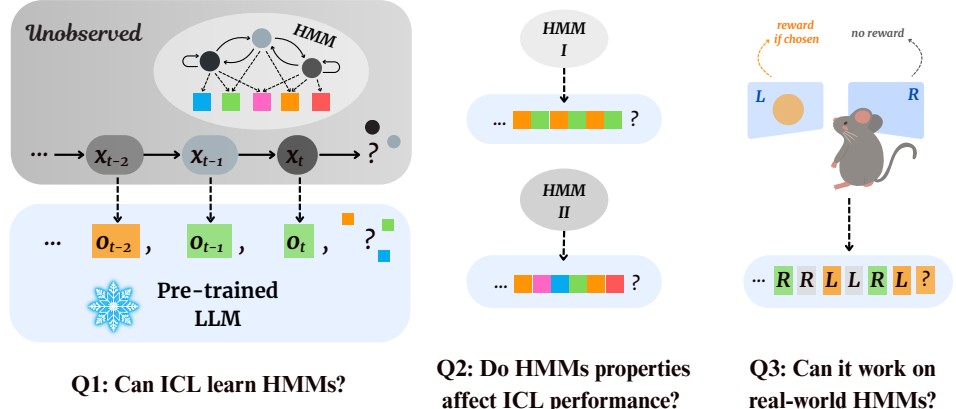

**Q1: Can ICL learn HMMs?**     **Q2: Do HMMs properties affect ICL performance?**     **Q3: Can it work on real-world HMMs?**

Figure 1: **Overview of our study.** We start by studying whether ICL using pre-trained LLMs can converge to theoretical optimum on HMM sequences (Q1, Section 2), then study how HMMs properties affect the convergence rate/gap with theoretical conjectures (Q2, Section 3), and finally we demonstrate how these findings translate to insights on real-world datasets for studying behaviors in science (Q3, Section 4).

In this paper, we present a comprehensive study on the ability of pre-trained LLMs to learn HMMs through in-context learning (Figure 1), revealing their surprisingly strong performance and offering actionable insights for real-world scientific experiments. A key finding is that pre-trained LLMs demonstrate a remarkable capacity to learn HMMs nearly optimally, achieving performance that approaches optimal Bayesian inference and often surpasses traditional statistical methods. These results not only advance our understanding of the emergent capabilities of in-context learning, but also introduce a novel and practical framework for using LLMs as powerful, efficient statistical tools in complex scientific data analysis. Our study makes three key contributions:

1. We conduct systematic, controlled experiments on synthetic HMMs and empirically show that pre-trained LLMs **outperform** traditional statistical methods such as Baum–Welch. Moreover, their prediction accuracy consistently **converges to the theoretical optimum**—as given by the Viterbi algorithm with ground-truth model parameters—across a wide range of HMM configurations (Section 2).

2. We identify and characterize empirical **scaling trends** showing that LLM performance improves with longer context windows, and that these trends are shaped by fundamental HMM properties such as mixing rate and entropy. We further provide **theoretical conjectures** to explain these phenomena, drawing connections to—and highlighting distinctions from—classical HMM learning paradigms, including spectral methods. These findings offer important insights into the learnability of stochastic systems through in-context learning (Section 3).

3. We translate our findings into **practical guidelines for scientists**, demonstrating how LLM in-context learning can serve as a diagnostic tool for assessing data complexity and uncovering underlying structure. When applied to real-world animal decision-making tasks, LLM ICL **performs competitively with domain-specific models** developed by human experts (Section 4).

## 2 Synthetic Experiments and ICL Convergence

We investigate the in-context learning capabilities of pre-trained LLMs on sequences generated by synthetic HMMs. We first review key HMM properties (Section 2.1) and outline our experimental setup (Section 2.2). We then empirically demonstrate that the prediction accuracy of pre-trained LLMs consistently converges to the theoretical optimum (Section 2.3).

### 2.1 HMM Background

**Hidden Markov model:** HMMs impose a set of probabilistic assumptions on how sequences of data are generated. The elements of the sequence are called observations, denoted at each step $t$ by $O_t$. The observations depend on a hidden state denoted by $X_t$, which evolves according to a

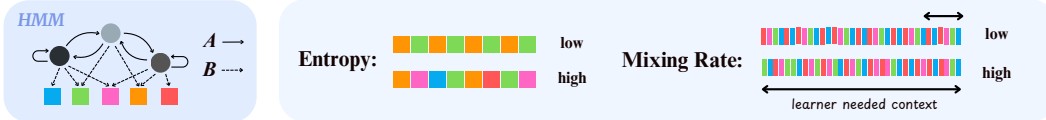

Figure 2: Properties of HMMs that impact pre-trained LLMs in-context learning performance.

**Markov chain.** A HMM is characterized by the Markov chain's *initial state distribution* and its *state transitions*, along with the *emission probabilities* of an observation given the hidden state. The key assumptions are that the state transition depends only on the previous state (Markov property), the observation depends only on the current hidden state (output independence), and both transition and emission probabilities are time-invariant (stationarity).

We focus on the setting with finitely many states and observations. Without loss of generality, states take values in $\mathcal{X} = \{1, 2, \ldots, M\}$ while observations take values in $\mathcal{O} = \{1, 2, \ldots, L\}$. The initial state distribution is denoted as $\boldsymbol{\pi} \in \mathbb{R}^M$ with $\pi_j$ the probability of starting in state $j$, the state transitions are describe by the matrix $\mathbf{A} \in \mathbb{R}^{M \times M}$ with elements $a_{ij}$ the probability of transitioning to state $j$ from state $i$, and the emission matrix $\mathbf{B} \in \mathbb{R}^{M \times L}$ contains $b_{jl}$ the probability of observing $l$ when in hidden state $j$. The triple $\lambda = (\boldsymbol{\pi}, \mathbf{A}, \mathbf{B})$ completely parameterizes a finite-alphabet HMM.

**Stationary distributions:** Under certain conditions (see Appendix A), Markov chains are guaranteed to converge to unique stationary distributions, which are given by the $\boldsymbol{\mu} \in \mathbb{R}^M$ satisfying $\boldsymbol{\mu} = \boldsymbol{\mu}\mathbf{A}$ [14]. The stationary distribution of the hidden state characterizes the long term behavior of the HMM, and therefore plays an important role in both predicting future observations and learning HMM parameters. The rate of convergence is characterized by the *mixing rate* which for finite-alphabet HMMs is equal to $\lambda_2$, the second-largest eigenvalue of $\mathbf{A}$. From any initial distribution, the hidden state distribution approaches the stationary distribution geometrically with multiplier $\lambda_2$. A smaller mixing rate indicates faster convergence to the stationary distribution.

**Entropy:** HMMs can describe processes which vary from deterministic to purely random, depending on how transition and emission probabilities are defined. Entropy is a measure of the randomness or unpredictability of a random variable. By considering the average entropy over the stochastic processes of hidden state and observation, we can quantify the entropy of a particular HMM by $H(\mathbf{A}) = -\sum_{i,j} \mu_i a_{ij} \log a_{ij}$ and $H(\mathbf{B}, \boldsymbol{\mu}) = -\sum_{j,l} \mu_j b_{jl} \log b_{jl}$. We additionally define normalized entropies $\tilde{H}(\mathbf{A}) = H(\mathbf{A})/\log M$ and $\tilde{H}(\mathbf{B}, \boldsymbol{\mu}) = H(\mathbf{B}, \boldsymbol{\mu})/\log L$ as metrics for visualization. A smaller entropy indicates a more predictable process. See Appendix A for further explanation.

## 2.2 Experimental Setup

**Experiment setting:** Our experiment follows a three-step protocol: First, we specify the HMM parameters $\lambda = (\boldsymbol{\pi}, \mathbf{A}, \mathbf{B})$ according to our control variables (described below). Second, we generate observation sequences $\{o_1, o_2, \ldots\}$ from this parameterized model. Third, we evaluate the ability of candidate models to predict the next observation $o_{t+1}$ given preceding observations $o_{1:t}$.

We systematically vary five control parameters and consider 234 total HMM settings: (1) state and observation space dimensions, with $M, L \in \{2, 4, 8, 16, 32, 64\}$; (2) mixing rate of the hidden Markov chain, with $\lambda_2 \in \{0.5, 0.75, 0.95, 0.99\}$, where $\lambda_2$ is the second-largest eigenvalue of $\mathbf{A}$; (3) skewness of the stationary distribution $\boldsymbol{\mu}$ (uniform or non-uniform); (4) entropy of the transition and emission matrices $\mathbf{A}$ and $\mathbf{B}$, ranging from deterministic (zero entropy) to maximum entropy (random); and (5) initial state distribution $\boldsymbol{\pi}$ (uniform or deterministic). While generating $\boldsymbol{\pi}$ and $\mathbf{B}$ is straight-forward, for matrix $\mathbf{A}$, we define a constrained optimization problem and solve using first order optimization. See Appendix B for additional details. For each parameter configuration, we sample 4,096 state-observation sequence pairs, each of length 2,048. We assess model performance across context lengths ranging from 4 to 2,048 observations, specifically $\{4, 8, 16, 32, 64, 128, 256, 512, 1024, 2048\}$. For each HMM setting, we report performance metrics averaged over the 4,096 samples. Our candidate models are open-source pre-trained LLMs (Qwen and Llama family). Note that we are not training the LLMs, only evaluating their capability for in-context learning. In the following sections, we evaluate LLMs' performance by comparison to several other approaches (Table 1).

| Method | Input | Variable | Description |
|---|---|---|---|
| Viterbi [51] | $O_{1:t}, \lambda$ | $\emptyset$ | Finds most likely hidden state sequence |
| $P(O_{t+1}\|O_{t-k:t})$ | $O_{t-k:t}, \lambda$ | $k$ | Direct conditional probability |
| Baum-Welch [5] | $O_{1:t}, M$ | $\emptyset$ | EM algorithm for HMM parameter estimation |
| Spectral [21] | $O_{1:t}, M$ | $\emptyset$ | Improper learning for spectral parameters (Section 3.3) |
| $n$-gram | $O_{1:t}$ | $n$ | Optimal $(n-1)$-th order Markov predictor |
| LSTM (RNN) | $O_{1:t}$ | $\emptyset$ | Neural network with memory cells |
| Transformer | $O_{1:t}$ | $\emptyset$ | Neural network with multi-head attention |

Table 1: Methods for HMM prediction task.

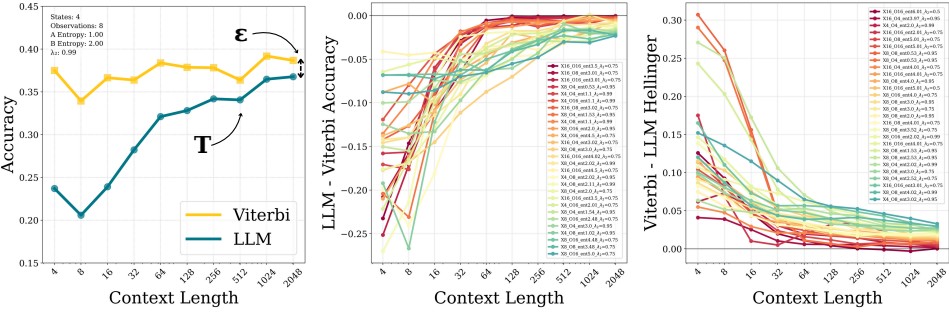

Figure 3: *(Left)* We define **T** as *when* LLM converges (see Appendix B for computation metric), and $\varepsilon$ as the final accuracy *gap* at sequence length 2048. *(Middle)* Examples when LLM accuracy converges to Viterbi. Each curve represents a different HMM parameter setting. LLM ICL shows consistent convergence behavior. *(Right)* Examples of convergence in Hellinger distance (distance between two probability distributions). LLM ICL is not just "guessing" the most probable output, but converging distributionally.

## 2.3 ICL Converges to Theoretical Optimum

We define *convergence* as achieving prediction accuracy comparable to the Viterbi algorithm. The Viterbi algorithm, given ground-truth HMM parameters $\lambda$, computes the most likely hidden state sequence $x_{1:t}$ from observations $o_{1:t}$ (see Appendix C for details). Since Viterbi has access to the true model parameters, its performance represents the theoretical optimum. Remarkably, ICL with pre-trained LLM achieves this near-optimal prediction accuracy across diverse HMM parameter configurations in our experiments. Figure 3 illustrates examples and conditions under which this convergence occurs. Convergence occurs reliably when HMM entropy is low and mixing is fast.

For challenging conditions where LLM convergence fails or proceeds exceptionally slowly—like the red areas shown in the left-hand side of Figure 4—Viterbi algorithm also exhibits diminished prediction accuracy and requires substantially longer context windows to achieve reliable performance. This degraded performance reflects the fundamental limits of stochastic system learnability due to random dynamics and long-range dependencies, affecting even the optimal inference methods. See Appendix D for detailed examples.

## 3 Impact of HMM Properties on Convergence

Having established that ICL with pre-trained LLM converges to the theoretically optimal predictions, we now provide an in depth characterization of their performance across variable settings. We summarize the scaling trends in terms of key HMM properties, compare these patterns against other popular methods for HMM prediction, and conclude by providing theoretical conjectures.

### 3.1 In-context Scaling Trends

Neural scaling laws describe empirical power-law relationships that characterize how neural network performance improves with increases in key resources such as model size, dataset size, and compute [24]. For in-context learning [29], it describes trends between prediction accuracy and context

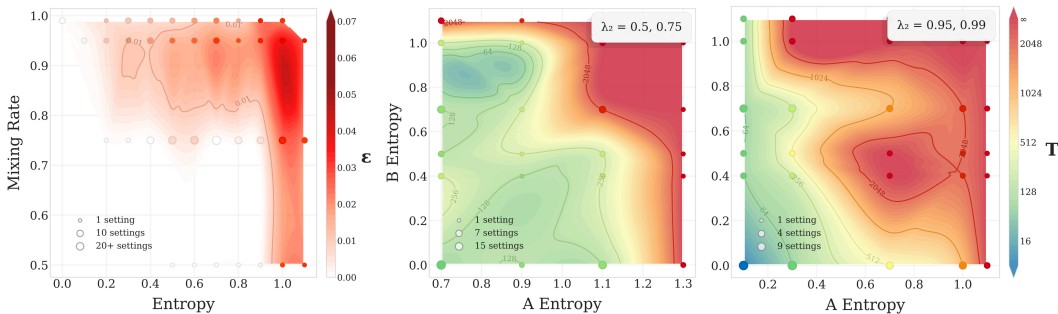

Figure 4: *(Left)* Convergence gap $\varepsilon$ increases with higher mixing rate (slower mixing) and higher entropy. This plot is showing results averaged across all HMM configurations we tested. *(Right)* Slower mixing ($\lambda_2 = 0.95, 0.99$) shows delayed convergence compared to *(Middle)* fast mixing ($\lambda_2 = 0.5, 0.75$) at similar entropy levels.

window length. We further show how these scaling trends depend on the underlying stochastic process characteristics: entropy, mixing rate, and state-space dimensionality.

**Context window length:** Given an observation sequence sampled from an HMM, LLM performance generally improves monotonically with increasing sequence length before plateauing. Representative examples are shown in Figure 3. Fluctuations may occur when the entropy of the sequence is high; additional examples are provided in Appendix D.

**Entropy of transitions and emissions:** Entropy determines the predictability of the observation sequence. The entropies of both the transition matrix $\mathbf{A}$ and the emission matrix $\mathbf{B}$ are positively correlated with the number of steps required for LLM convergence, as shown in the middle and right plots of Figure 4. However, this relationship is not strictly monotonic in practice.

**Mixing rate:** We control the mixing rate of synthetic HMMs using $\lambda_2$, the second-largest eigenvalue of $\mathbf{A}$. Lower values of $\lambda_2$ indicate faster mixing. As illustrated in Figure 4 (middle vs. right), for the same entropy level, convergence occurs significantly later when mixing is slow—indicating that slower mixing delays LLM learning.

**Number of hidden states and observations:** The dimensionality of the state and observation spaces affects the maximum possible entropy. While larger state spaces intuitively allow for higher entropy, our experiments show that when entropy is held constant, varying the number of states does not impact LLM convergence rates. Detailed examples and discussion are provided in Appendix D.

## 3.2 Comparison to Baselines

We compare the in-context learning performance of pre-trained LLMs against several established methods commonly used for HMM prediction tasks, as summarized in Table 1. An oracle conditional predictor, $P(O_{t+1}|O_{t-k:t})$, uses the true HMM parameters but truncates the history to model limited memory (cf. Viterbi-style oracle inference). For learning-based baselines, we include the classical Baum–Welch (BW) expectation-maximization algorithm, which remains the statistical state of the art for HMM parameter estimation [56]. Spectral methods have empirical evidence of converging to the theoretical optimum on a range of HMM prediction tasks [31, 42]. We also train two neural models: an LSTM (reflecting common RNN practice in neuroscience [54]) and a 2-layer, 4-head Transformer with self-attention architecture akin to pretrained LLMs; due to training cost, these results are averaged over 16 samples (vs. 4,096 elsewhere). Finally, $n$-gram baselines connect our HMM results to recent ICL findings in Markovian settings [40]. Full algorithm descriptions and implementation details are provided in Appendix C.

Across a range of HMM configurations, we find that LLM consistently outperforms empirical learning baselines. A representative example is shown in Figure 5 left two panels. Among the $n$-gram models with $n \in \{1, \ldots, 4\}$, the bigram model performs best, aligning with its role as the maximum likelihood estimator for first-order Markov chains. However, because HMM observations are not Markovian when $H(\mathbf{B}) > 0$, the bigram model is inherently suboptimal. Trigram models converge more slowly than bigram due to increased data sparsity and resulting estimation bias.

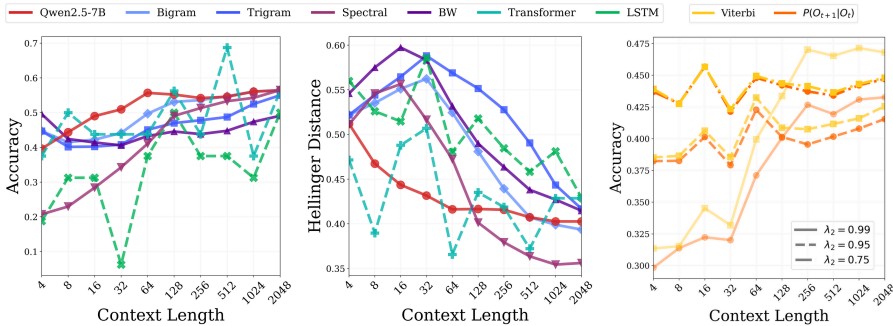

Figure 5: HMM parameters $M = 8$, $L = 8$, $H(\mathbf{A}) = 1.5$, $H(\mathbf{B}) = 1$. *(Left)* Accuracy comparison between LLM ICL (Qwen2.5-7B) and baselines; Hellinger distance is between model and ground-truth predictive distributions. The dash line means the result is averaged over 16 samples (vs. 4,096 elsewhere). *(Right)* The gap between $P(O_{t+1}|O_t)$ and Viterbi is small when mixing is fast. The line styles differentiate mixing rates.

Spectral learning algorithm converges to optimal prediction with longer context. Note that at early context its hidden-state belief update becomes numerically unstable because the sample complexity of spectral learning scales as $\mathcal{O}(M^2 L)$, which is 512 context length in the Figure 5 setting. The Baum–Welch (BW) algorithm, while given the correct HMM structure and leveraging expectation-maximization, suffers from nonconvex optimization. Its global convergence is not guaranteed. Even when averaging across multiple random seeds and 4,096 samples per setting, BW converges slowly and often unreliably. LSTM and Transformer baselines, despite their flexibility, require significant computational resources and exhibit unstable accuracy across varying context lengths. Transformer performs better than LSTM, which suggests attention mechanism plays important role for learning to predict HMMs. Notably, LLMs via ICL demonstrate clearly superior behavior across all baselines—achieving faster, more stable convergence to the ground-truth distribution and highlighting their surprising efficiency in modeling HMMs.

On the other hand, in Figure 5 right, the conditional predictor $P(O_{t+1} \mid O_{t-k:t})$ can approach Viterbi-level performance, particularly when the mixing rate is low. This observation suggests that approximate prediction using a truncated observation history can be nearly as effective as statistically optimal inference in fast-mixing regimes, motivating the conjecture discussed in Section 3.3.

### 3.3 A Possible Theoretical Explanation

In this section, we give a potential explanation for the ICL behavior of pre-trained LLMs for HMM sequence prediction by comparing it with the spectral learning algorithm [21]. This is motivated by empirical evidences [31, 42] showing the convergence of spectral learning prediction to theoretical optimum (in limited settings). Furthermore, for partially observed linear dynamical systems, Li et al. [27] observes that transformers can learn statistically optimal predictions in-context when trained on many similar tasks in the meta-learning setting. These findings suggest that ICL by LLMs may exhibit similar performance characteristics to spectral learning algorithms.

A key idea in spectral learning literature is to compute the probability of observation sequences in terms of *observation operators* [21]: For any $o \in \mathcal{O}$, define $\mathbf{A}_o := \mathbf{A}^\top \mathbf{diag}([\mathbf{B}]_{1,o}, \ldots, [\mathbf{B}]_{M,o})$, where $[\mathbf{B}]_{i,j}$ denotes the $ij$-th element of $\mathbf{B}$. Then for any $t > 0$,

$$\mathbb{P}(O_{t+1} = o_{t+1} \mid O_{1:t} = o_1, \ldots, o_t) = \frac{\mathbf{1}^\top \mathbf{A}_{o_{t+1}}^\top \mathbf{A}_{o_t}^\top \cdots \mathbf{A}_{o_1}^\top \boldsymbol{\pi}}{\mathbf{1}^\top \mathbf{A}_{o_t}^\top \cdots \mathbf{A}_{o_1}^\top \boldsymbol{\pi}}, \tag{1}$$

where $\mathbf{1}$ is all one vector of appropriate dimension. In this formulation, the conditional probability is estimated by first learning the spectral parameters (Appendix F) using training samples (in-context observations). Then, one can predict the next observation directly using these parameters along-with hidden state belief updates, without explicitly learning the matrices $\mathbf{A}$, and $\mathbf{B}$. This is therefore an example of *improper learning*, which has been extensively studied in related areas like linear dynamical systems [45]. The spectral learning algorithm is theoretically well understood. The

following theorem is obtained by extending the results by Hsu et al. [21] to single trajectory spectral learning.

**Theorem 1** *(Informal) Fix $\epsilon, \delta > 0$. Let $\mathbf{\Sigma}_2$ denote the pairwise probability matrix of observations such that $[\mathbf{\Sigma}_2]_{ij} = \mathbb{P}(O_{t+1} = i, O_t = j)$. Suppose $\boldsymbol{\pi} > 0$ element-wise, and $\mathbf{A}$, $\mathbf{B}$ are rank $M$. Suppose $L \geq M$, and let $\sigma_M(\cdot)$ denote the $M$-th largest singular value. Suppose the observation operator $\mathbf{A}_o > 0$ element-wise for all $o \in \mathcal{O}$, and*

$$t \gtrsim \frac{1}{1 - \lambda_2(\mathbf{A})} \left( \frac{M^2 L}{\epsilon^4 \sigma_M(\mathbf{B})^2 \sigma_M(\mathbf{\Sigma}_2)^4} + \frac{ML}{\epsilon^2 \sigma_M(\mathbf{B})^2 \sigma_M(\mathbf{\Sigma}_2)^2} \right) \log\left( \frac{1}{\delta} \right), \qquad (2)$$

*Then, with probability at least $1 - \delta$, the next observation prediction $\hat{\mathbb{P}}(\cdot \mid O_{1:t})$ using spectral learning algorithm (detailed in Appendix F) satisfies the following upper bound in Hellinger distance,*

$$H^2 \left( \mathbb{P}(O_{t+1} \mid O_{1:t} = o_1, \ldots, o_t), \hat{\mathbb{P}}(O_{t+1} \mid O_{1:t} = o_1, \ldots, o_t) \right) \leq \epsilon \qquad (3)$$

Theorem 1 indicates that the scaling trends we observed in Section 3.1 are similar to those of spectral learning based predictions. Specifically, like our observations in Section 3.1, the prediction accuracy improves with more samples (i.e., larger $t$). The mixing rate, captured by $\frac{1}{1-\lambda_2(\mathbf{A})}$, affects ICL and spectral learning similarly. This occurs because spectral parameter estimation from a single trajectory degrades with $\frac{1}{1-\lambda_2(\mathbf{A})}$—the faster the HMM mixes, the smaller the estimation error. Finally, the effect of entropy is captured by the observability conditions in Theorem 1. Estimation error is maximized when the HMM is unobservable, which corresponds to maximum entropy rate. The relationship between entropy and HMM observability has been well studied in literature [28, 32].

One of the practical limitations of spectral learning algorithm is the requirement of rank conditions of $\mathbf{A}$ and $\mathbf{B}$. Furthermore, the spectral learning algorithm is sensitive to the conditioning[2] of the observed sequence, making its numerical performance robust only in limited settings. ICL by pre-trained LLMs seems to handle such issues more gracefully, pointing to an intriguing gap in our statistical understanding for learning HMMs.

## 4 Guidelines for Practitioners: How to (*creatively*) use LLMs for your data?

LLMs' capacity in deciphering complex sequential patterns in language can be repurposed: as demonstrated in our synthetic experiments (Sections 2 and 3), pre-trained LLMs can effectively model HMM-generated sequences through ICL, achieving theoretically optimal prediction accuracy under favorable conditions. This section first translates these findings into practical guidelines for scientists, then demonstrates our observations on real-world data in animal behavior (Section 4.1).

**Guideline 1:** *LLM in-context learning as a diagnostic tool for data structure and learnability.* Our synthetic experiments (Sections 2.3 and 3.1) reveal that key HMM properties—notably entropy and mixing rate—strongly influence LLM ICL convergence behavior. Practitioners can leverage this relationship as a diagnostic tool for their own sequential data. If you observe that an LLM's ICL prediction accuracy on your data sequence steadily improves and saturates with increased context length (like Figure 3), this strongly indicates a learnable, non-random underlying structure. Our findings show that LLMs achieve near-optimal prediction accuracy on HMMs with clear, learnable patterns (Section 2.3).

The characteristics of this convergence provide further insight: faster convergence and higher final accuracy in LLM ICL experiments are consistently associated with HMMs having lower entropy (less randomness) and faster mixing rates, as shown in Section 3.1 and Figure 4. Conversely, if LLM ICL on your data converges slowly, requires exceptionally long contexts, or plateaus at low accuracy, this suggests the underlying process has high entropy or slow mixing dynamics—characteristics that inherently limit predictability and affect even optimal methods like Viterbi (Section 2.3). While calculating intrinsic HMM parameters from real-world data is challenging, you can qualitatively assess your data's learnability by comparing its ICL convergence profile to our synthetic HMM experiments (Figures 3 and 4).

---

[2]This issue should not be insurmountable, similar to how (appropriately tuned) regularization can overcome poor conditioning in ridge regression [37]. However, we are unaware of prior work which provides a solution.

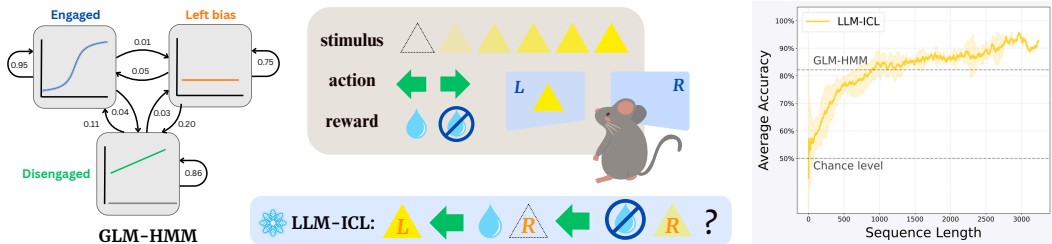

Figure 6: IBL dataset mice decision-making task. *(Left)* GLM-HMM model developed by neuroscientists. *(Middle)* A cartoon illustration of the task. A mouse observes a visual stimulus presented on one side of a screen, with one of six possible intensity levels. It then chooses a side, receiving a water reward if the choice matches the stimulus location. *(Right)* LLM ICL performance curve averaged across all animals, with 1-$\sigma$ error bar. Its prediction accuracy steadily increase with longer context window, exceeding the domain-specific model performance.

**Guideline 2:** *LLMs are data efficient in giving accurate next observation prediction in-context.* Pre-trained LLMs offer a remarkably data-efficient and accessible approach for next-observation prediction through ICL, particularly valuable when rapid insights are needed or when data for training bespoke models is scarce. Our analyses (Section 3.2) show that LLM ICL achieves strong predictive performance with fewer domain-specific assumptions than Baum-Welch (which faces non-convexity issues) and fewer training resources than specialized sequence models like LSTMs/RNNs (which require substantial data and careful tuning). LLMs therefore deliver immediate predictive capabilities with stable performance on limited data.

A key practical advantage of LLM ICL is accessibility: while traditional methods require substantial computational expertise, applying pre-trained LLMs simply involves formatting data as text prompts, dramatically lowering barriers to sequence analysis. We are not positioning LLM ICL as a universal replacement for meticulously tuned, domain-specific models. Rather, its strength lies in providing strong, often surprisingly near-optimal predictions (Section 2.3) without any task-specific parameter updates or fine-tuning. Our key observation is that general-purpose LLMs can effectively model HMM-generated sequences and real-world scientific data tasks for which they were not explicitly pre-trained on. This highlights vast untapped potential and suggests that future LLM ICL development could yield transformative scientific tools.

## 4.1 Real World Examples

We extend our synthetic HMM findings to real-world biological decision processes, focusing on two extensively studied behavioral neuroscience datasets. Understanding how animals make decisions and learn efficiently remains a fundamental challenge, with researchers investing tremendous effort in high-precision modeling to capture underlying cognitive mechanisms. These datasets serve as ideal testbeds given the neuroscience community's modeling efforts and the inherent connections between agentic decision-making and HMMs (Figure 6). We represent animal decisions as discrete token sequences and compare LLM ICL performance against established domain-specific models in predicting future actions.

**Decision-making Mice Dataset:** This dataset, developed by the International Brain Laboratory (IBL) [25], has gained significant traction for studying mouse behavior within the neuroscience community. A popular study [3] characterizes mice choice behavior as an interplay among multiple interleaved strategies governed by hidden states in a HMM. Their GLM-HMM model (Figure 6 left) achieved an average prediction accuracy of 82.2%, outperforming standard approaches like the classic lapse model by 2.8%. For scientists investigating animal decision-making, these performance improvements are significant for advancing model fidelity and experimental interpretation.

We compare GLM-HMM to in-context LLMs on data from 7 mice, following the descriptions in Ashwood et al. [3]. The experimental data consists of three components: stimulus, choice, and reward. For each trial, the mouse perceives a visual stimulus presented to their left or right, makes a choice by turning a steering wheel, and receives a water drop as reward when correct (Figure 6

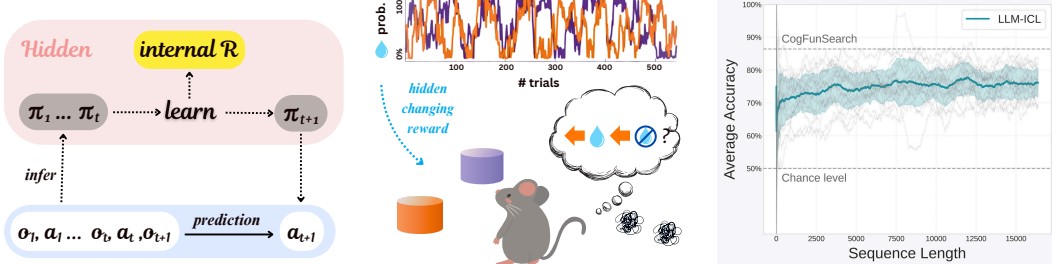

Figure 7: Rat reward-learning task. *(Left)* Analog agent learning to HMMs. *(Middle)* A cartoon illustration of the more challenging task. No stimulus is presented on either side; instead, the reward probabilities for left and right choices evolve independently via random walks. As the optimal choice changes over time, the rat must learn and adapt its decisions based solely on the history of past rewards. *(Right)* LLM ICL performance curve averaged across all animals, with 1-$\sigma$ error bar. Its performance curve improves only marginally with increasing context length.

middle). Each mouse is described by one sequence, composed of trials. The trials are ordered sequentially as they occurred during experiments.

Remarkably, when provided with a context of more than 1000 trials, LLM ICL consistently achieved higher prediction accuracy (average of 86.2%) than the expert-developed GLM-HMM (Figure 6 right). More importantly, the convergence trend of LLM ICL mirrors the in-context scaling we observed in synthetic experiments, particularly when entropy is relatively low. This observation suggests that mouse decision-making processes contain learnable structures that LLMs, even without task-specific training, can effectively identify and leverage for prediction.

**Reward-learning Rats Dataset:** The dataset from Miller et al. [36] allows us to explore LLM ICL capabilities on more complex learning behaviors. This task presents a significantly greater challenge than the IBL dataset for two primary reasons: first, animals receive no explicit stimuli to guide their choices towards potential rewards; secondly, the dataset captures the entire dynamic learning process itself, rather than behavior after learning. Consequently, the underlying behavioral dynamics are expected to be more complex and less stationary. To benchmark LLM ICL in this scenario, we compare its performance against a state-of-the-art model from recent work [10], which employed code generation and evolutionary search to discover interpretable symbolic programs well-fitted to this dataset. It is important to note that this state-of-the-art model results from an extensive, computationally intensive evolutionary search, setting a very high performance bar.

As shown in Figure 7, the prediction accuracy of LLM ICL on this dataset improves only marginally with increasing context length. This limited improvement parallels the ICL behavior we observed for synthetic HMMs characterized by high entropy and slow mixing rates. LLM ICL exhibits a substantial performance gap when compared to the specialized model. This outcome is consistent with our hypothesis that the underlying dynamics of this naturalistic learning process are complex, potentially pushing the limits of what current off-the-shelf LLMs can capture through ICL alone.

## 5    Related Works

**LLMs and In-Context Learning.** The surprising ability of LLMs to perform ICL [7] has led to significant interest in understanding its underlying mechanisms [11, 22, 53]. Several works [20, 52, 55] interpret ICL as implicit Bayesian inference, suggesting that LLMs naturally perform posterior updates through attention mechanisms. Theoretical works analyzing transformers' ability to model Markovian data [6, 13, 33, 40, 41] show that they can efficiently learn fully observed Markov chains. However, the transition from observable Markov sequences to latent-variable models like HMMs remains underexplored. Recent studies also evaluate LLMs' predictive performance on structured tasks, including dynamical systems [29], density estimation [30], and time series forecasting [19, 49]. While these works explore LLMs' empirical capabilities, they do not systematically analyze how intrinsic properties of underlying stochastic processes—such as mixing time and entropy—affect ICL performance. Our work fills this gap by providing a controlled study on synthetic HMMs and offering theoretical conjectures for the observed scaling trends in ICL performance.

**Spectral learning (SL) HMMs.** SL algorithms have emerged as a compelling tools for learning HMMs from observations, using method-of-moments to learn the spectral parameters. Several works [2, 21, 42] construct matrices with observations, perform singular value decomposition and projection to obtain the beliefs of the HMM operators. Recent work [31] improves the practicality of SL by projecting the probability beliefs onto simplex after every belief update. Despite these, SL algorithms have practical limitations and make assumptions on how observations carry information about the HMM dynamics. ICL seems to handle such issues by learning better observation operators without requiring the limiting assumptions of SL algorithms.

**Neuroscience and Animal Behavior.** Many neuroscience studies model animal behavior as HMMs [47, 50]. A common modern approach involves training data-specific RNNs and finding attractor dynamics [4, 23, 54], which can be highly data-inefficient. Large generative models have accelerated neuroscience discoveries, from data processing [44, 46] to model discovery [10]. In our work, we present a novel approach, leveraging the frontier of AI to help scientists understand their data, focusing on discrete behaviors [36, 43].

## 6    Discussion & Takeaways

**LLMs are surprisingly effective HMM learners through in-context learning:** We observe that LLM prediction accuracy often converges towards theoretical optimum achieved by the Viterbi algorithm, which knows true model parameters. Contrasting with iterative and computationally intensive traditional HMM estimation algorithms or trained neural architectures (e.g., LSTMs), LLM ICL offers simplicity as a tool. As demonstrated by the competitive performance to domain-specific models on real-world animal decision-making tasks, LLM ICL offers a new avenue for rapid data exploration. LLMs can serve as a "zero-shot statistical tool", enabling scientists to diagnose data complexity and generate future predictions without the overhead of extensive model development—addressing a common bottleneck in many scientific workflows.

**Existing gaps:** While we observe promising trends, our experiments also point to existing gaps in the broad application of LLM ICL. A primary bottleneck is the reliance on discrete tokenization, which poses challenges for modeling continuous, real-valued, or high-dimensional observations—such as neural recordings—within the ICL framework. Although our experiments successfully employed tokenization strategies for discrete sequences (see Appendix E.2 for ablations), adapting LLMs to handle continuous state-space models or direct real-valued inputs remains an open question. More-over, despite achieving high predictive accuracy, the inherently "black-box" nature of LLMs limits interpretability. While our findings demonstrate that LLMs can effectively model HMM dynamics, extracting explicit and interpretable parameters—such as transition or emission probabilities—from the model's internal representations is nontrivial. Yet, such interpretability is often central to the goals of scientists and practitioners seeking to understand underlying system dynamics. We hope this work lays the groundwork for future research into extending ICL to continuous domains and developing tools for extracting interpretable structure from LLMs.

**Call to action:** Realizing the full potential of LLMs and HMMs to advance our understanding of complex systems demands a multidisciplinary effort. There is a growing need for next-generation foundation models specifically designed to meet the challenges of scientific data—ranging from structured sequences to high-dimensional, continuous signals. Moving beyond adaptations of NLP-focused models, such advancements are critical not only for enabling more effective scientific analysis, but also for deepening our understanding of in-context learning and the structure embedded within human language corpora. Ultimately, this progress will be essential to unlocking the transformative potential of LLMs in scientific discovery across a broad range of disciplines.

## Acknowledgements

YD thanks Kristin Branson and Kimberly Stachenfeld for insightful technical discussions, and Owen Oertell for their support. ZG is supported by LinkedIn through the LinkedIn–Cornell Grant. This work was partly funded by NSF CCF 2312774, NSF OAC-2311521, NSF IIS-2442137, NSF IIS-2505098, a gift to the LinkedIn-Cornell Bowers CIS Strategic Partnership, and an AI2050 Early Career Fellowship program at Schmidt Science.

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

# Appendices

**Table of Contents**

## A   Additional Background on HMMs

In this section, we define in detail the HMM settings we are interested in, including the conditions for Markov chains to converge to unique stationary distributions. Recall that a HMM is characterized by the Markov chain's *initial state distribution* and its *state transitions*, along with the *emission probabilities* of an observation given the hidden state. With finitely many states and observations, without loss of generality, states take values in $\mathcal{X} = \{1, 2, \ldots, M\}$ while observations take values in $\mathcal{O} = \{1, 2, \ldots, L\}$. The initial state distribution is denoted as $\boldsymbol{\pi} \in \mathbb{R}^M$ with $\pi_j$ the probability of starting in state $j$, the state transitions are describe by the matrix $\mathbf{A} \in \mathbb{R}^{M \times M}$ with elements $a_{ij}$ the probability of transitioning to state $j$ from state $i$, and the emission matrix $\mathbf{B} \in \mathbb{R}^{M \times L}$ contains $b_{jl}$ the probability of observing $l$ when in hidden state $j$. The triple $\lambda = (\boldsymbol{\pi}, \mathbf{A}, \mathbf{B})$ completely parameterizes a finite-alphabet HMM.

Let $\{X_1, X_2, \ldots\}$ denote a discrete-time Markov chain taking values in $\mathcal{X}$ with transition matrix $\mathbf{A}$. Let $p_{ij}^{(n)} = \mathbb{P}(X_{t+n} = j | X_t = i)$ denote the $n$-step transition probability between states $i, j \in \mathcal{X}$. State $j$ is said to be *accessible* from state $i$ if there exists an integer $n \geq 1$ such that $p_{ij}^{(n)} > 0$. A subset $\mathcal{C} \subseteq \mathcal{X}$ is called *irreducible* if every pair of states $i, j \in \mathcal{C}$ is mutually accessible. The *period* of state $i$ is defined as $c(i) = \gcd\{n \geq 1 : p_{ii}^{(n)} > 0\}$, the greatest common divisor of all possible return times. State $i$ is *aperiodic* if $c(i) = 1$. A Markov chain is termed *geometrically ergodic* if it is irreducible and aperiodic, which guarantees convergence to a unique *stationary distribution* $\boldsymbol{\mu} \in \mathbb{R}^M$ satisfying $\boldsymbol{\mu} = \boldsymbol{\mu}\mathbf{A}$. The *mixing rate* $\rho \in [0, 1)$ is such that for all states $i, j \in \mathcal{X}$, there exists a constant $C \geq 0$ for which $|p_{ij}^{(n)} - \mu_j| \leq C\rho^n$ for all $n \geq 1$. For a finite-alphabet HMM, $\rho$ equals $\lambda_2$, the second-largest eigenvalue of $\mathbf{A}$. We run experiments on a few non-ergodic cases, while the majority of HMMs are with ergodic state transitions to avoid dependence on the initial state.

The *entropy* $H(X)$ of a discrete random variable $X$ is defined as $H(X) = -\sum_{x \in \mathcal{X}} p(x) \log p(x)$. A fundamental property of entropy is that conditioning reduces uncertainty: for any two random variables $X$ and $Y$, we have $H(X|Y) \leq H(X)$, with equality holding if and only if $X$ and $Y$ are statistically independent [12]. By applying the chain rule of entropy, the joint entropy of a stochastic process can be expressed as $H(X_1, X_2, \ldots, X_n) = \sum_{i=1}^n H(X_i | X_{i-1}, \ldots, X_1)$. For a Markov chain with stationary distribution $\boldsymbol{\mu}$, the *entropy rate* is defined as $H(\mathcal{X}) = \lim_{n \to \infty} \frac{1}{n} H(X_1, X_2, \ldots, X_n) = -\sum_{i,j} \mu_i a_{ij} \log a_{ij}$, which depends solely on the transition matrix $\mathbf{A}$. We additionally define the entropy of the emission matrix $\mathbf{B}$ as $-\sum_{j,l} \mu_j b_{jl} \log b_{jl}$, which quantifies the average uncertainty in observations given the underlying states. Although the entropy rate of the observation process in a HMM has no known closed-form expression, it can be bounded as $H(O_n | O_{n-1}, X_{n-1}, \ldots, O_1, X_1) \leq H(\mathcal{O}) \leq H(O_n | O_{n-1}, \ldots, O_1)$. As $\mathbf{A}$ defines transitions from $X_t$ to $X_{t+1}$, and $\mathbf{B}$ determines sampling $O_t$ from $X_t$, the entropies of $\mathbf{A}$ and $\mathbf{B}$ combined help us to control the entropy lower bound of the sampled HMM sequence.

# B Additional Details of Experimental Setup

**Construct A with specific mixing rate, entropy, and steady state distribution.** For an ergodic Markov chain that converges to a unique stationary distribution, the stochastic matrix $\mathbf{A}$ can be decomposed into eigenvalues and eigenvectors with the ordering shown in Figure 8, where $\vec{1} \in \mathbb{R}^M$ is a vector of ones, $\lambda_2$ is the second-largest eigenvalue of $\mathbf{A}$, and $\boldsymbol{\mu}$ is the stationary distribution [15]. We leverage this decomposition to construct $\mathbf{A}$ with predefined $\lambda_2$ and $\boldsymbol{\mu}$. To determine the remaining eigenvalues and eigenvectors, we formulate an optimization problem based on the following requirements: (1) all entries of $\mathbf{A}$ are non-negative; (2) each row of $\mathbf{A}$ sums to 1; (3) $\mathbf{U}^{-1}\mathbf{U} = 1$; and (4) all remaining eigenvalues have magnitudes not exceeding $\lambda_2$. The optimization problem has the following form, where we translate the constraints above into penality terms.

$$\min_{\lambda_{3:M}, \mathbf{V}_2} \sum_{i,j=1}^{M} \max\{-a_{ij}, 0\} + \sum_{j=1}^{M} \left( \left( \sum_{i=1}^{M} a_{ij} \right) - 1 \right)^2 + \sum_{i,j=1}^{M} (\mathbf{VU} - \mathbf{I})_{ij}^2 + \sum_{i=3}^{M} \max\{\lambda_i - \lambda_2, 0\}$$

$$\text{s.t.} \quad \mathbf{A} = \mathbf{V}\mathrm{diag}(1, \lambda_2, \lambda_3, ..., \lambda_M)\mathbf{U}, \quad \mathbf{V} = \begin{bmatrix} \mathbf{1} & \mathbf{V}_2 \end{bmatrix}, \quad \mathbf{U} = \begin{bmatrix} \boldsymbol{\mu} \\ \mathbf{V}_2^\dagger \end{bmatrix}$$

This is a nonconvex problem, which we solve using first order methods with `pytorch`. We randomly initialize the free variables $\lambda_3, ..., \lambda_M$ and $\mathbf{V}_2$ and then run 5000 iterations of Adam with step size 0.01 and default values for other parameters. After the optimizer terminates, we reject instances which do not satisfy the constraints exactly. By initializing with multiple random seeds, we generate matrices spanning the desired entropy spectrum.

$$A = U^{-1}\Lambda U =$$

Figure 8: The singular value decomposition of ergodic unichain Markov matrix $\mathbf{A}$. The darker shaded region is pre-defined for our controlled experiments. The lighter shaded region is randomly initialized and calculated using a neural network.

**Steady state distribution.** We construct steady state distributions with varying skewness using the Beta distribution with $\alpha = 1$ and different values of $\beta$. When $\alpha = 1$ and $\beta = 1$, the resulting steady state distribution is uniform. As $\beta$ increases, the distribution becomes increasingly skewed toward smaller state indices. Unless otherwise specified (Appendix D.3), we use a uniform steady state distribution as the default configuration.

**Entropy for visualizations.** The entropy definitions $H(\mathbf{A})$ and $H(\mathbf{B}, \boldsymbol{\mu})$ we introduced in Section 2.1 are used for constructing HMM parameters and sampling trajectories. For graphing Figure 4 *(Left)*, we define normalized entropy considering both matrices:

$$\tilde{H}(\mathbf{A}, \mathbf{B}, \boldsymbol{\mu}) = \frac{H(\mathbf{A}) + H(\mathbf{B}, \boldsymbol{\mu})}{\log M + \log L}.$$

We define $\tilde{H}(\mathbf{A}) = H(\mathbf{A})/\log M$ and $\tilde{H}(\mathbf{B}, \boldsymbol{\mu}) = H(\mathbf{B}, \boldsymbol{\mu})/\log L$ for Figure 4 *(Middle)* and *(Right)*.

**$T$ is when LLM converges to Viterbi.** The concept of "convergence", though intuitive to human eyes, requires a specific numerical definition for plots like Figure 4. We define convergence as the point where two conditions are simultaneously satisfied: (i) the accuracy difference between Viterbi and LLM is within 0.025, and (ii) LLM achieves at least 95% of Viterbi's accuracy. We use both constant and relative thresholds to ensure a strict convergence definition that accounts for different baseline performance levels across experimental conditions.

**Hellinger Distance.** For two discrete probability distributions $P, Q \in \mathbb{R}^L$, the Hellinger distance is defined as

$$D_{Hellinger}(P, Q) = \frac{1}{\sqrt{2}} \sqrt{\sum_{i=1}^{L} (\sqrt{P_i} - \sqrt{Q_i})^2}.$$

## C Details of Benchmark Models

In this section, we provide descriptions and pseudocode for the benchmark models we use (Table 1). The executable code for all methods are included in supplemental materials.

---

**Algorithm 1:** Viterbi Algorithm

---

**Input:** States $\mathcal{X} = \{1, 2, \ldots, M\}$, initial distribution $\boldsymbol{\mu}$, transition matrix $\mathbf{A}$, emission matrix $\mathbf{B}$, and observation sequence $\{o_1, \ldots, o_T\}$.

**Output:** Most likely state sequence path $= \{x_1, \ldots, x_T\}$

**Initialization:** $\mathbf{P}[0][s] \leftarrow \boldsymbol{\mu}[s] \cdot \mathbf{B}[s][o_1]$ for all $s \in \mathcal{X}$;

**Forward recursion: for** $t = 1$ **to** $T - 1$ **do**

    **for** $s \in \mathcal{X}$ **do**

        $\mathbf{P}[t][s] \leftarrow \max_{r \in \mathcal{X}} \{\mathbf{P}[t-1][r] \cdot \mathbf{A}[r][s] \cdot \mathbf{B}[s][o_t]\}$;

        $\mathbf{Q}[t][s] \leftarrow \arg\max_{r \in \mathcal{X}} \{\mathbf{P}[t-1][r] \cdot \mathbf{A}[r][s] \cdot \mathbf{B}[s][o_t]\}$;

    **end**

**end**

**Backtracking:** $\mathrm{path}[T-1] \leftarrow \arg\max_{s \in \mathcal{X}} \mathbf{P}[T-1][s]$;

$\mathrm{path}[t] \leftarrow \mathbf{Q}[t+1][\mathrm{path}[t+1]]$ for $t = T-2, \ldots, 0$;

**return** *path*

---

**Viterbi algorithm.** The Viterbi algorithm is a dynamic programming technique for efficiently finding the most likely sequence of hidden states in a Markov model, given a sequence of observations. It iteratively computes the highest probability path to each state at time $t$ by considering all possible predecessor states at time $t-1$, their transition probabilities, and the emission probabilities of the current observation. Rather than exhaustively evaluating all $M^T$ possible state sequences, Viterbi maintains only the $M$ most promising paths at each time step, storing both their probabilities and the penultimate states that maximize these probabilities. After computing probabilities for all time steps, the algorithm traces backward from the most probable final state to reconstruct the optimal state sequence. We use the most probable final state and ground-truth $\mathbf{A}$ and $\mathbf{B}$ to calculate the prediction distribution of the next observation.

---

**Algorithm 2:** Compute $P(O_{t+1}|O_{t-k:t})$

---

**Input:** States $\mathcal{X} = \{1, 2, \ldots, M\}$, initial distribution $\boldsymbol{\mu}$, transition matrix $\mathbf{A}$, emission matrix $\mathbf{B}$, and observation sequence $\{o_{t-k}, \ldots, o_t\}$.

**Output:** Probability of next observation $P(o_{t+1}|o_{t-k:t})$

**Forward pass over observation window:** $\boldsymbol{\alpha}_{t-k}[s] \leftarrow \mathbf{B}[s][o_{t-k}] \cdot \boldsymbol{\mu}[s]$ for all $s \in \mathcal{X}$;

$\boldsymbol{\alpha}_i[s] \leftarrow \mathbf{B}[s][o_i] \cdot \sum_{r \in \mathcal{X}} \mathbf{A}[r][s] \cdot \boldsymbol{\alpha}_{i-1}[r]$ for $i = t-k+1, \ldots, t, s \in \mathcal{X}$;

**Normalize to get posterior:** $P(s|o_{t-k:t}) \leftarrow \frac{\boldsymbol{\alpha}_t[s]}{\sum_{s' \in \mathcal{X}} \boldsymbol{\alpha}_t[s']}$ for all $s \in \mathcal{X}$;

**Prediction step:** $P(s|o_{t-k:t}) \leftarrow \sum_{r \in \mathcal{X}} \mathbf{A}[r][s] \cdot P(r|o_{t-k:t})$ for all $s \in \mathcal{X}$;

**Marginalize over states:** $P(o_{t+1}|o_{t-k:t}) \leftarrow \sum_{s \in \mathcal{X}} \mathbf{B}[s][o_{t+1}] \cdot P(s|o_{t-k:t})$;

**return** $P(o_{t+1}|o_{t-k:t})$

---

**Optimal inference with truncated memory** $P(O_{t+1}|O_{t-k:t})$**.** The forward-based prediction algorithm computes the probability of the next observation in a hidden Markov model by using a three-step approach. First, it calculates the posterior distribution over current hidden states via the forward algorithm, recursively processing the observation window while accounting for transitions and emissions. Second, it projects this belief state forward by applying the transition matrix to compute the distribution over next possible states. Finally, it determines $P(o_{t+1}|o_{t-k:t})$ by marginalizing over all possible next states, weighting each by its emission probability.

**Baum-Welch algorithm.** The Baum-Welch algorithm is an expectation-maximization method for estimating hidden Markov model parameters. It iteratively alternates between computing state posteriors $\gamma_t(s)$ and transition posteriors $\xi_t(s, r)$ via forward-backward recursion (E-step), and updating parameters to maximize likelihood (M-step): setting the initial distribution to $\gamma_1$, transition

---

**Algorithm 3:** Baum-Welch Algorithm

---

**Input:** States $\mathcal{X} = \{1, 2, \ldots, M\}$, observations $\mathcal{Y} = \{1, 2, \ldots, L\}$, observation sequence
$\{o_1, \ldots, o_T\}$, initial parameters $\boldsymbol{\mu}^{(0)}, \mathbf{A}^{(0)}, \mathbf{B}^{(0)}$, and threshold $\epsilon$.

**Output:** Refined parameters $\boldsymbol{\mu}, \mathbf{A}, \mathbf{B}$

**Initialize:** $\boldsymbol{\mu} \leftarrow \boldsymbol{\mu}^{(0)}, \mathbf{A} \leftarrow \mathbf{A}^{(0)}, \mathbf{B} \leftarrow \mathbf{B}^{(0)}, \mathcal{L}_{\text{prev}} \leftarrow -\infty$;

**repeat**

    $\mathcal{L}_{\text{prev}} \leftarrow \mathcal{L}$;

    **E-Step:**

    **Forward pass:** $\boldsymbol{\alpha}_1[s] \leftarrow \mathbf{B}[s][o_1] \cdot \boldsymbol{\mu}[s]$ for all $s \in \mathcal{X}$;

    $\boldsymbol{\alpha}_t[s] \leftarrow \mathbf{B}[s][o_t] \cdot \sum_r \mathbf{A}[r][s] \cdot \boldsymbol{\alpha}_{t-1}[r]$ for $t = 2, \ldots, T, s \in \mathcal{X}$;

    $\mathcal{L} \leftarrow \sum_s \boldsymbol{\alpha}_T[s]$;

    **Backward pass:** $\boldsymbol{\beta}_T[s] \leftarrow 1$ for all $s \in \mathcal{X}$;

    $\boldsymbol{\beta}_t[s] \leftarrow \sum_r \mathbf{A}[s][r] \cdot \mathbf{B}[r][o_{t+1}] \cdot \boldsymbol{\beta}_{t+1}[r]$ for $t = T-1, \ldots, 1, s \in \mathcal{X}$;

    **Expected counts:** $\gamma_t[s] \leftarrow \frac{\boldsymbol{\alpha}_t[s] \cdot \boldsymbol{\beta}_t[s]}{\mathcal{L}}$ for $t = 1, \ldots, T, s \in \mathcal{X}$;

    $\xi_t[s][r] \leftarrow \frac{\boldsymbol{\alpha}_t[s] \cdot \mathbf{A}[s][r] \cdot \mathbf{B}[r][o_{t+1}] \cdot \boldsymbol{\beta}_{t+1}[r]}{\mathcal{L}}$ for $t = 1, \ldots, T-1, s, r \in \mathcal{X}$;

    **M-Step:** $\boldsymbol{\mu}[s] \leftarrow \gamma_1[s]$ for all $s \in \mathcal{X}$;

    $\mathbf{A}[s][r] \leftarrow \frac{\sum_{t=1}^{T-1} \xi_t[s][r]}{\sum_{t=1}^{T-1} \gamma_t[s]}$ for all $s, r \in \mathcal{X}$;

    $\mathbf{B}[s][v] \leftarrow \frac{\sum_{t=1}^{T} \gamma_t[s] \cdot \mathbf{1}(o_t = v)}{\sum_{t=1}^{T} \gamma_t[s]}$ for all $s \in \mathcal{X}, v \in \mathcal{Y}$;

**until** $|\mathcal{L} - \mathcal{L}_{prev}| < \epsilon$;

**return** $\boldsymbol{\mu}, \mathbf{A}, \mathbf{B}$

---

probabilities to normalized expected transitions, and emission probabilities to normalized observation counts per state. This process continues until the log-likelihood converges, yielding locally optimal parameters that maximize the probability of generating the observed sequence. We use the learned parameters to predict next observation similar to the Viterbi algorithm.

---

**Algorithm 4:** $n$-gram Based Next-Observation Prediction

---

**Input:** Observation sequence $O = \{o_1, \ldots, o_T\}$, context length $n-1$, smoothing parameter $\delta$

**Output:** $n$-gram model for predicting $P(o_t | o_{t-(n-1):t-1})$

**Count extraction:** $\text{counts}_n, \text{counts}_{n-1} \leftarrow$ empty associative arrays;

**for** $t = n-1$ **to** $T - 1$ **do**

    context $\leftarrow [o_{t-(n-1)}, \ldots, o_{t-1}]$;

    Increment $\text{counts}_n[\text{context} \oplus o_t]$ and $\text{counts}_{n-1}[\text{context}]$;

**end**

**Model construction with smoothing:** $V \leftarrow$ number of unique symbols in $O$;

**for** *each observed context $c$ in $\text{counts}_{n-1}$* **do**

    **for** *each unique observation $o$ in $O$* **do**

        $\text{model}[c, o] \leftarrow \frac{\text{counts}_n[c \oplus o] + \delta}{V \cdot \delta + \text{counts}_{n-1}[c]}$;

    **end**

**end**

**Back-off for unseen contexts:** $P_{unif}(o) \leftarrow \frac{1}{V}$ for all $o$;

$P(o_t | o_{t-(n-1):t-1}) \leftarrow \begin{cases} \text{model}[[o_{t-(n-1)}, \ldots, o_{t-1}], o_t], & \text{if context observed} \\ P_{unif}(o_t), & \text{otherwise} \end{cases}$;

**return** *model*

---

$n$**-gram.** $n$-gram models provide an elegant, computationally efficient framework for next-observation prediction in Markov chain processes by directly estimating conditional probabilities from observed sequences. These models embody the Markov assumption that $P(O_{t+1} | O_{1:t}) \approx P(O_{t+1} | O_{t-n+2:t})$, making them particularly effective for stochastic processes where future states depend only on a limited history of previous states. For first-order Markov chains, bigram models ($n = 2$) precisely capture the underlying transition dynamics, while higher-order dependencies can be modeled by increasing $n$.

**Algorithm 5:** Trained Neural Networks for Single Sequence Prediction

---

**Input:** sequence $O = \{o_1, \ldots, o_T\}$; vocab size $V$; context window $K$; model family $\mathcal{F}$;
     parameters $\theta$; optimizer $\mathcal{A}$; epochs $E$

**Output:** trained $\theta$ approximating $P(o_{t+1} \mid o_{1:t})$

**Model:** Token embedding $E : \{1, \ldots, V\} \to \mathbb{R}^d$;   optional position map $\Pi : \{1, \ldots, K\} \to \mathbb{R}^d$;

Core network $f_\theta : (\mathbb{R}^d)^{\leq K} \to \mathbb{R}^V$ (*e.g.*, LSTM/GRU/Transformer/MLP with causal constraint);

Output distribution $p_\theta(\cdot \mid x) = \mathrm{softmax}(f_\theta(x))$;   loss CE (cross-entropy);

**Training: for** *epoch* $= 1$ **to** $E$ **do**

    **for** $t = 1$ **to** $T - 1$ **do**

        $s \leftarrow \max(1, t - K + 1)$;   $x \leftarrow E(O_{s:t}) \oplus \Pi_{1:|O_{s:t}|}$ ;       // embed (+ positions)

        $y \leftarrow O_{t+1}$;

        $\ell \leftarrow f_\theta(x)$ ;                // logits for next token

        $L_t \leftarrow \mathrm{CE}(\mathrm{softmax}(\ell), y)$;

        $\theta \leftarrow \mathcal{A}(\theta, \nabla_\theta L_t)$ ;              // optimizer step

    **end**

**end**

**Inference:**;

Given prefix $O_{1:t}$, return $P(o_{t+1} \mid O_{1:t}) = \mathrm{softmax}(f_\theta(E(O_{s:t}) \oplus \Pi))$ with
$s = \max(1, t - K + 1)$.;

**return** $\theta$

---

**RNN LSTM.** LSTM networks are specialized recurrent neural architectures designed to model sequential data through memory cells regulated by input, forget, and output gates. These gates control information flow, allowing LSTMs to selectively retain relevant historical patterns while discarding irrelevant information. LSTMs excel at next-observation prediction tasks by capturing both short-term correlations and long-term dependencies in the observation history. The network processes a window of prior observations sequentially, updating its hidden state to encode temporal patterns, then projects this state through a softmax layer to generate a probability distribution over possible next observations—making LSTMs particularly effective for forecasting future values in time series where the prediction depends on complex patterns spanning multiple time scales.

In our experiments, we set the number of observations as the vocab size, use a two-layer LSTM with an embedding dimension of 16 and a hidden dimension of 8, and train for 10 epochs using the Adam optimizer with a learning rate of 1e-3. The results are averaged over 16 sequences.

**Transformer.** Transformers predict the next observation using *causal self-attention*, which reweights the visible prefix without recurrence. Given embedded inputs $X$ (tokens + positions), define $Q = XW_Q$, $K = XW_K$, $V = XW_V$ (queries, keys, values as learned linear projections). Attention is

$$\mathrm{Attn}(Q, K, V) = \mathrm{softmax}\left(\frac{QK^\top}{\sqrt{d_k}} + M\right) V,$$

where $M$ is a causal mask that blocks future positions. Multi-head attention computes this in parallel across heads and concatenates the results. A feed-forward layer then maps the attended representations, and the state at position $t$ is projected to logits; the next-token distribution is $P(O_{t+1} \mid O_{1:t}) = \mathrm{softmax}(\mathrm{logits}_t)$.

In our experiments, we set the number of observations as the vocab size, use a two-layer Transformer with an embedding dimension of 16, a hidden dimension of 8, and 5 attention heads, and train for 10 epochs using the Adam optimizer with a learning rate of 1e-3. The results are averaged over 16 sequences.

# D    Additional Synthetic Experiment Results

This section presents additional results from synthetic experiments. All methods are evaluated using the average performance over 4,096 sequences, with the exception of LSTM and Transformer, which are evaluated on 16 sequences due to their high computational cost. Consequently, the LSTM and Transformer results exhibit higher variance. Nonetheless, in metrics such as Hellinger distance—which account for the full output distribution rather than relying solely on the argmax for accuracy—LSTM and Transformer underperforms compared to the LLM most of the time.

## D.1    Varying Entropy of A

In this section, we present detailed results on varying the entropy of $A$ matrix over 4/8/16 states and emissions, reporting accuracies and Hellinger distances.

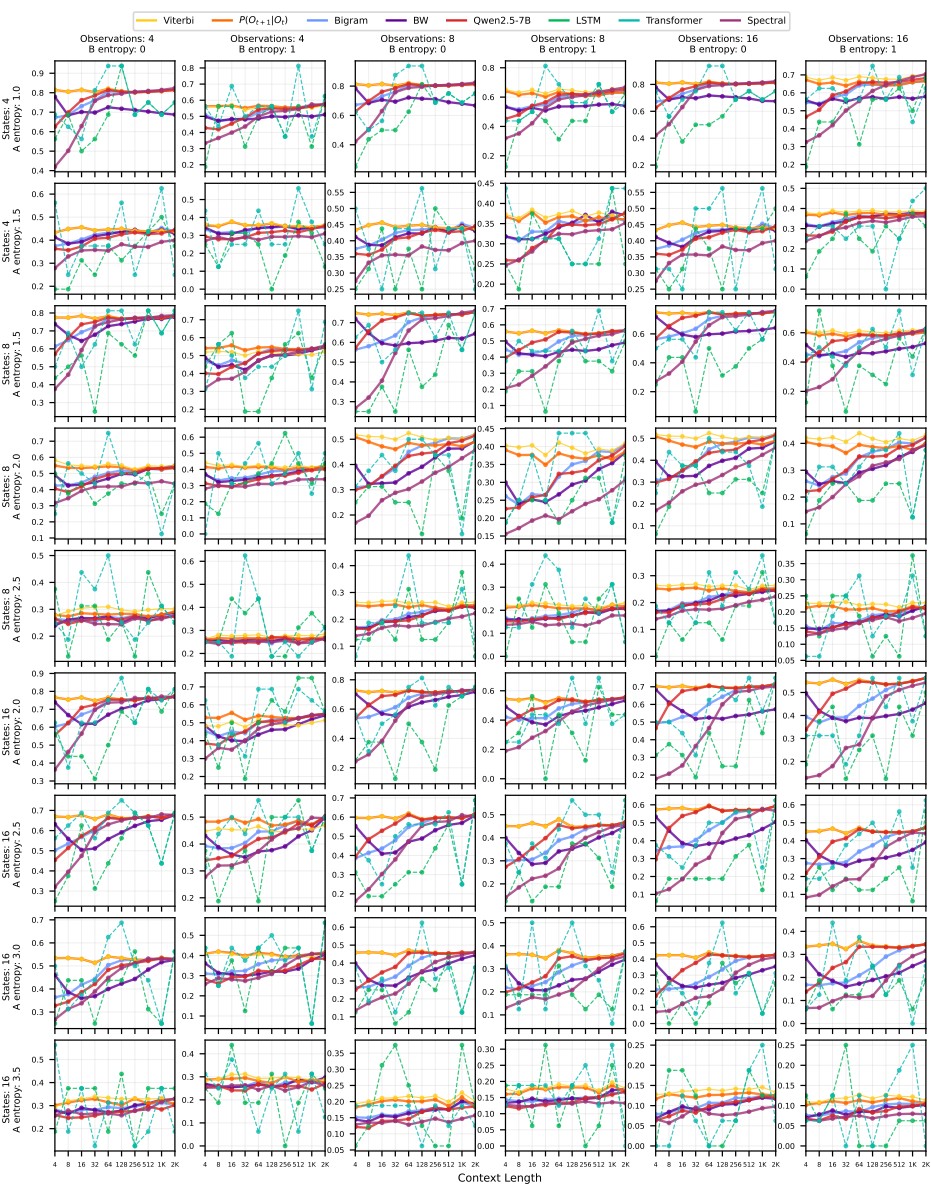

Figure 9: Accuracies of six methods across different **A** entropy, **B** entropy, number of states, and number of emissions with $\lambda_2 = 0.75$ and uniform steady state distribution.

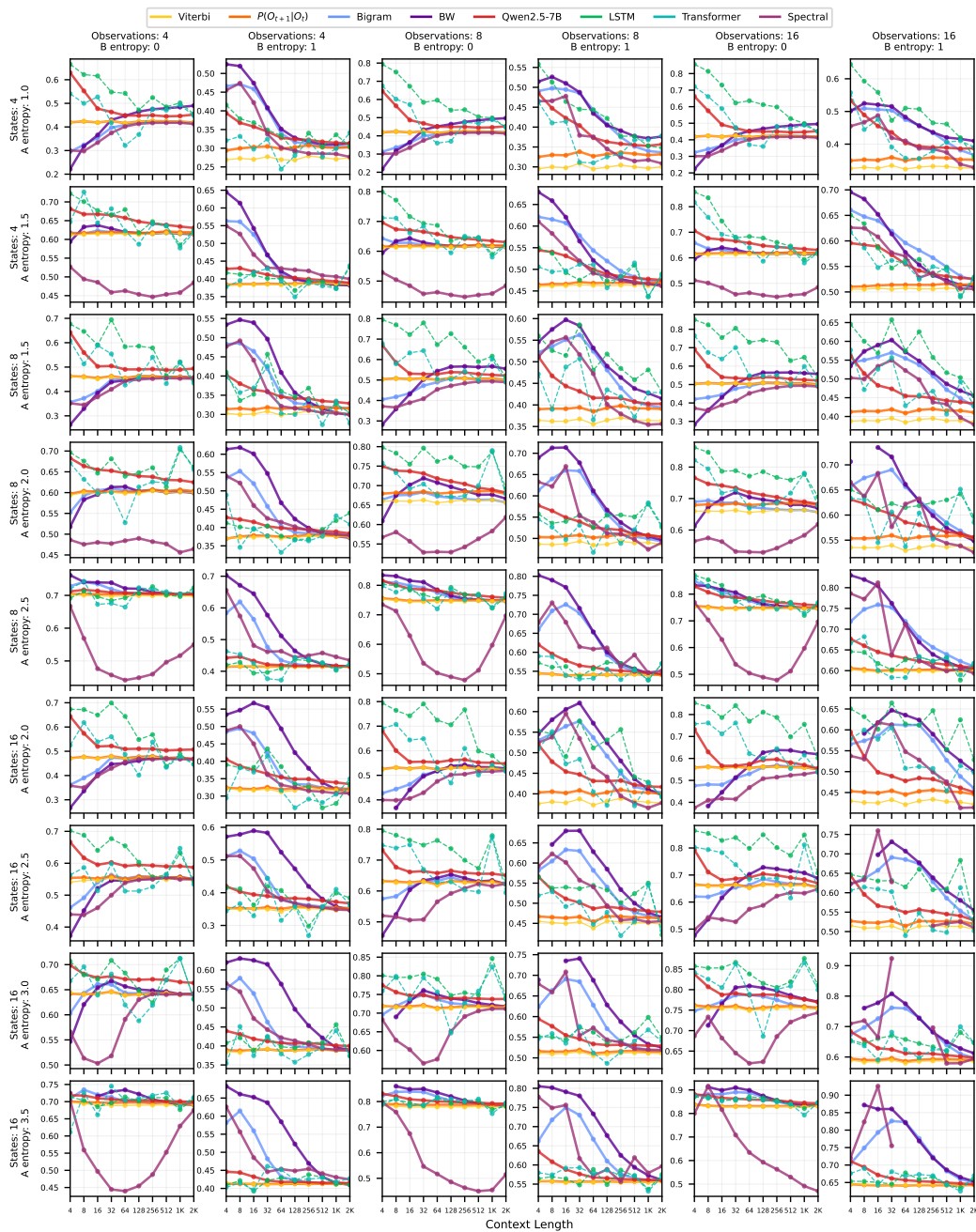

Figure 10: Hellinger distances of six methods across different $\mathbf{A}$ entropy, $\mathbf{B}$ entropy, number of states, and number of emissions with $\lambda_2 = 0.75$ and uniform steady state distribution.

## D.2 Varying Mixing Rate of A

In this section, we present detailed results on varying the mixing rate ($\lambda_2$) of $A$ matrix over 4/8/16 states and emissions, reporting accuracies and Hellinger distances.

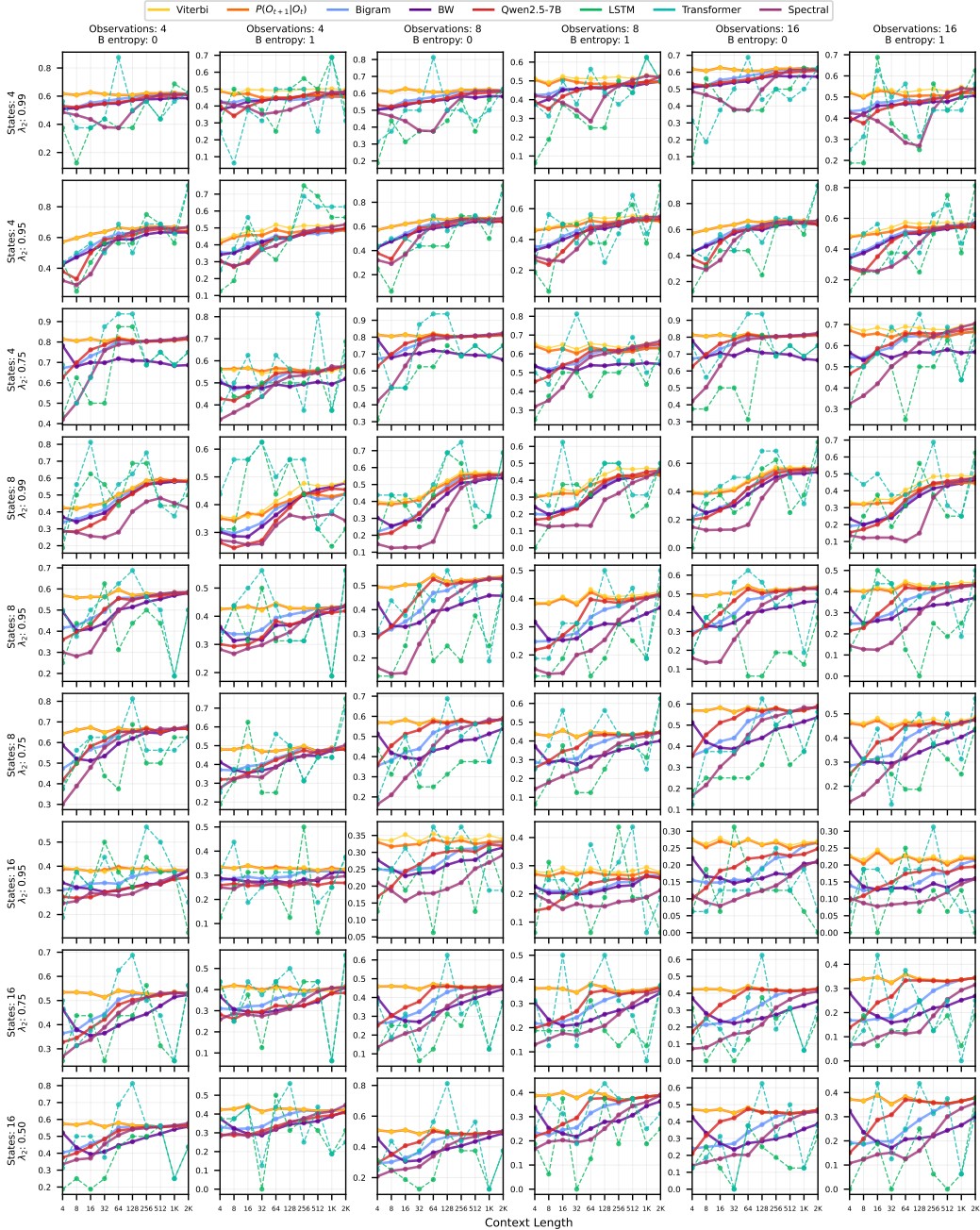

Figure 11: Accuracies of six methods across different mixing rates ($\lambda_2$), **B** entropy, number of states, and number of emissions with uniform steady state distribution and (1, 2, 3) **A** entropy for (4, 8, 16) states respectively.

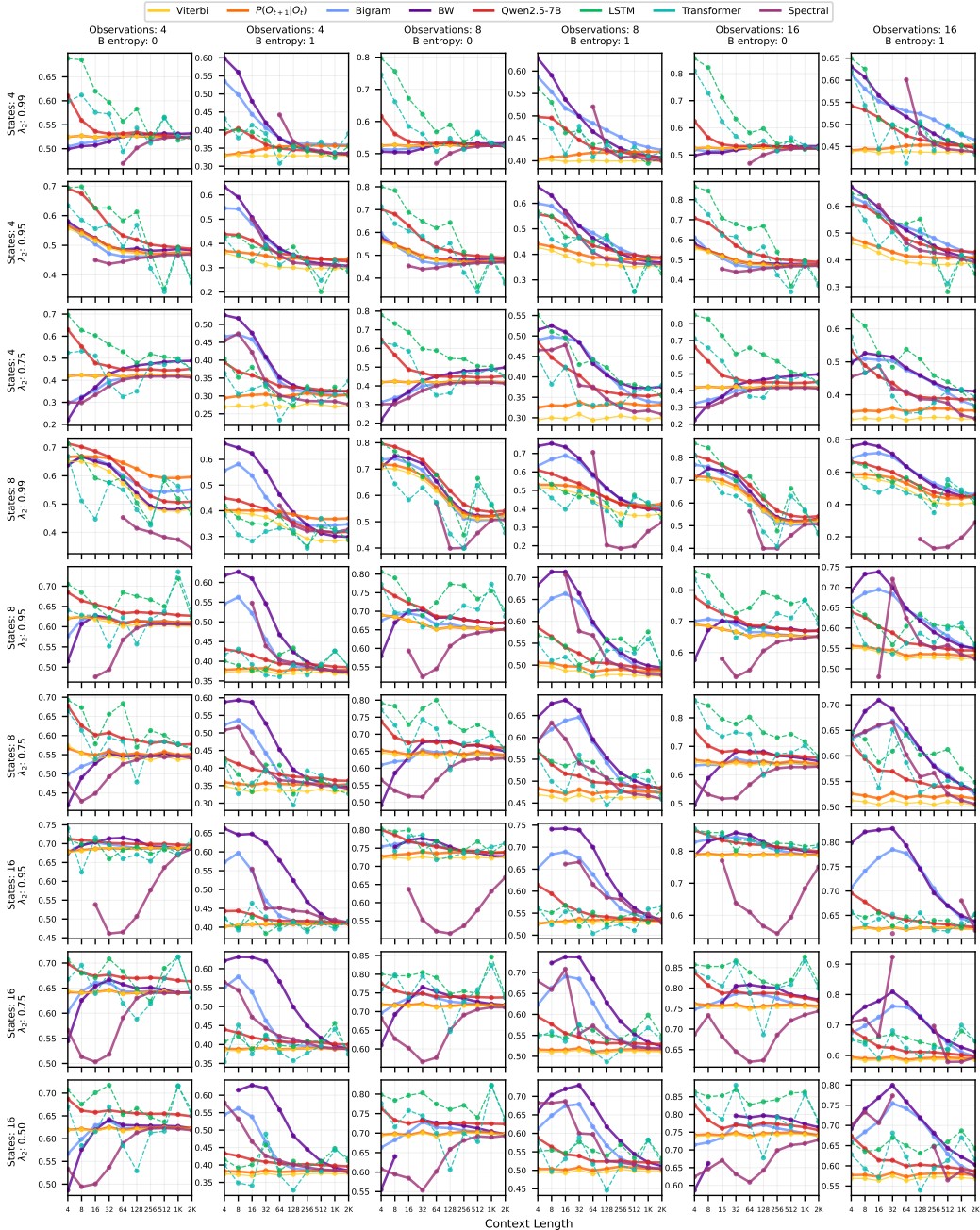

Figure 12: Hellinger distances of six methods across different mixing rates ($\lambda_2$), **B** entropy, number of states, and number of emissions with uniform steady state distribution and (1, 2, 3) **A** entropy for (4, 8, 16) states respectively.

## D.3 Varying Steady State Distribution of $A$

In this section, we present detailed results on varying the steady state distributions of $A$ matrix over 4/8/16 states and emissions, reporting accuracies and Hellinger distances. We construct steady states with different skewness using Beta distribution with $\alpha = 1$. Notably, with $\alpha = 1$ and $\beta = 1$, the steady state distribution is uniform. As we increase $\beta$, the distribution becomes more skewed. We test with $\beta = 1, 2, 3$, representing uniform, skewed, and very skewed respectively.

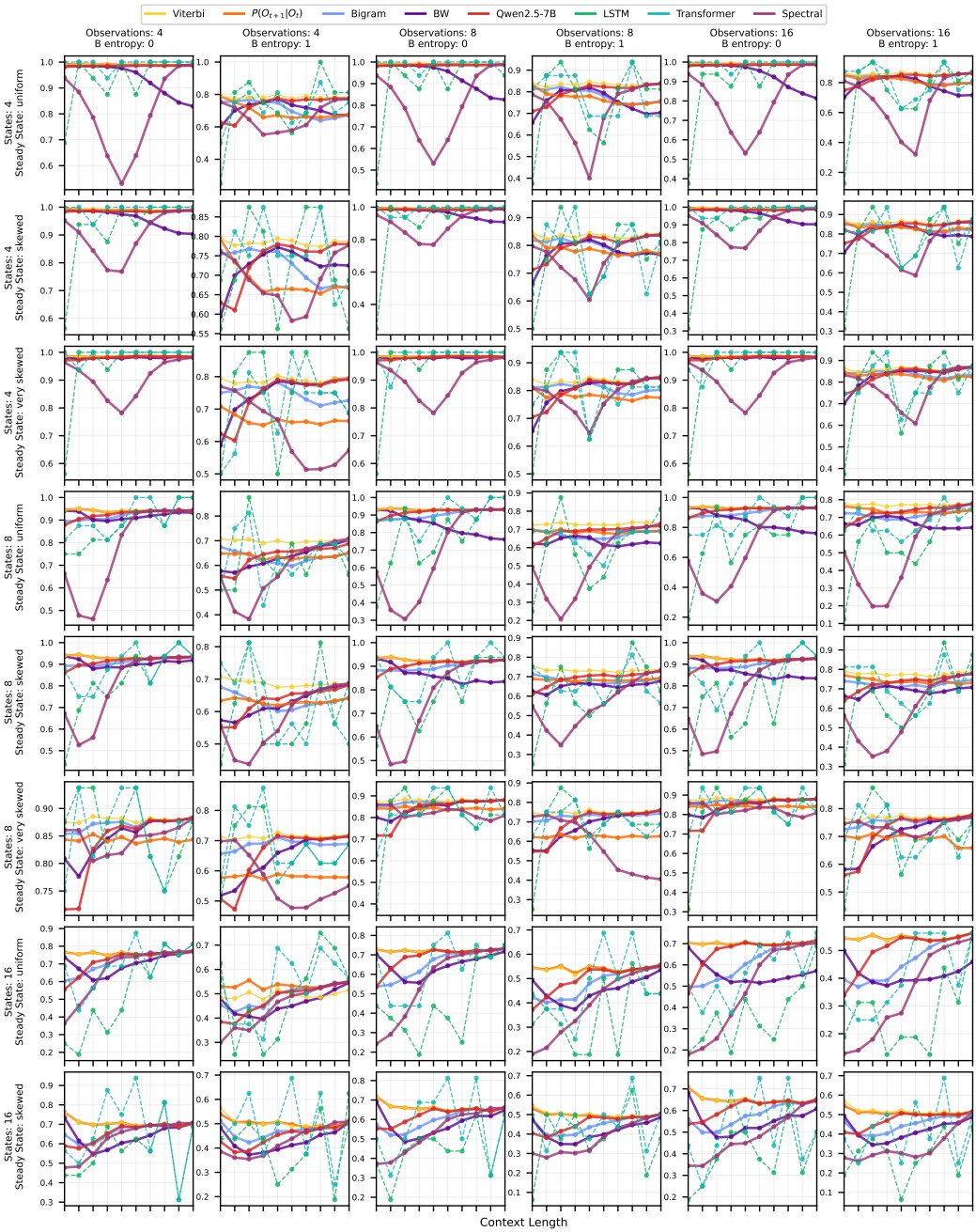

Figure 13: Accuracies of six methods across different steady state distributions, $\mathbf{B}$ entropy, number of states, and number of emissions with $(0, 0.5, 2)$ $\mathbf{A}$ entropy for $(4, 8, 16)$ states respectively and $(0.99, 0.95, 0.75)$ $\lambda_2$ for $(4, 8, 16)$ states respectively.

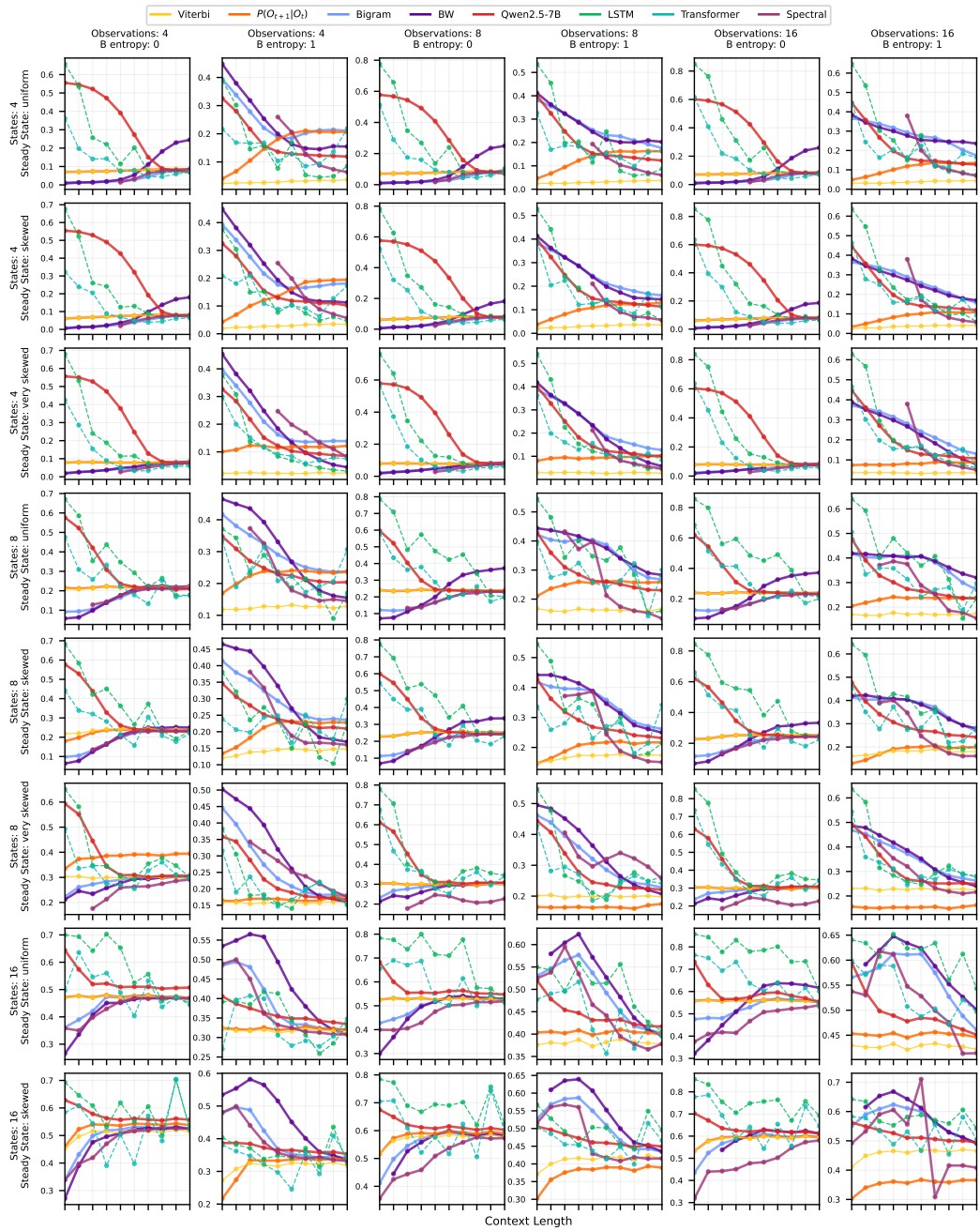

Figure 14: Hellinger distances of six methods across different steady state distributions, **B** entropy, number of states, and number of emissions with (0, 0.5, 2) **A** entropy for (4, 8, 16) states respectively and (0.99, 0.95, 0.75) $\lambda_2$ for (4, 8, 16) states respectively.

### D.4 Discussions

**When LLMs fail to converge.** While LLMs converge to Viterbi performance efficiently under most HMM parameter settings (scaling trends summarized in Section 3.1), we identify two conditions where convergence fails or proceeds exceptionally slowly. First, when entropy of $\mathbf{A}$ or $\mathbf{B}$ is approaches its maximum ($\log M$ or $\log L$ respectively), the prediction accuracy gap $\varepsilon$ at context length 2048 remains substantial. For instance, in the last row of Figure 9 with $M = 16$ and the entropy of $\mathbf{A}$ is 3.5 (near the maximum of $\log 16 = 4$), the LLM (Qwen2.5-7B) exhibits gradual convergence with a persistent gap. Second, when mixing is slow ($\lambda_2$ approaches 1), such as in the third-to-last row of Figure 11 with $M = 16$ and $\lambda_2 = 0.95$, a performance gap persists even at maximum context length.

Importantly, the Viterbi algorithm also struggles under these challenging conditions. Under high entropy, as shown in the last row of Figure 9, Viterbi accuracy barely exceeds random prediction (0.25/0.125/0.0625 for $L = 4/8/16$). Under slow mixing, such as in the fourth row of Figure 11 with $M = 8$ and $\lambda_2 = 0.99$, Viterbi algorithm requires context length 512 to achieve peak performance. These results demonstrate that LLM performance degradation under high entropy and slow mixing conditions reflects fundamental limits of stochastic system learnability—arising from random dynamics and long-range dependencies—that affect even optimal inference methods.

**Monotonicity of LLM performance with respect to context length.** We observe that LLM performance almost always improves monotonically with longer context length—a property notably absent in other learning baselines. Even excluding LSTM from this comparison (due to high variance from averaging over fewer sequences, as discussed in the first paragraph in Appendix D), both Baum-Welch and bigram models lack monotonic convergence behavior. For Baum-Welch, the accuracy graphs (Figures 9, 11, and 13) reveal multiple cases where performance "dips" and recovers, or deteriorates as context length increases. The Hellinger distance graphs (Figures 10, 12, and 14) provide clearer evidence that both BW and bigram exhibit non-monotonic learning patterns. In most cases, LLM Hellinger distance decreases monotonically, while BW and bigram display erratic behavior: sometimes experiencing early-context "bumps", other times starting very close to the ground truth emission distribution (occasionally even closer than the oracle Viterbi by empirical chance) before gradually converging to statistically sound distributions. Importantly, when BW or bigram achieve lower Hellinger distances, this does not necessarily indicate better performance—the corresponding prediction accuracy graphs often show poor results, highlighting the distinction between distributional similarity and predictive capability.

**When (normalized) entropy $\tilde{H}$ is held constant, varying the number of states does not affect the LLM convergence rate.** We provide concrete evidence for this claim in Figure 9, where rows 1, 3, and 6 all have the same normalized entropy $\tilde{H}(\mathbf{A}) = 0.5$. Across each column, the Qwen2.5-7B convergence curves for these three rows exhibit nearly identical shapes, demonstrating that the convergence rate depends primarily on normalized entropy rather than absolute state space size.

We emphasize that convergence rate differs from convergence target—the Viterbi performance. While the rate of improvement remains consistent across different state space sizes (when normalized entropy is fixed), larger state spaces result in lower achievable prediction accuracy due to increased task difficulty.

# E    Ablations on LLMs

In this section, we provide the results on the families, sizes, and tokenization of the LLMs.

## E.1    LLM Size

We compare Qwen and Llama model families with seven different models. We found that their performances are similar, with slight degradation when the model size is small.

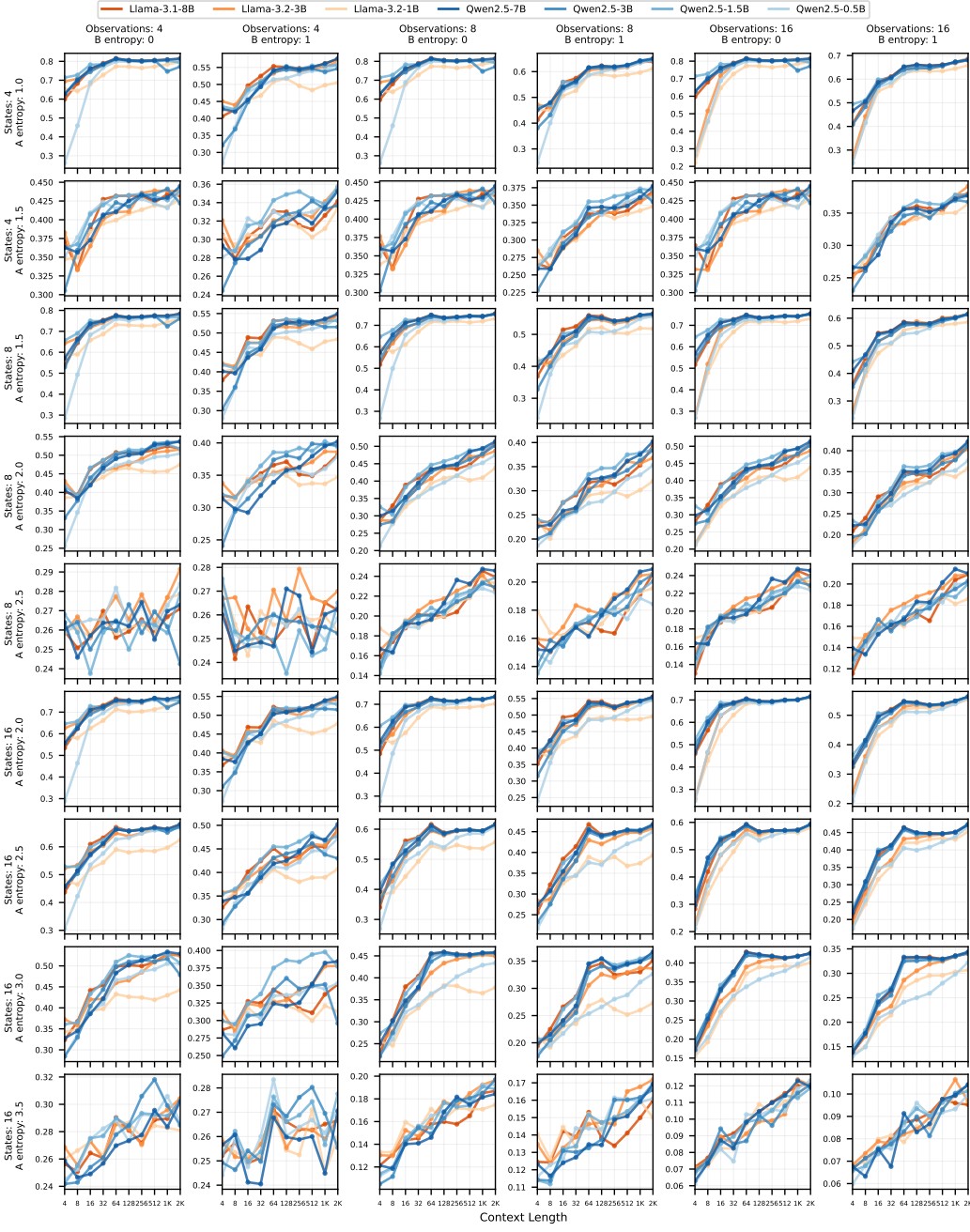

Figure 15: Accuracies of seven models across different **A** entropy, **B** entropy, number of states, and number of emissions with $\lambda_2 = 0.75$ and uniform steady state distribution. Lighter color represents smaller models. The two smallest models from each family have suboptimal performance.

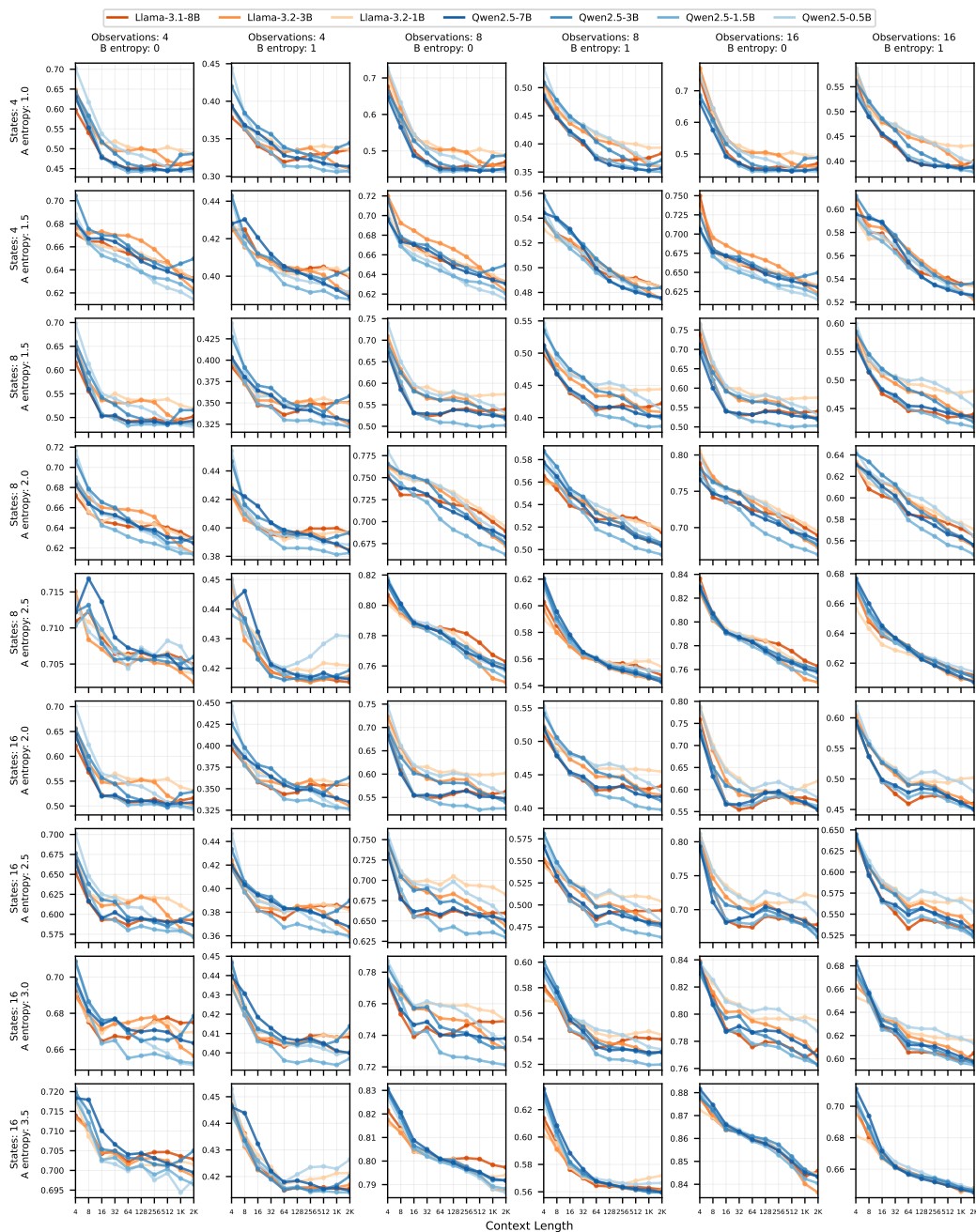

Figure 16: Hellinger distances of seven models across different **A** entropy, **B** entropy, number of states, and number of emissions with $\lambda_2 = 0.75$ and uniform steady state distribution. The models converge similarly, especially when entropy is high.

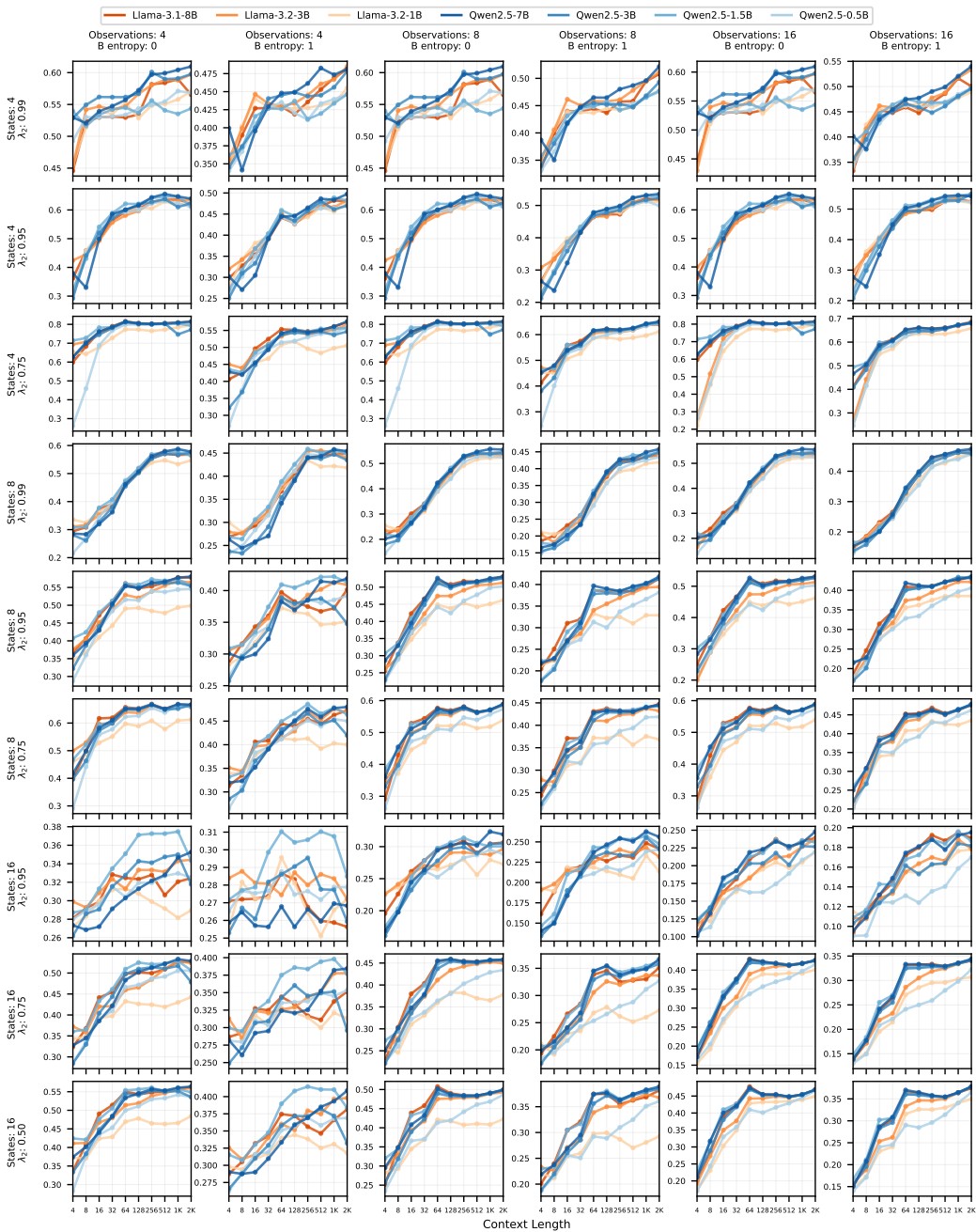

Figure 17: Accuracies of seven models across different mixing rates ($\lambda_2$), **B** entropy, number of states, and number of emissions with uniform steady state distribution and (1, 2, 3) **A** entropy for (4, 8, 16) states respectively. The two smallest models from each family have suboptimal performance, especially when mixing is fast.

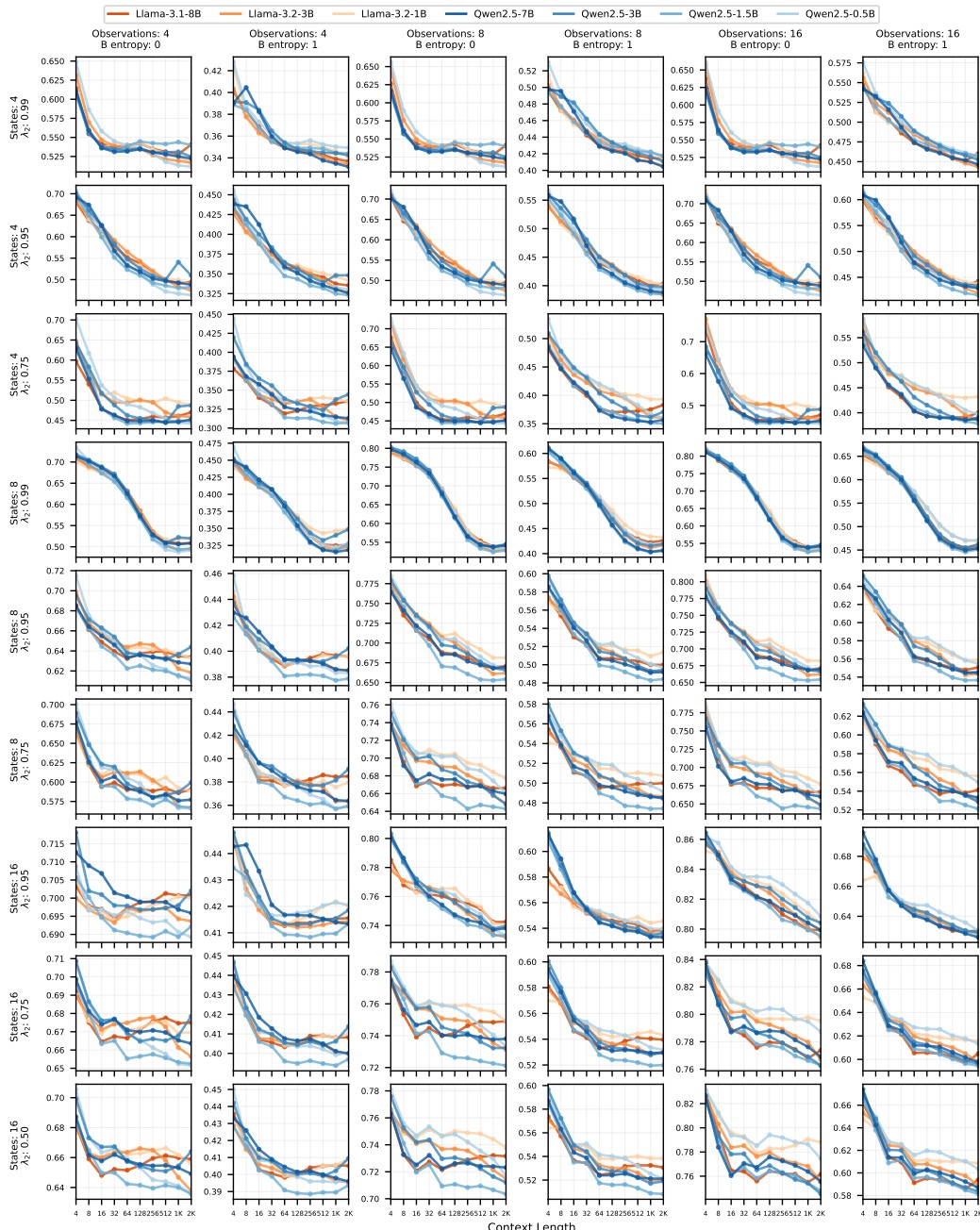

Figure 18: Hellinger distances of seven models across different mixing rates ($\lambda_2$), **B** entropy, number of states, and number of emissions with uniform steady state distribution and $(1, 2, 3)$ **A** entropy for $(4, 8, 16)$ states respectively. The models converge similarly, especially when mixing is slow.

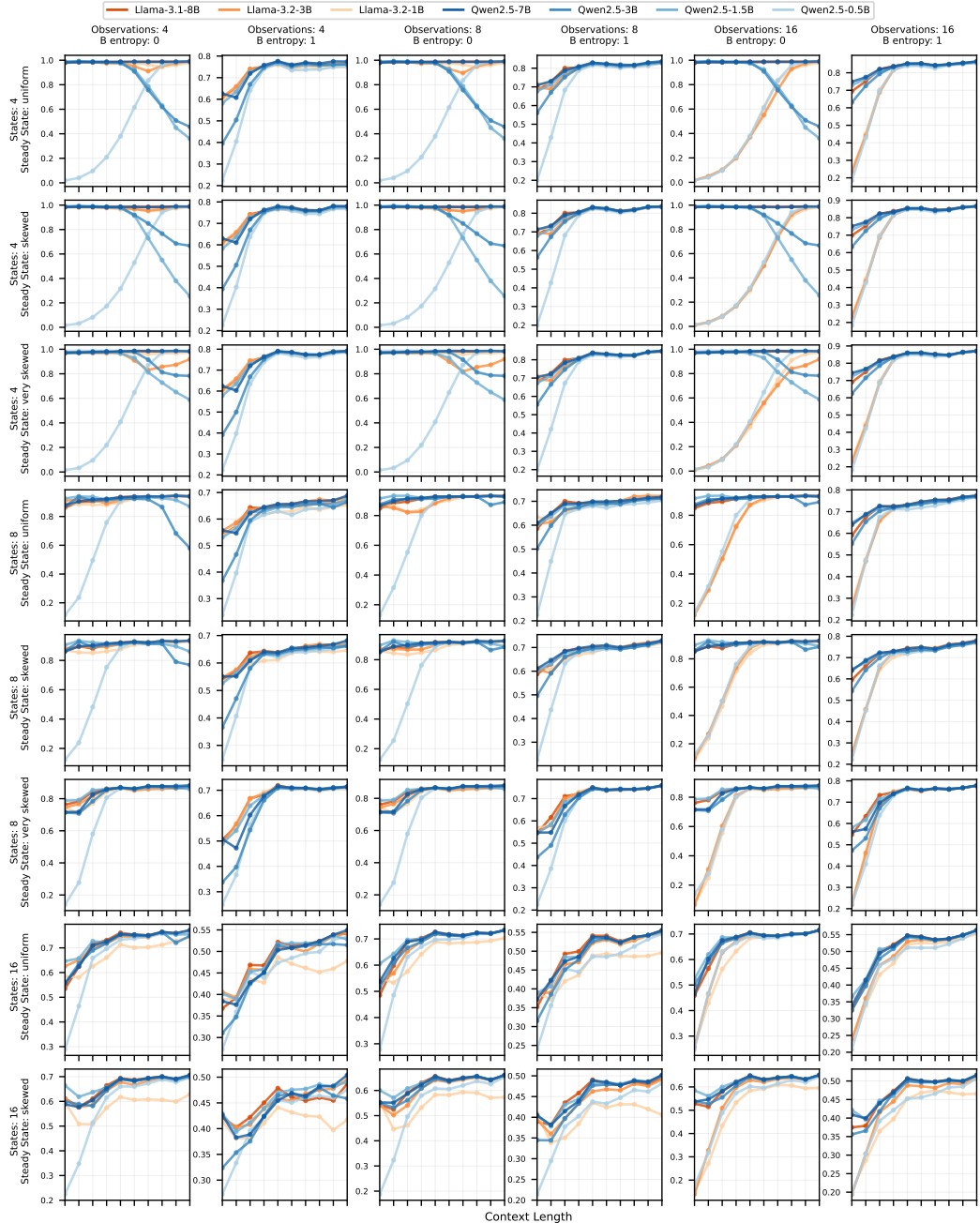

Figure 19: Accuracies of seven models across different steady state distributions, **B** entropy, number of states, and number of emissions with (0, 0.5, 2) **A** entropy for (4, 8, 16) states respectively and (0.99, 0.95, 0.75) $\lambda_2$ for (4, 8, 16) states respectively. The poor performance observed in smaller models at short context length under low **A&B** entropy settings may be attributed to the filtering of repeated n-grams during pretraining, as discussed in Appendix E.2.

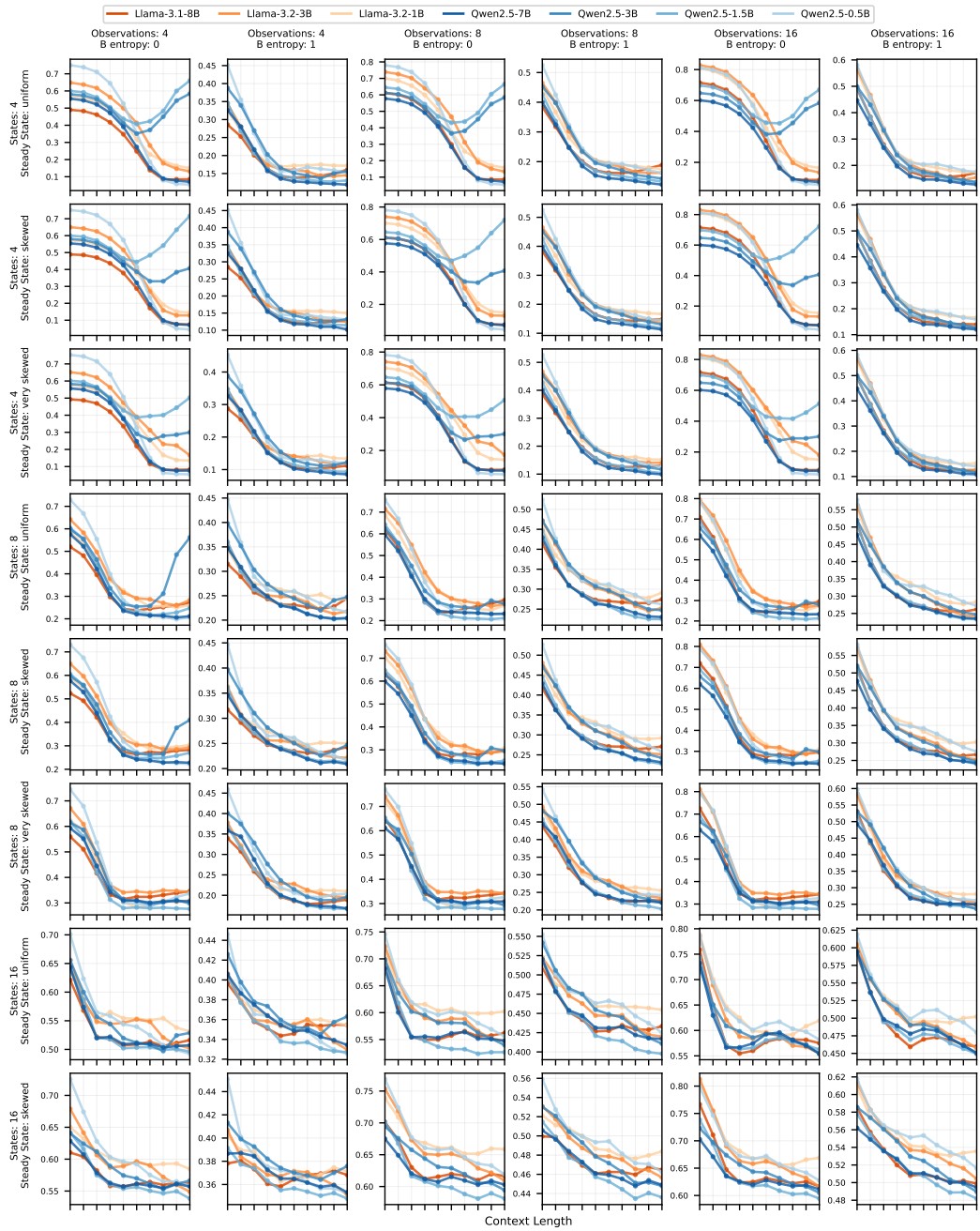

Figure 20: Hellinger distances of seven models across different steady state distributions, **B** entropy, number of states, and number of emissions with (0, 0.5, 2) **A** entropy for (4, 8, 16) states respectively and (0.99, 0.95, 0.75) $\lambda_2$ for (4, 8, 16) states respectively.

## E.2 Tokenization

In this section, we evaluate three tokenization strategies: **ABC**, which encodes emissions as single letters; **123**, which encodes them as single digits; and **random**, which maps emissions to random tokens from the LLM's tokenizer. For the **random** strategy, we specifically map emissions to special tokens (!@#$). All experiments are conducted using the Qwen2.5-1.5B model, and the results are presented below.

We observe that all tokenization methods converge to similar performance levels in terms of accuracy, with **ABC** converging slightly faster when the entropy of $A$ is large. This suggests that the choice of tokenization has limited impact on final performance. In our experiments, we adopt the **ABC** tokenization for maximum performance on the LLM. However, when the entropy of matrix $A$ is low, **ABC** tokenization exhibits significantly lower initial accuracy and a higher Hellinger distance with short context length. We hypothesize that this is due to the increased likelihood of repetitive state sequences early in the sequence—for example, 'AAAAA...'. During pretraining, such repeated n-gram patterns are often filtered out, as they could cause loss spikes [38]. As a result, the model may have limited exposure to these patterns, leading to poor initial performance on such inputs.

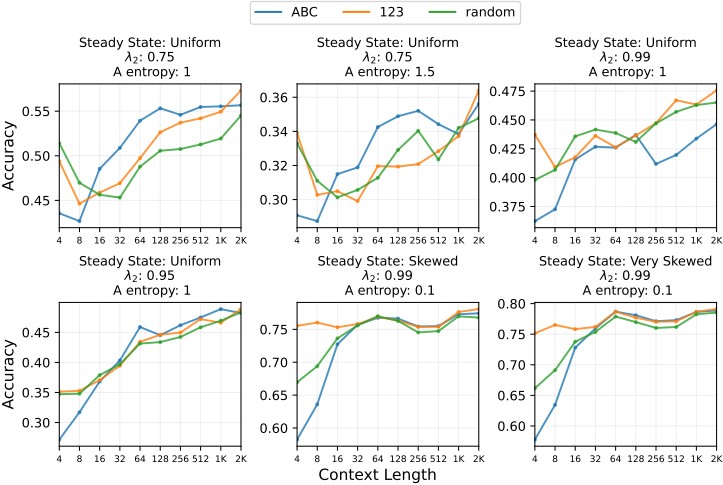

Figure 21: Accuracy of three tokenization methods across different mixing rates ($\lambda_2$), **A** entropy, and steady states with 4 states, 4 emissions, and 1 for **B** entropy.

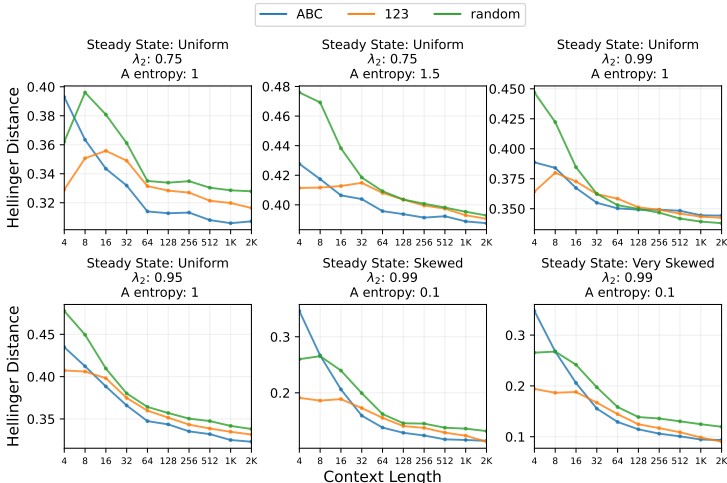

Figure 22: Hellinger distance of three tokenization methods across different mixing rates ($\lambda_2$), **A** entropy, and steady states with 4 states, 4 emissions, and 1 for **B** entropy.

# F  Spectral Learning HMMs for Prediction Task

**Notations:** We use $[\mathbf{X}]_{i,j}$ to denote the element of matrix $\mathbf{X}$ at its $i$-th row and $j$-th column. The indicator function $\mathbf{1}_{\{x=i\}}$ is 1 only when $x = i$ and is 0 otherwise. We use $\mathbf{1}_M$ to denote a vector of all 1's with dimension $M$. We use the notation $[L] = \{1, 2, \ldots, L\}$. $\|\cdot\|$ denotes the Frobenius norm for matrices, and depending on the context it denotes $\ell_1$ or $\ell_2$ norm for vectors.

---

**Algorithm 6:** Spectral Learning-Based Prediction

---

**Input:** Number of hidden states $M$, number of observations $L$, sequence $\{o_1, \ldots, o_N\}$

**Output:** Conditional probability distribution $\hat{P}(O_{N+1}|O_{1:N} = o_{1:N})$

**Estimate empirical probabilities: for** *all combinations* $i, j, n \in [L]$ **do**

$\quad [\hat{\mathbf{P}}_1]_i \leftarrow \frac{1}{N} \sum_{k=1}^N \mathbf{1}_{\{o_k=i\}}$;

$\quad [\hat{\mathbf{P}}_2]_{i,j} \leftarrow \frac{1}{N} \sum_{k=1}^N \mathbf{1}_{\{o_k=i, o_{k-1}=j\}}$;

$\quad [\hat{\mathbf{P}}_{3,n}]_{i,j} \leftarrow \frac{1}{N} \sum_{k=1}^N \mathbf{1}_{\{o_k=i, o_{k-1}=n, o_{k-2}=j\}}$;

**end**

**Compute SVD for dimensionality reduction:**

$\quad \hat{\mathbf{U}} \leftarrow$ left singular vectors of $\hat{\mathbf{P}}_2$ corresponding to $M$ largest singular values;

**Estimate spectral parameters:** $\hat{\mathbf{b}}_1 \leftarrow \hat{\mathbf{U}}^\top \hat{\mathbf{P}}_1$;

$\hat{\mathbf{b}}_\infty \leftarrow (\hat{\mathbf{P}}_2^\top \hat{\mathbf{U}})^\dagger \hat{\mathbf{P}}_1$;

**for** *each observation* $o \in [L]$ **do**

$\quad \hat{\mathbf{C}}_o \leftarrow \hat{\mathbf{U}}^\top \hat{\mathbf{P}}_{3,o}(\hat{\mathbf{U}}^\top \hat{\mathbf{P}}_2)^\dagger$;

**end**

**Hidden state belief update:** $\hat{\mathbf{b}}_1 \leftarrow$ initial belief;

**for** $\tau = 1$ **to** $N$ **do**

$\quad \hat{\mathbf{b}}_{\tau+1} \leftarrow \frac{\hat{\mathbf{C}}_{o_\tau} \hat{\mathbf{b}}_\tau}{\hat{\mathbf{b}}_\infty^\top \hat{\mathbf{C}}_{o_\tau} \hat{\mathbf{b}}_\tau}$;

**end**

**Conditional probability prediction: for** *each possible next observation* $o_{N+1} \in [L]$ **do**

$\quad \hat{P}(O_{N+1} = o_{N+1}|O_{1:N} = o_{1:N}) \leftarrow \frac{\hat{\mathbf{b}}_\infty^\top \hat{\mathbf{C}}_{o_{N+1}} \hat{\mathbf{b}}_{N+1}}{\sum_{k=1}^L \hat{\mathbf{b}}_\infty^\top \hat{\mathbf{C}}_k \hat{\mathbf{b}}_{N+1}}$;

**end**

**return** $\hat{P}(O_{N+1}|O_{1:N} = o_{1:N})$

---

## F.1  Preliminaries

For a Markov chain with transition matrix $\mathbf{A}$, we let $\boldsymbol{\pi} \in \mathbb{R}_+^M$ denote the initial state distribution. We assume that $\boldsymbol{\pi}$ is also the stationary distribution of the Markov chain. This can be achieved by taking samples after a burn-in time which is proportional to $\frac{1}{1-\lambda_2(\mathbf{A})}$. Note that $\boldsymbol{\pi}_t = (\mathbf{A}^t)^\top \boldsymbol{\pi}$ is essentially a convex combination of rows of matrix $\mathbf{A}^t$, then by triangle inequality, we have $\|\boldsymbol{\pi}_t - \boldsymbol{\pi}_\infty\|_1 \leq \max_{i \in [M]} \|([\mathbf{A}^t]_{i,:})^\top - \boldsymbol{\pi}_\infty\|_1$. Thus, for an ergodic Markov matrix $\mathbf{A}$, we define the following to quantify the convergence of $\|\boldsymbol{\pi}_t - \boldsymbol{\pi}_\infty\|_1$. For an ergodic Markov matrix $\mathbf{A} \in \mathbb{R}_+^{M \times M}$, let $\tau_{\mathrm{MC}} > 1$ and $\rho_{\mathrm{MC}} \in (\lambda_2(\mathbf{A}), 1)$ be two constants [26, Theorem 4.9] such that

$$\max_{i \in [M]} \|([\mathbf{A}^t]_{i,:})^\top - \boldsymbol{\pi}_\infty\|_1 \leq \tau_{\mathrm{MC}} \rho_{\mathrm{MC}}^t. \tag{4}$$

Furthermore, we define the mixing time of $\mathbf{A}$ as

$$t_{\mathrm{MC}}(\epsilon) := \min \left\{ t \in \mathbb{N} : \max_{i \in [M]} \frac{1}{2} \|([\mathbf{A}^t]_{i,:})^\top - \boldsymbol{\pi}_\infty\|_1 \leq \epsilon \right\}. \tag{5}$$

Note that $\tau(\mathbf{M})$ and $\tau_{\mathrm{MC}}$ have similar roles except $\tau(\mathbf{M})$ is usually used to study state matrices while $\tau_{\mathrm{MC}}$ is for Markov matrices. For a square $\mathbf{M}$, we have $\|\mathbf{M}^k\| \leq \tau(\mathbf{M})\rho(\mathbf{M})^k$, and for a Markov matrix, we have $\|\mathbf{A}^t - \mathbf{1}_M \boldsymbol{\pi}_\infty^\top\| \leq \tau_{\mathrm{MC}} \rho_{\mathrm{MC}}^t$.

## F.2 Sample Complexity Analysis

In this section, we analyze the sample complexity of spectral learning algorithm (Alg 6) when the observation sequence is coming from a single trajectory. Our proof builds on [21] by modifying their analysis in Appendix A to incorporate single trajectory learning. We only present the Sample complexity analysis here and refer the reader to [21] for the remaining proofs.

## F.3 Proof of Theorem 1

Fix $2 < T < N$, and recall from [21] that $[\mathbf{P}_1]_i = \mathbb{E}[\mathbf{1}_{\{o_T=i\}}]$, $[\mathbf{P}_2]_{i,j} = \mathbb{E}[\mathbf{1}_{\{o_T=i,o_{T-1}=j\}}]$, $[\mathbf{P}_{3,k}]_{i,j} = \mathbb{E}[\mathbf{1}_{\{o_T=i,o_{T-1}=k,o_{T-2}=j\}}]$, for all $k \in [L]$, when the initial distribution $\boldsymbol{\pi}$ is the stationary distribution of the Markov chain. In the following, we will present three different estimators for each of these quantities and analyze their convergence.

● **Estimation of $\mathbf{P}_1$:** Let $\bar{N} := \lfloor \frac{N}{T} \rfloor$, and without loss of generality, suppose $\frac{N}{T}$ is an integer. Suppose $\{o_T^{(1)}, \ldots, o_T^{(\bar{N})}\}$ be the i.i.d. samples obtained from $\bar{N}$ independent trajectories of the HMM. We define the following three estimators of $\mathbf{P}_1$,

$$[\hat{\mathbf{P}}_1]_i = \frac{\sum_{k=1}^{N} \mathbf{1}_{\{o_k=i\}}}{N} \qquad [\hat{\mathbf{P}}_1^{(\ell)}]_i = \frac{\sum_{k=1}^{\bar{N}} \mathbf{1}_{\{o_{kT-\ell}=i\}}}{\bar{N}} \qquad [\hat{\mathbf{P}}_1^{(\perp)}]_i = \frac{\sum_{k=1}^{\bar{N}} \mathbf{1}_{\{o_T^{(k)}=i\}}}{\bar{N}}, \quad (6)$$

for all $\ell = 0, \ldots, T-1$. By triangle inequality, we have

$$\|\hat{\mathbf{P}}_1 - \mathbf{P}_1\| \leq \|\hat{\mathbf{P}}_1 - \hat{\mathbf{P}}_1^{(\perp)}\| + \|\hat{\mathbf{P}}_1^{(\perp)} - \mathbf{P}_1\|. \quad (7)$$

[21] showed that, with probability at least $1 - \delta$, we have, $\|\hat{\mathbf{P}}_1^{(\perp)} - \mathbf{P}_1\| \lesssim \sqrt{\frac{\log(1/\delta)}{\bar{N}}} + \sqrt{\frac{1}{\bar{N}}}$. In the following, we will upper bound the term $\|\hat{\mathbf{P}}_1 - \hat{\mathbf{P}}_1^{(\perp)}\|$ by considering entry-wise concentration of each $\ell$-th subtrajectory as follows: We have

$$[\hat{\mathbf{P}}_1^{(\ell)}]_i - [\hat{\mathbf{P}}_1^{(\perp)}]_i = \frac{\sum_{k=1}^{\bar{N}} \left( \mathbf{1}_{\{o_{kT-\ell}=i\}} - \mathbf{1}_{\{o_T^{(k)}=i\}} \right)}{\bar{N}}. \quad (8)$$

First, we observe that $\mathbb{E}\left[ \mathbf{1}_{\{o_{kT-\ell}=i\}} - \mathbf{1}_{\{o_T^{(k)}=i\}} \right]$ 0. Moreover, $|\mathbf{1}_{\{o_{kT-\ell}=i\}} - \mathbf{1}_{\{o_T^{(k)}=i\}}| \leq 1$, almost surely. However, the summation in (8) has weakly dependent terms. Therefore, we use the Bernstein type inequality for a class of weakly dependent and bounded random variables proposed in [35]. Before that, we need to upper bound the variance of the summation in (8). Observing that $\mathbb{E}\left[ [\hat{\mathbf{P}}_1^{(\ell)}]_i - [\hat{\mathbf{P}}_1^{(\perp)}]_i \right] = 0$, we have,

$$\mathbf{Var}\left( [\hat{\mathbf{P}}_1^{(\ell)}]_i - [\hat{\mathbf{P}}_1^{(\perp)}]_i \right) := \mathbb{E}\left[ \left( [\hat{\mathbf{P}}_1^{(\ell)}]_i - [\hat{\mathbf{P}}_1^{(\perp)}]_i \right)^2 \right],$$

$$= \mathbb{E}\left[ \left( [\hat{\mathbf{P}}_1^{(\ell)}]_i \right)^2 \right] + \mathbb{E}\left[ \left( [\hat{\mathbf{P}}_1^{(\perp)}]_i \right)^2 \right] - 2\,\mathbb{E}\left[ [\hat{\mathbf{P}}_1^{(\ell)}]_i [\hat{\mathbf{P}}_1^{(\perp)}]_i \right]. \quad (9)$$

In the following, we will upper bound each term in (9) separately. We begin with,

$$\mathbb{E}\left[ \left( [\hat{\mathbf{P}}_1^{(\ell)}]_i \right)^2 \right] = \frac{1}{\bar{N}^2} \mathbb{E}\left[ \sum_{k=1}^{\bar{N}} \sum_{k'=1}^{\bar{N}} \mathbf{1}_{\{o_{kT-\ell}=i\}} \mathbf{1}_{\{o_{k'T-\ell}=i\}} \right],$$

$$= \frac{1}{\bar{N}^2} \sum_{k=1}^{\bar{N}} \sum_{k'=1}^{\bar{N}} \mathbb{E}\left[ \mathbf{1}_{\{o_{kT-\ell}=i,o_{k'T-\ell}=i\}} \right],$$

$$= \frac{1}{\bar{N}^2} \sum_{k=1}^{\bar{N}} \sum_{k'=1}^{\bar{N}} \mathbb{P}\left( O_{kT-\ell} = i, O_{k'T-\ell} = i \right),$$

$$= \frac{[\mathbf{B}^\top \boldsymbol{\pi}]_i}{\bar{N}} + \frac{1}{\bar{N}^2} \sum_{k=1}^{\bar{N}} \sum_{\substack{k'=1 \\ k' \neq k}}^{\bar{N}} \left[ \mathbf{B}^\top \mathbf{diag}(\boldsymbol{\pi}) \mathbf{A}^{|k-k'|T} \mathbf{B} \right]_{i,i}. \quad (10)$$

Next, we have,

$$\mathbb{E}\left[\left([\hat{\mathbf{P}}_1^{(\perp)}]_i\right)^2\right] = \frac{1}{\bar{N}^2}\mathbb{E}\left[\sum_{k=1}^{\bar{N}}\sum_{k'=1}^{\bar{N}}\mathbf{1}_{\{o_T^{(k)}=i\}}\mathbf{1}_{\{o_T^{(k')}=i\}}\right],$$

$$= \frac{1}{\bar{N}^2}\sum_{k=1}^{\bar{N}}\sum_{k'=1}^{\bar{N}}\mathbb{E}\left[\mathbf{1}_{\{o_T^{(k)}=i,o_T^{(k')}=i\}}\right],$$

$$= \frac{1}{\bar{N}^2}\sum_{k=1}^{\bar{N}}\sum_{k'=1}^{\bar{N}}\mathbb{P}\left(O_T^{(k)}=i,O_T^{(k')}=i\right) = \frac{[\mathbf{B}^\top\boldsymbol{\pi}]_i}{\bar{N}} + (\bar{N}-1)\frac{[\mathbf{B}^\top\boldsymbol{\pi}]_i^2}{\bar{N}}. \quad (11)$$

Lastly, we have

$$\mathbb{E}\left[\left([\hat{\mathbf{P}}_1^{(\ell)}]_i\right)\left([\hat{\mathbf{P}}_1^{(\perp)}]_i\right)\right] = \frac{1}{\bar{N}^2}\mathbb{E}\left[\sum_{k=1}^{\bar{N}}\sum_{k'=1}^{\bar{N}}\mathbf{1}_{\{o_{kT-\ell}=i\}}\mathbf{1}_{\{o_T^{(k')}=i\}}\right],$$

$$= \frac{1}{\bar{N}^2}\sum_{k=1}^{\bar{N}}\sum_{k'=1}^{\bar{N}}\mathbb{E}\left[\mathbf{1}_{\{o_{kT-\ell}=i,o_T^{(k')}=i\}}\right],$$

$$= \frac{1}{\bar{N}^2}\sum_{k=1}^{\bar{N}}\sum_{k'=1}^{\bar{N}}\mathbb{P}\left(O_{kT-\ell}=i,O_T^{(k')}=i\right) = [\mathbf{B}^\top\boldsymbol{\pi}]_i^2. \quad (12)$$

Combining (10), (11), and (12) into (9), we get

$$\mathbf{Var}\left([\hat{\mathbf{P}}_1^{(\ell)}]_i - [\hat{\mathbf{P}}_1^{(\perp)}]_i\right) = \frac{2[\mathbf{B}^\top\boldsymbol{\pi}]_i}{\bar{N}} + \frac{1}{\bar{N}^2}\sum_{k=1}^{\bar{N}}\sum_{\substack{k'=1\\k'\neq k}}^{\bar{N}}\left[\mathbf{B}^\top\mathbf{diag}(\boldsymbol{\pi})\mathbf{A}^{|k-k'|T}\mathbf{B}\right]_{i,i}$$

$$- (\bar{N}+1)\frac{[\mathbf{B}^\top\boldsymbol{\pi}]_i^2}{\bar{N}},$$

$$= \frac{2([\mathbf{B}^\top\boldsymbol{\pi}]_i - [\mathbf{B}^\top\boldsymbol{\pi}]_i^2)}{\bar{N}} + \frac{1}{\bar{N}^2}\sum_{k=1}^{\bar{N}}\sum_{\substack{k'=1\\k'\neq k}}^{\bar{N}}\left[\mathbf{B}^\top\mathbf{diag}(\boldsymbol{\pi})\mathbf{A}^{|k-k'|T}\mathbf{B}\right]_{i,i}$$

$$- (\bar{N}-1)\frac{[\mathbf{B}^\top\boldsymbol{\pi}]_i^2}{\bar{N}},$$

$$= \frac{2(\mathbf{b}_i^\top\boldsymbol{\pi} - (\mathbf{b}_i^\top\boldsymbol{\pi})^2)}{\bar{N}}$$

$$+ \frac{1}{\bar{N}^2}\sum_{k=1}^{\bar{N}}\sum_{\substack{k'=1\\k'\neq k}}^{\bar{N}}\mathbf{b}_i^\top\mathbf{diag}(\boldsymbol{\pi})\left(\mathbf{A}^{|k-k'|T} - \mathbf{1}_M\boldsymbol{\pi}^\top\right)\mathbf{b}_i,$$

$$\lesssim \frac{\mathbf{b}_i^\top\boldsymbol{\pi} - (\mathbf{b}_i^\top\boldsymbol{\pi})^2}{\bar{N}} + \frac{\|\mathbf{b}_i\|^2\tau_{\mathrm{MC}}\rho_{\mathrm{MC}}^T}{\bar{N}(1-\rho_{\mathrm{MC}}^T)} \lesssim \frac{\mathbf{b}_i^\top\boldsymbol{\pi} - (\mathbf{b}_i^\top\boldsymbol{\pi})^2}{\bar{N}}, \quad (13)$$

where $\mathbf{b}_i$ denotes the $i$-th column of $\mathbf{B}$ and we get the last inequality by choosing,

$$T \gtrsim \log\left(\frac{\|\mathbf{b}_i\|^2\tau_{\mathrm{MC}}}{(\mathbf{b}_i^\top\boldsymbol{\pi} - (\mathbf{b}_i^\top\boldsymbol{\pi})^2)(1-\rho_{\mathrm{MC}}^T)}\right)/(1-\rho). \quad (14)$$

Hence, using the Bernstein type inequality for weakly dependent and bounded random variables (Theorem 1 in [35]), together with (13) (14), and the observations we made right after (8), with probability at least $1 - \delta$, we have

$$|[\hat{\mathbf{P}}_1^{(\ell)}]_i - [\hat{\mathbf{P}}_1^{(\perp)}]_i| \lesssim \sqrt{\frac{(\mathbf{b}_i^\top\boldsymbol{\pi} - (\mathbf{b}_i^\top\boldsymbol{\pi})^2)}{\bar{N}}\log\left(\frac{1}{\delta}\right)}. \quad (15)$$

Union bounding over all $i \in [L]$, and $\ell \in \{0, 1, \ldots, T-1\}$, with probability at least $1 - \delta$, we have

$$\|[\hat{\mathbf{P}}_1^{(\ell)} - \hat{\mathbf{P}}_1^{(\perp)}]\| \lesssim \sqrt{\frac{\mathbf{1}_L^\top \mathbf{B}^\top \boldsymbol{\pi} - \|\mathbf{B}^\top \boldsymbol{\pi}\|^2}{\bar{N}} \log\left(\frac{LT}{\delta}\right)}, \tag{16}$$

given $T \gtrsim \max_{i \in [L]} \left\{ \log\left( \frac{\|\mathbf{b}_i\|^2 \tau_{\mathrm{MC}}}{(\mathbf{b}_i^\top \boldsymbol{\pi} - (\mathbf{b}_i^\top \boldsymbol{\pi})^2)(1 - \rho_{\mathrm{MC}}^T)} \right) \right\} / (1 - \rho)$. This further implies that, with probability at least $1 - \delta$, the same upper bound also holds for $\|[\hat{\mathbf{P}}_1 - \hat{\mathbf{P}}_1^{(\perp)}]\|$. Combining this with (7) and [21], with probability at least $1 - \delta$, we have

$$\|\hat{\mathbf{P}}_1 - \mathbf{P}_1\| \lesssim \sqrt{\frac{\log(1/\delta)}{\bar{N}}} + \sqrt{\frac{1}{\bar{N}}} + \sqrt{\frac{\mathbf{1}_L^\top \mathbf{B}^\top \boldsymbol{\pi} - \|\mathbf{B}^\top \boldsymbol{\pi}\|^2}{\bar{N}} \log\left(\frac{LT}{\delta}\right)}. \tag{17}$$

• **Estimation of $\mathbf{P}_2$:** Here, we follow a similar line of reasoning as above. We begin with defining the three estimators of $\mathbf{P}_2$ as follows,

$$[\hat{\mathbf{P}}_2]_{i,j} = \frac{\sum_{k=1}^{N} \mathbf{1}_{\{o_k = i, o_{k-1} = j\}}}{N} \qquad [\hat{\mathbf{P}}_2^{(\ell)}]_{i,j} = \frac{\sum_{k=1}^{\bar{N}} \mathbf{1}_{\{o_{kT-\ell} = i, o_{kT-\ell-1} = j\}}}{\bar{N}},$$

$$[\hat{\mathbf{P}}_2^{(\perp)}]_{i,j} = \frac{\sum_{k=1}^{\bar{N}} \mathbf{1}_{\{o_T^{(k)} = i, o_{T-1}^{(k)} = j\}}}{\bar{N}} \tag{18}$$

Similar to $\mathbf{P}_1$, we consider the entry-wise concentration of each $\ell$-th subtrajectory as follows,

$$[\hat{\mathbf{P}}_2^{(\ell)}]_{i,j} - [\hat{\mathbf{P}}_2^{(\perp)}]_{i,j} = \frac{\sum_{k=1}^{\bar{N}} \left( \mathbf{1}_{\{o_{kT-\ell} = i, o_{kT-\ell-1} = j\}} - \mathbf{1}_{\{o_T^{(k)} = i, o_{T-1}^{(k)} = j\}} \right)}{\bar{N}}. \tag{19}$$

Observing that $\mathbb{E}\left[ [\hat{\mathbf{P}}_2^{(\ell)}]_{i,j} - [\hat{\mathbf{P}}_2^{(\perp)}]_{i,j} \right] = 0$, we have,

$$\mathbf{Var}\left( [\hat{\mathbf{P}}_2^{(\ell)}]_{i,j} - [\hat{\mathbf{P}}_2^{(\perp)}]_{i,j} \right) = \mathbb{E}\left[ \left( [\hat{\mathbf{P}}_2^{(\ell)}]_{i,j} - [\hat{\mathbf{P}}_2^{(\perp)}]_{i,j} \right)^2 \right],$$

$$= \mathbb{E}\left[ \left( [\hat{\mathbf{P}}_2^{(\ell)}]_{i,j} \right)^2 \right] + \mathbb{E}\left[ \left( [\hat{\mathbf{P}}_2^{(\perp)}]_{i,j} \right)^2 \right] - 2\,\mathbb{E}\left[ [\hat{\mathbf{P}}_2^{(\ell)}]_{i,j} [\hat{\mathbf{P}}_2^{(\perp)}]_{i,j} \right]. \tag{20}$$

In the following, we will upper bound each term in (20) separately. We begin with,

$$\mathbb{E}\left[ \left( [\hat{\mathbf{P}}_2^{(\ell)}]_{i,j} \right)^2 \right] = \frac{1}{\bar{N}^2} \mathbb{E}\left[ \sum_{k=1}^{\bar{N}} \sum_{k'=1}^{\bar{N}} \mathbf{1}_{\{o_{kT-\ell} = i, o_{kT-\ell-1} = j\}} \mathbf{1}_{\{o_{k'T-\ell} = i, o_{k'T-\ell-1} = j\}} \right],$$

$$= \frac{1}{\bar{N}^2} \sum_{k=1}^{\bar{N}} \sum_{k'=1}^{\bar{N}} \mathbb{E}\left[ \mathbf{1}_{\{o_{kT-\ell} = i, o_{kT-\ell-1} = j, o_{k'T-\ell} = i, o_{k'T-\ell-1} = j\}} \right],$$

$$= \frac{1}{\bar{N}^2} \sum_{k=1}^{\bar{N}} \sum_{k'=1}^{\bar{N}} \mathbb{P}\left( O_{kT-\ell} = i, O_{kT-\ell-1} = j, O_{k'T-\ell} = i, O_{k'T-\ell-1} = j \right),$$

$$= \frac{\boldsymbol{\pi}^\top \mathbf{D}_{j,i} \mathbf{1}_M}{\bar{N}} + \frac{1}{\bar{N}^2} \sum_{k=1}^{\bar{N}} \sum_{\substack{k'=1 \\ k' \neq k}}^{\bar{N}} \boldsymbol{\pi}^\top \mathbf{D}_{j,i} \mathbf{A}^{|k-k'|T-1} \mathbf{D}_{j,i} \mathbf{1}_M, \tag{21}$$

where, given the $i$-th column $\mathbf{b}_i$, and the $j$-th column $\mathbf{b}_j$ of $\mathbf{B}$, we define

$$\mathbf{D}_{j,i} := \mathbf{diag}\left( \mathbf{b}_j \right) \mathbf{A} \, \mathbf{diag}\left( \mathbf{b}_i \right). \tag{22}$$

Next, we have

$$\mathbb{E}\left[\left([\hat{\mathbf{P}}_2^{(\perp)}]_{i,j}\right)^2\right] = \frac{1}{\bar{N}^2}\mathbb{E}\left[\sum_{k=1}^{\bar{N}}\sum_{k'=1}^{\bar{N}}\mathbf{1}_{\{o_T^{(k)}=i,o_{T-1}^{(k)}=j\}}\mathbf{1}_{\{o_T^{(k')}=i,o_{T-1}^{(k')}=j\}}\right],$$

$$= \frac{1}{\bar{N}^2}\sum_{k=1}^{\bar{N}}\sum_{k'=1}^{\bar{N}}\mathbb{E}\left[\mathbf{1}_{\{o_T^{(k)}=i,o_{T-1}^{(k)}=j,o_T^{(k')}=i,o_{T-1}^{(k')}=j\}}\right],$$

$$= \frac{1}{\bar{N}^2}\sum_{k=1}^{\bar{N}}\sum_{k'=1}^{\bar{N}}\mathbb{P}\left(O_T^{(k)}=i,O_{T-1}^{(k)}=j,O_T^{(k')}=i,O_{T-1}^{(k')}=j\right),$$

$$= \frac{\boldsymbol{\pi}^\top\mathbf{D}_{j,i}\mathbf{1}_M}{\bar{N}} + (\bar{N}-1)\frac{\left(\boldsymbol{\pi}^\top\mathbf{D}_{j,i}\mathbf{1}_M\right)^2}{\bar{N}}. \tag{23}$$

Lastly, we have

$$\mathbb{E}\left[\left([\hat{\mathbf{P}}_2^{(\ell)}]_{i,j}\right)\left([\hat{\mathbf{P}}_2^{(\perp)}]_{i,j}\right)\right] = \frac{1}{\bar{N}^2}\mathbb{E}\left[\sum_{k=1}^{\bar{N}}\sum_{k'=1}^{\bar{N}}\mathbf{1}_{\{o_{kT-\ell}=i,o_{kT-\ell-1}=j\}}\mathbf{1}_{\{o_T^{(k')}=i,o_{T-1}^{(k')}=j\}}\right],$$

$$= \frac{1}{\bar{N}^2}\sum_{k=1}^{\bar{N}}\sum_{k'=1}^{\bar{N}}\mathbb{E}\left[\mathbf{1}_{\{o_{kT-\ell}=i,o_{kT-\ell-1}=j,o_T^{(k')}=i,o_{T-1}^{(k')}=j\}}\right],$$

$$= \frac{1}{\bar{N}^2}\sum_{k=1}^{\bar{N}}\sum_{k'=1}^{\bar{N}}\mathbb{P}\left(O_{kT-\ell}=i,O_{kT-\ell-1}=j,O_T^{(k')}=i,O_{T-1}^{(k')}=j\right),$$

$$= \left(\boldsymbol{\pi}^\top\mathbf{D}_{j,i}\mathbf{1}_M\right)^2. \tag{24}$$

Combining (21), (23), and (24) into (20), we get

$$\mathbf{Var}\left([\hat{\mathbf{P}}_2^{(\ell)}]_{i,j}-[\hat{\mathbf{P}}_2^{(\perp)}]_{i,j}\right) = \frac{2\boldsymbol{\pi}^\top\mathbf{D}_{j,i}\mathbf{1}_M}{\bar{N}} + \frac{1}{\bar{N}^2}\sum_{k=1}^{\bar{N}}\sum_{\substack{k'=1\\k'\neq k}}^{\bar{N}}\boldsymbol{\pi}^\top\mathbf{D}_{j,i}\mathbf{A}^{|k-k'|T-1}\mathbf{D}_{j,i}\mathbf{1}_M$$

$$- (\bar{N}+1)\frac{\left(\boldsymbol{\pi}^\top\mathbf{D}_{j,i}\mathbf{1}_M\right)^2}{\bar{N}},$$

$$= \frac{2\left(\boldsymbol{\pi}^\top\mathbf{D}_{j,i}\mathbf{1}_M - (\boldsymbol{\pi}^\top\mathbf{D}_{j,i}\mathbf{1}_M)^2\right)}{\bar{N}}$$

$$+ \frac{1}{\bar{N}^2}\sum_{k=1}^{\bar{N}}\sum_{\substack{k'=1\\k'\neq k}}^{\bar{N}}\boldsymbol{\pi}^\top\mathbf{D}_{j,i}\left(\mathbf{A}^{|k-k'|T-1}-\mathbf{1}_M\boldsymbol{\pi}^\top\right)\mathbf{D}_{j,i}\mathbf{1}_M,$$

$$\lesssim \frac{2\left(\boldsymbol{\pi}^\top\mathbf{D}_{j,i}\mathbf{1}_M - (\boldsymbol{\pi}^\top\mathbf{D}_{j,i}\mathbf{1}_M)^2\right)}{\bar{N}} + \frac{\|\boldsymbol{\pi}^\top\mathbf{D}_{j,i}\|\|\mathbf{D}_{j,i}\mathbf{1}_M\|\tau_{\mathrm{MC}}\rho_{\mathrm{MC}}^{T-1}}{\bar{N}(1-\rho_{\mathrm{MC}}^T)},$$

$$\lesssim \frac{\boldsymbol{\pi}^\top\mathbf{D}_{j,i}\mathbf{1}_M - (\boldsymbol{\pi}^\top\mathbf{D}_{j,i}\mathbf{1}_M)^2}{\bar{N}}, \tag{25}$$

where we get the last inequality by choosing,

$$T \gtrsim 1 + \log\left(\frac{\|\boldsymbol{\pi}^\top\mathbf{D}_{j,i}\|\|\mathbf{D}_{j,i}\mathbf{1}_M\|\tau_{\mathrm{MC}}}{\left(\boldsymbol{\pi}^\top\mathbf{D}_{j,i}\mathbf{1}_M - (\boldsymbol{\pi}^\top\mathbf{D}_{j,i}\mathbf{1}_M)^2\right)\left(1-\rho_{\mathrm{MC}}^T\right)}\right)\Big/(1-\rho). \tag{26}$$

Hence, using similar line of reasoning as we did in the case of $P_1$, with probability at least $1-\delta$, we have

$$\|[\hat{\mathbf{P}}_2^{(\ell)}-\hat{\mathbf{P}}_2^{(\perp)}\| \lesssim \sqrt{\frac{\sum_{i,j=1}^L\left(\boldsymbol{\pi}^\top\mathbf{D}_{j,i}\mathbf{1}_M - (\boldsymbol{\pi}^\top\mathbf{D}_{j,i}\mathbf{1}_M)^2\right)}{\bar{N}}\log\left(\frac{L^2T}{\delta}\right)}, \tag{27}$$

given $T \gtrsim 1 + \max_{i,j \in [L]} \left\{ \log \left( \frac{\|\boldsymbol{\pi}^\top \mathbf{D}_{j,i}\| \|\mathbf{D}_{j,i} \mathbf{1}_M\| \tau_{\mathrm{MC}}}{(\boldsymbol{\pi}^\top \mathbf{D}_{j,i} \mathbf{1}_M - (\boldsymbol{\pi}^\top \mathbf{D}_{j,i} \mathbf{1}_M)^2)(1 - \rho_{\mathrm{MC}}^T)} \right) \right\} / (1 - \rho)$. This further implies that, with probability at least $1 - \delta$, the same upper bound also holds for $\|[\hat{\mathbf{P}}_2 - \hat{\mathbf{P}}_2^{(\perp)}]\|$. Combining this with the triangle inequality and [21], with probability at least $1 - \delta$, we have

$$\|\hat{\mathbf{P}}_2 - \mathbf{P}_2\| \lesssim \sqrt{\frac{\log(1/\delta)}{\bar{N}}} + \sqrt{\frac{1}{\bar{N}}} + \sqrt{\frac{\sum_{i,j=1}^L \left(\boldsymbol{\pi}^\top \mathbf{D}_{j,i} \mathbf{1}_M - (\boldsymbol{\pi}^\top \mathbf{D}_{j,i} \mathbf{1}_M)^2\right)}{\bar{N}}} \log\left(\frac{L^2 T}{\delta}\right).$$
(28)

• **Estimation of $\mathbf{P}_3$:** Here, we follow a similar line of reasoning as above. We begin with defining the three estimators of $\mathbf{P}_3$ as follows,

$$[\hat{\mathbf{P}}_{3,n}]_{i,j} = \frac{\sum_{k=1}^N \mathbf{1}_{\{o_k=i, o_{k-1}=n, o_{k-2}=j\}}}{N} \qquad [\hat{\mathbf{P}}_{3,n}^{(\ell)}]_{i,j} = \frac{\sum_{k=1}^{\bar{N}} \mathbf{1}_{\{o_{kT-\ell}=i, o_{kT-\ell-1}=n, o_{kT-\ell-2}=j\}}}{\bar{N}},$$

$$[\hat{\mathbf{P}}_{3,n}^{(\perp)}]_{i,j} = \frac{\sum_{k=1}^{\bar{N}} \mathbf{1}_{\{o_T^{(k)}=i, o_{T-1}^{(k)}=n o_{T-2}^{(k)}=j\}}}{\bar{N}}$$
(29)

Following the same line of reasoning as we did in the case of $\mathbf{P}_2$, with probability at least $1 - \delta$, we have

$$\|\hat{\mathbf{P}}_{3,n} - \mathbf{P}_{3,n}\|$$
$$\lesssim \sqrt{\frac{\log(1/\delta)}{\bar{N}}} + \sqrt{\frac{1}{\bar{N}}} + \sqrt{\frac{\sum_{i,j,n=1}^L \left(\boldsymbol{\pi}^\top \mathbf{D}_{j,n,i} \mathbf{1}_M - (\boldsymbol{\pi}^\top \mathbf{D}_{j,n,i} \mathbf{1}_M)^2\right)}{\bar{N}}} \log\left(\frac{L^3 T}{\delta}\right),$$
(30)

provided that,

$$T \gtrsim 2 + \max_{i,j,n \in [L]} \left\{ \log \left( \frac{\|\boldsymbol{\pi}^\top \mathbf{D}_{j,n,i}\| \|\mathbf{D}_{j,n,i} \mathbf{1}_M\| \tau_{\mathrm{MC}}}{(\boldsymbol{\pi}^\top \mathbf{D}_{j,n,i} \mathbf{1}_M - (\boldsymbol{\pi}^\top \mathbf{D}_{j,n,i} \mathbf{1}_M)^2)(1 - \rho_{\mathrm{MC}}^T)} \right) \right\} / (1 - \rho),$$
(31)

where, given the $i$-th column $\mathbf{b}_i$, the $j$-th column $\mathbf{b}_j$ and the $n$-th column $\mathbf{b}_n$ of $\mathbf{B}$, we define

$$\mathbf{D}_{j,n,i} := \mathbf{diag}\left(\mathbf{b}_j\right) \mathbf{A} \, \mathbf{diag}\left(\mathbf{b}_n\right) \mathbf{A} \, \mathbf{diag}\left(\mathbf{b}_i\right)$$
(32)

• **Finalizing the proof:** Theorem 1 follows by repeating the proof of Theorem 7 in [21], with the i.i.d. estimators replaced by the single trajectory estimators, and the values of $\epsilon_1, \epsilon_{2,1}$ and $\epsilon_{3,x,1}$ replaced by,

$$\epsilon_1 \lesssim \sqrt{\frac{\log(1/\delta)}{\bar{N}}} + \sqrt{\frac{1}{\bar{N}}} + \sqrt{\frac{\mathbf{1}_L^\top \mathbf{B}^\top \boldsymbol{\pi} - \|\mathbf{B}^\top \boldsymbol{\pi}\|^2}{\bar{N}}} \log\left(\frac{LT}{\delta}\right),$$

$$\epsilon_{2,1} \lesssim \sqrt{\frac{\log(1/\delta)}{\bar{N}}} + \sqrt{\frac{1}{\bar{N}}} + \sqrt{\frac{\sum_{i,j=1}^L \left(\boldsymbol{\pi}^\top \mathbf{D}_{j,i} \mathbf{1}_M - (\boldsymbol{\pi}^\top \mathbf{D}_{j,i} \mathbf{1}_M)^2\right)}{\bar{N}}} \log\left(\frac{L^2 T}{\delta}\right),$$

$$\epsilon_{3,x,1} \lesssim \sqrt{\frac{\log(1/\delta)}{\bar{N}}} + \sqrt{\frac{1}{\bar{N}}} + \sqrt{\frac{\sum_{i,j,n=1}^L \left(\boldsymbol{\pi}^\top \mathbf{D}_{j,n,i} \mathbf{1}_M - (\boldsymbol{\pi}^\top \mathbf{D}_{j,n,i} \mathbf{1}_M)^2\right)}{\bar{N}}} \log\left(\frac{L^3 T}{\delta}\right),$$

where $\bar{N} = \lfloor \frac{N}{T} \rfloor = \mathcal{O}\left(N(1 - \lambda_2(\mathbf{A}))\right)$. The proof is completed by upper bounding the Hellinger-distance in terms of KL-distance.

# G    Additional Real World Experiments

We design an additional experiment using real-world datasets to validate our findings. We artificially simulate different emission entropy levels for the same underlying hidden transition process by controlling the amount of information included in the observation sequence. Using complete information corresponds to low emission entropy, while limiting information artificially increases emission entropy.

We use the IBL decision-making mice dataset [25]. In our LLM in-context learning experiment, we implement four ablation conditions that vary the information presented in each trial: (i) "choice only"; (ii) "choice reward"; (iii) "stimulus choice"; (iv) "stimulus choice reward". Note that the baseline GLM-HMM uses all available information as in condition (iv). These ablations describe the same underlying mouse decision-making sequences but with varying levels of environmental state detail.

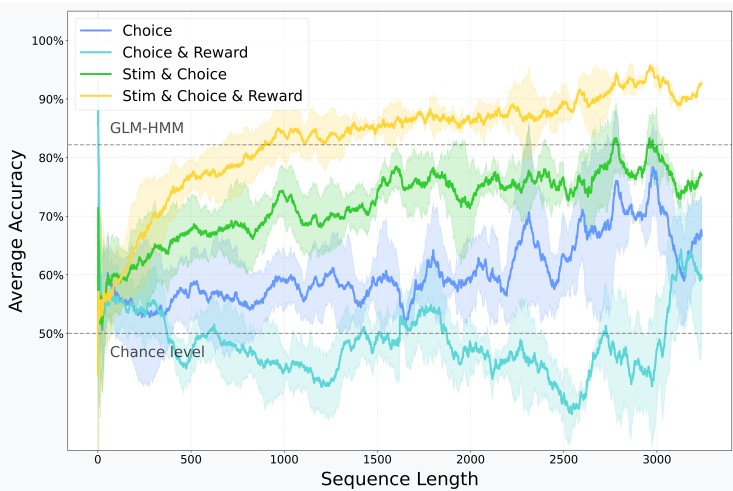

Figure 23: LLM in-context learning prediction accuracy for mice decision-making task with varying types of information in the observed sequences. Each line is averaged over 7 mice, with 1-$\sigma$ error bar. The model we use is Qwen2.5-7B.

The results shown in Figure 23 reveal significant differences across ablation conditions: while "stimulus choice reward" achieves performance exceeding GLM-HMM, "choice reward" is merely at chance level with its convergence trend similar to the synthetic experiments when the transitions or emissions are near random. This demonstrates that accurately modeling mouse decision-making in this task requires both stimulus and reward information.

These findings highlight a broader principle: obtaining appropriate information (corresponding to low emission entropy) is essential for successful task modeling. This experiment parallels real-world experimental design, where scientists must choose which signals to collect when studying task structure. When researchers omit critical information needed to describe a sequence, it can easily lead to incorrect conclusions about the underlying process.

