# OpenReview forum: "Pre-trained Large Language Models Learn to Predict Hidden Markov Models In-context"
_NeurIPS.cc/2025/Conference — NeurIPS 2025 poster_

### Official Review · Reviewer_EN5o · 2025-06-30

**Clarity:** 3
**Significance:** 1
**Originality:** 1
**Rating:** 2
**Confidence:** 3

**Summary:**

The paper investigates the ability of a black-box Large Language Model to learn Hidden Markov Models in-context. This investigation is conducted primarily through empirical evidence derived from simulations, and through the lens of a prediction task, rather than by directly addressing probabilities. The paper showcases that LLMs can be used as efficient and capable predictors of the next token for data coming from a Markov process. The gap the paper aims to close concerns the "hidden" setup, where the states are only observed through emissions. Moreover, the paper somewhat evaluates (see comments below) interesting real-world data, which is considered to come from a Markovian process.

**Questions:**

Addressed in Weaknesses

**Ethical Concerns:**

["NO or VERY MINOR ethics concerns only"]

**Final Justification:**

In general I believe non of my points was resolved, and been honest, they seem to mostly been avoided.
My justication lay in those comments :

This paper receives a low score due to two fundamental flaws that significantly limit its contribution:

While the authors claim the only gap to be fill is the bridge between fully observed and hidden Markov models, they didn't addresed what makes states truly "hidden." The paper provides little focus on emission probabilities and ablation studies on this critical component. This is particularly problematic since emission variance directly determines the degree of "hiddenness" this crucial distinction is ignored despite extensive ablation on transition matrices.

The core novelty claim is undermined by a fundamental confusion between HMM learning and next-token prediction. True HMM learning involves parameter estimation, or at least some validated claim about the model implicitly decode the transition probabilities. Or, even some way to inference those via calibration or any other method. But, the paper only demonstrates predictive performance on the most likely token, which is not novel by itself, specifically given we don't know the emissions and ICL prediction capabilities on Markovien data are proven already. The authors acknowledge that distinguishing next-token prediction on HMM emissions from Markov chains is non-trivial. But, I do believe that given the limited scope to begin with, narrowing it further to the hidden aspects, and then the lack of ablations (comment 1), leaves the paper with insufficient novelty.

Also, there are many other, what i believe, are an insufficient or "out of the point" analyses and evaluations, all described in my comment.

**Limitations:**

Addressed in Weaknesses

**Paper Formatting Concerns:**

Not that I can see.

**Quality:**

2

**Strengths And Weaknesses:**

# Strengths

The paper is very well-written and easy to follow, contributing to its accessibility. Its design is equally well-done, featuring clear and illustrative figures that greatly aid comprehension. Furthermore, the evaluation plots are nicely presented, enhancing the clarity of the results. The authors have thoughtfully addressed most intriguing aspects of Markov models (not as much of the "hidden" aspects, see Weaknesses), and their methodology for constructing the benchmark simulations is robust and well-conceived. Finally, the inclusion of the real-world data is particularly interesting and adds a valuable dimension to the empirical investigation.

# Weaknesses

Summery of the main weaknesses :
1) The paper focuses on an LLM's ability in a *prediction* task for data sampled from an HMM. It's unclear how this relates to the LLM's ability to genuinely "learn" the underlying Markov model. In this case, the novelty is questionable, given its already known that LLM can do ICL for much more complex temporal data.
2) The paper aims to address the "hidden" aspect of HMMs, but this is largely unaddressed in both the text and the simulations.
3) Following (1) and (2), the scope is very limited.

About point (1) :

The paper suggests LLMs can be efficient next-token predictors for data sampled from a Markov process, but this doesn't equate to learning a Markov model. Predicting the next state (maximum likelihood) is distinct from inferring probabilities (reconstruction), which is a primary function of Markov models. Specifically here, The "overconfidence" of LLMs, which might make them better next-token predictors, could paradoxically make them less convincing at reconstructing true Markov models - where the distribution of the next token should come from the transition probabilities. This appears to be the case in the evaluations where LLMs perform best on low-entropy transition matrices (where the most likely token has a high probability).

Moreover on the same point, given the task is predicting next token, the LLM's memory bandwidth can allow it to use past observations (not the current) with lower sample complexity with comparison to traditional reconstruction algorithms which assume the data is Markovian. This also appears to be the case in the evaluations where LLMs easily learn models with fast mixing, indicating ample "signal from the past" in finite sequences.

About point (2) :

The paper claim is to close the gap for "hidden" Markov models, but nonetheless, the paper barely address the hidden aspect of the HMM. There's a lack of analysis on how emission probabilities or the loss from misclassifying observations to hidden states affect predictor results. Also it's unclear whether sequential information is truly critical, or if the LLM is simply better at picking states (which is often more important in practice).

Also, the benchmarked graphs ignored the more interesting Markov process like a jumping processes or a cyclic processes which have unique properties.

As a summery, the claim that LLM "learns" a HMM in ICL is not something the paper shows, but rather that an LLM can be a good next token predictor in ICL for a data sampled from an "easy" distribution like a simple Markov Model (no jumping process etc..).

Smaller points :
* The assumption that the real-world data example originates from a Markov process is not justified. It would be more robust to learn a Markov model from the real data and then sample from that as a benchmark.
* The main paper doesn't contains enough details about the experimental setup, specifically regarding the emissions and their prior distributions.
* Theorem 1 appears to be quite obvious for any learner, and it's unclear what makes an LLM particularly sensitive to these parameters.
* In context of Markov models, an insightful experimental design would involve giving the model a sequence to learn, and then providing a single new sample to predict the next, rather than always providing the full context.

---

> ### Author Rebuttal · Authors · 2025-07-31
>
> We thank the reviewer for their feedback—we want to clarify that several comments appear to stem from **misunderstandings** of the problem setup and our experiments, which we clarify below.
>
> > *Sum1*. This investigation... through the lens of a prediction task, rather than by directly addressing probabilities.
>
> Our study does address probabilities. We evaluate Hellinger distance between the LLM‑predicted next‑emission distribution and the ground‑truth HMM (Main Fig. 5 right; App. D Fig. 3).
>
> > *Sum2*. The paper showcases that LLMs can be used as efficient and capable predictors of the next token for data coming from a Markov process.
>
> Our experiments are on HMM emissions, not simple Markov chains (Sec. 2.1; App. A). The distinction between next‑token prediction on sequences from a Markov chain versus on HMM emissions is non‑trivial. There is no existing theory that provides finite‑sample complexity or error bounds for next‑token prediction given a single finite trajectory of emissions from an HMM.
>
> > *Sum3*. Moreover…interesting real-world data, which is considered to come from a Markovian process.
>
> Our real‑world animal data are commonly modeled with HMMs [1,2], but are not assumed to be Markovian.
>
> > **W1**: It's unclear... complex temporal data.
>
> Unlike Baum‑Welch (BW), which uses non‑convex EM to infer HMM parameters, LLM ICL achieves near‑optimal prediction (Sec. 2.3). This suggests LLMs act like spectral methods—directly mapping sequences to next‑emission distributions. Empirically, LLMs behave similarly to spectral learning (reviewer oWyV W1 & W2). We provide theoretical guarantees for spectral learning (Sec. 3.3), explaining LLM performance trends (Sec. 3.1–3.2).
>
> Evaluating LLMs on theoretically grounded tasks like HMMs is non‑trivial and novel. Even in the simpler case of **fully observed** Markov models, the performance evaluation of LLM’s or transformers has received significant attention from works published at top conferences [3-5]. Our results show LLMs outperform non‑convex BW and align with direct methods like spectral learning.
>
> Finally, although LLMs are known to handle complex temporal sequences, the community remains interested in when and how ICL succeeds. This is why benchmarking LLMs on tasks with clear theoretical baselines—even relatively simple tasks like linear regression [6,7]—is considered valuable. Our work follows this philosophy.
>
> > **W1.1**: This appears... low-entropy transition matrices (where the most likely token has a high probability).
>
> It is expected that low‑entropy transition matrices naturally produce high‑probability next emissions. However, our evaluation is **not** based on raw accuracy approaching 1, but rather on the **gap to the theoretical optimum** for each HMM (see Main Paper Fig. 3).
>
> > **W1.2**: LLM's memory bandwidth... ample "signal from the past" in finite sequences.
>
> We acknowledge the reviewer’s observation that LLMs can leverage long‑range context through attention. For a fully observed Markov sequence, this indeed provides an advantage because induction heads could perform $n$-gram‑style prediction.
>
> However, in our setting with **hidden** Markov models, the emissions themselves are not Markovian. Memorizing recent emissions alone is insufficient for optimal prediction. Leveraging longer context is statistically valid and necessary, as classical methods such as Baum‑Welch also utilize the entire observed sequence via the forward‑backward algorithm.
>
> > **W2**: The paper aims to address the “hidden” aspect of HMMs, but this is largely unaddressed in both the text and the simulations.
>
> We would like to clarify that the “**hidden**” aspect of HMMs is central to our study. All of our numerical experiments are conducted on synthetic data generated from true HMMs, and real‑world datasets that can reasonably be modeled as HMMs [1,2].
>
> > **W2.1**: There's a lack of analysis on how emission probabilities or the loss from misclassifying observations to hidden states affect predictor results.
>
> We do not assume LLMs explicitly classify observations into hidden states. In fact, our results suggest the **opposite**—LLMs ICL is qualitatively different from the Baum‑Welch algorithm (Sec. 3.2).
>
> > **W2.2**: Also it's unclear whether sequential information is truly critical, or if the LLM is simply better at picking states (which is often more important in practice).
>
> Our experimental design evaluates performances across different mixing rates (Sec. 3.1). The fact that LLM performance varies with mixing rate (e.g., performing better on fast‑mixing HMMs) demonstrates that it is leveraging sequential structure. If sequential information were irrelevant, the mixing rate would have no impact on results.
>
> > **W2.3**. The benchmarked graphs ignored the more interesting Markov processes like jumping processes or cyclic processes which have unique properties.
>
> We appreciate the reviewer’s observation that hidden Markov jump processes (HMJPs) and cyclic (periodic) chains exhibit dynamics that differ from the ergodic HMMs we focus on.
> - HMJPs introduce an additional holding‑time random variable, which is application‑specific (e.g., queuing systems). Modeling HMJPs requires predictors to capture additional sojourn‑time distribution. Classical HMM baselines like Baum‑Welch and Viterbi do not directly apply to HMJPs without significant modifications.
> - Cyclic chains' dynamics become more structured once the chain enters a cycle and the prediction space narrows. However, such processes are typically non‑stationary, which requires careful treatment for finite‑trajectory evaluation.
>
> Our benchmark focuses on ergodic HMMs, which allows us to study sequential prediction in a theoretically tractable setting [8,9]. Extending our evaluation to HMJPs or cyclic processes is an exciting future direction. However, we expect that the relative performance trends among pre‑trained LLMs, Baum‑Welch, and spectral methods would remain consistent, as all predictors would be equally affected by the increased complexity.
>
> > **W3**: It would be more robust to learn a Markov model from the real data and then sample from that as a benchmark.
>
> We respectfully disagree with the suggestion to first learn a (hidden) Markov model from the real‑world dataset and then benchmark on samples from that learned model.
>
> Our study is specifically motivated by the real‑world scientific setting, where researchers observe a single finite trajectory and must make predictions or analyze patterns without knowing the true latent model. Learning a model from the data and then resampling would (1) create a simulation‑to‑real gap, (2) reduce the problem to a fully synthetic benchmark—already covered in the first part of our experiments, and (3) fail to test the LLM’s ability to handle the messiness and variability of real emission sequences directly.
>
> Instead, we justify in the paper that the chosen dataset can reasonably be modeled as an HMM, and we evaluate LLMs directly using real-world data.
>
> > **W4**: The main paper doesn't contain enough details about the experimental setup, specifically regarding the emissions and their prior distributions.
>
> The experimental details of initial state distributions $\pi$ are in Section 2.2 (uniform or deterministic). We did suggest generating the emission matrix is relatively straight-forward (Sec. 2.2), but we would love to explain how we construct $\textbf B$ here.
>
> The code (`hmm.py`, 311–326) builds a sequence of stochastic transition matrices with increasing entropy. It starts with a deterministic matrix (zero entropy), then, for each target entropy up to the maximum, computes a row distribution with one dominant entry and the rest uniform, repeating it across rows to form a homogeneous matrix.
>
> > **W5**: Theorem 1 appears to be quite obvious for any learner, and it's unclear what makes an LLM particularly sensitive to these parameters.
>
> Deriving sample‑complexity and error bounds for spectral learning on a single trajectory is non‑trivial due to output correlations. Standard mixing arguments are challenging because we observe only one realization rather than i.i.d. samples. We address this by proving that the single‑trajectory estimates of $\hat P_1, \hat P_2, \hat P_{3,n}$ in Algorithm 6 (Appendix F) converge to the corresponding multi‑trajectory (i.i.d.) estimates.
>
> Regarding the LLM’s sensitivity to Theorem 1 parameters, we refer the reviewer to our responses to Q4 (reviewer 1stE) and W1&W2 (reviewer oWyV).
>
> > **W6**: In the context of Markov models... rather than always providing the full context.
>
> For HMMs, evaluating on a single new observation is different from a sequence of observations. It’s necessary to infer the hidden state sequence using the entire observation sequence.
>
> ---
>
> [1] Ashwood Z.C. et al. *Mice alternate between discrete strategies during perceptual decision‑making*. Nat. Neurosci. 25(2), 201‑212 (2022).
>
> [2] McClintock B.T. et al. *Uncovering ecological state dynamics with hidden Markov models*. Ecol. Lett. 23(12), 1878‑1903 (2020).
>
> [3] Edelman E. et al. *The evolution of statistical induction heads: In‑context learning Markov chains*. NeurIPS 37, 64273‑64311 (2024).
>
> [4] Rajaraman N. et al. *Transformers on Markov data: Constant depth suffices*. NeurIPS 37, 137521‑137556 (2024).
>
> [5] Rajaraman N., Jiao J., & Ramchandran K. *An analysis of tokenization: Transformers under Markov data*. NeurIPS 37, 62503‑62556 (2024).
>
> [6] Li Y. et al. *Transformers as algorithms: Generalization and stability in in‑context learning*. ICML (2023).
>
> [7] Garg S. et al. *What can transformers learn in‑context? A case study of simple function classes*. NeurIPS 35, 30583‑30598 (2022).
>
> [8] Gassiat E., Cleynen A., & Robin S. *Inference in finite‑state non‑parametric hidden Markov models*. Stat. Comput. 26, 61‑71 (2016).
>
> [9] Abraham K., Gassiat E., & Naulet Z. *Fundamental limits for learning hidden Markov model parameters*. IEEE Trans. Inf. Theory 69(3), 1777‑1794 (2022).

---

> > ### Comment · Reviewer_EN5o · 2025-08-05
> > **Reviewer response**
> >
> > Thank you for the reviewer response. After reading all the comments, it is not clear to me what the misunderstanding was, but I am willing to be proven wrong.
> >
> > First, before commenting on the response, let me please address the main points in a categorical way for better clarity:
> >
> > ## 1. Emission Probabilities and Hidden States
> > In rows 301-302, the paper claims that "Theoretical works analyzing transformers' ability to model... show that they can efficiently learn fully observed Markov chains" and "The transition from observable Markov sequences to latent-variable models like HMMs remains underexplored."
> >
> > Do the authors agree that the "gap" closed by this work lies in the difference between fully observed and hidden states?
> > In this case, how come the emission probabilities are not specified across the paper and there is no ablation study performed on them?
> >
> > Do the authors agree that, for example, an HMM with states distributed as N(1,1), N(2,1), N(3,1) will be enormously different from one distributed by N(1,10^-6), N(2,10^-6), N(3,10^-6), which is in practice non-hidden?
> >
> > This is more pronounced given the deep ablation done on the transition matrix. While I understand the authors' underlying comment that the paper aims to check whether the ability to learn MM in ICL still exists when the states are at least somewhat hidden, this limits the scope and also does not explain the missing address to the emission distributions.
> >
> > ## 2. Definition of "Learning an HMM"
> >
> > The main point of the review lies in the question of the definition of "learning an HMM."
> > Learning an HMM commonly addresses the challenge in the "inverse problem" (or reconstruction), that is, estimating the underlying parameters of the latent process.
> > Even if we are less strict about the definition and scope, we should at least ask whether the LLM holds some kind of an *implicit* Markov model which we can somewhat validate.
> >
> > But, to my understanding, the paper did not show any of these. The paper indeed showed that LLMs can perform efficiently at next token prediction on data generated by a hidden Markov model, but this is not an indication of the first two. Most current time series learning methods can learn from data generated by HMMs (which is indeed simple), and I am not aware of any work describing this ability as "learning an HMM." While it is not trivial that an LLM can do this in ICL, given the previous works and the many papers that demonstrated much more complex phenomena, this by itself is not novel enough in my opinion.
> >
> > With this said, I do acknowledge the authors' comment about "The distinction between next token prediction on sequences from a Markov chain versus on HMM emissions is non-trivial. There is no existing theory that provides finite sample complexity or error bounds for next token prediction given a single finite trajectory of emissions from an HMM." **A**nd I agree to some extent, but this does not change the point about the novelty.
> >
> > I also acknowledge the authors' comment about "benchmarking LLMs on tasks with clear theoretical baselines even relatively simple tasks like linear regression [6,7] is considered valuable. Our work follows this philosophy." **B**ut I do believe that given the limited scope to begin with, narrowing it further to the hidden aspects, and then the lack of ablations (comment 1), leaves the paper with insufficient novelty.
> >
> > ---
> >
> > ## Responses to Specific Points
> >
> > **Sum1.** This investigation... through the lens of a prediction task, rather than by directly addressing probabilities.
> >
> > *Our study does address probabilities. We evaluate Hellinger distance between the LLM predicted next emission distribution and the ground truth HMM (Main Fig. 5 right; App. D Fig. 3).*
> >
> > **ReviewComment (RC):** Sorry if I wasn't clear, but I believe this answer is not fair and escapes the need to address the point. My question was about the ICL "addressing probabilities," while your reference is about the Viterbi algorithm. I hope this is an honest mistake.
> >
> > *The experimental details of initial state distributions are in Section 2.2 (uniform or deterministic). We did suggest generating the emission matrix is relatively straightforward (Sec. 2.2), but we would love to explain how we construct it here. The code (hmm.py, 311–326) builds a sequence of stochastic transition matrices with increasing entropy. It starts with a deterministic matrix (zero entropy), then, for each target entropy up to the maximum, computes a row distribution with one dominant entry and the rest uniform, repeating it across rows to form a homogeneous matrix.*
> >
> > **RC:** Again, with all due respect, the review specifically addressed the *emissions and their prior distributions*.

---

> > > ### Comment · Reviewer_EN5o · 2025-08-05
> > >
> > > **Sum2.** The paper showcases that LLMs can be used as efficient and capable predictors of the next token for data coming from a Markov process.
> > >
> > > *Our experiments are on HMM emissions, not simple Markov chains (Sec. 2.1; App. A). The distinction between next token prediction on sequences from a Markov chain versus on HMM emissions is non-trivial. There is no existing theory that provides finite sample complexity or error bounds for next token prediction given a single finite trajectory of emissions from an HMM.*
> > >
> > > **RC:** Addressed above.
> > >
> > > **W1:** It's unclear... complex temporal data.
> > >
> > > *Unlike Baum-Welch (BW), which uses non-convex EM to infer HMM parameters, LLM ICL achieves near-optimal prediction (Sec. 2.3). This suggests LLMs act like spectral methods—directly mapping sequences to next emission distributions. Empirically, LLMs behave similarly to spectral learning (reviewer oWyV W1 & W2). We provide theoretical guarantees for spectral learning (Sec. 3.3), explaining LLM performance trends (Sec. 3.1–3.2). Evaluating LLMs on theoretically grounded tasks like HMMs is non-trivial and novel. Even in the simpler case of fully observed Markov models, the performance evaluation of LLMs or transformers has received significant attention from works published at top conferences [3-5]. Our results show LLMs outperform non-convex BW and align with direct methods like spectral learning.*
> > >
> > > **RC:** "LLM ICL achieves near-optimal prediction (Sec. 2.3). This suggests LLMs act like spectral methods directly mapping sequences to next emission distributions." I believe this is not a claim that can be made given the analysis in this work. The fact that two methods work well does not mean they are the same.
> > >
> > > Given this, I believe my original question still remains – I cannot see how this proposal, even if elegant, distinguishes ICL from any other method in terms of what makes it specifically capable of "learning" HMMs.
> > >
> > > *I addressed the last paragraph above.
> > >
> > > **W1.1:** This appears... low-entropy transition matrices (where the most likely token has a high probability).
> > >
> > > **RC:** Thank you, point taken.
> > >
> > > **W1.2:** LLM's memory bandwidth... ample "signal from the past" in finite sequences.
> > >
> > > *We acknowledge the reviewer's observation that LLMs can leverage long-range context through attention. For a fully observed Markov sequence, this indeed provides an advantage because induction heads could perform n-gram style prediction. However, in our setting with hidden Markov models, the emissions themselves are not Markovian. Memorizing recent emissions alone is insufficient for optimal prediction. Leveraging longer context is statistically valid and necessary, as classical methods such as Baum-Welch also utilize the entire observed sequence via the forward-backward algorithm.*
> > >
> > > **RC:** The point is not about the emissions. Looking at the transition alone, the ICL is not limited by the need to optimize Σ₁ᴺ p(xₜ|xₜ₋₁) given that it can decode p(xₜ|xₜ₋₁, xₜ₋₂...).
> > >
> > > **W2:** The paper aims to address the "hidden" aspect of HMMs, but this is largely unaddressed in both the text and the simulations.
> > >
> > > **RC:** Addressed above.
> > >
> > > **W2.1:** There's a lack of analysis on how emission probabilities or the loss from misclassifying observations to hidden states affect predictor results.
> > >
> > > *We do not assume LLMs explicitly classify observations into hidden states. In fact, our results suggest the opposite—LLMs ICL is qualitatively different from the Baum-Welch algorithm (Sec. 3.2).*
> > >
> > > **RC:** I believe the comment comes from a misunderstanding. As addressed above, the emission probabilities that are missing are those that created the data, and which define to the greatest extent the "misclassification" of states to *any model* (even when the prior is known, hence the classification is optimal from a Bayesian perspective).
> > >
> > > **W2.2:** Also it's unclear whether sequential information is truly critical, or if the LLM is simply better at picking states (which is often more important in practice).
> > >
> > > *Our experimental design evaluates performance across different mixing rates (Sec. 3.1). The fact that LLM performance varies with mixing rate (e.g., performing better on fast-mixing HMMs) demonstrates that it is leveraging sequential structure. If sequential information were irrelevant, the mixing rate would have no impact on results.*
> > >
> > > **RC:** Thank you, I agree.

---

> > > > ### Comment · Reviewer_EN5o · 2025-08-05
> > > >
> > > > **W3:** It would be more robust to learn a Markov model from the real data and then sample from that as a benchmark.
> > > >
> > > > *We respectfully disagree with the suggestion to first learn a (hidden) Markov model from the real-world dataset and then benchmark on samples from that learned model. Our study is specifically motivated by the real-world scientific setting, where researchers observe a single finite trajectory and must make predictions or analyze patterns without knowing the true latent model. Learning a model from the data and then resampling would (1) create a simulation-to-real gap, (2) reduce the problem to a fully synthetic benchmark—already covered in the first part of our experiments, and (3) fail to test the LLM's ability to handle the messiness and variability of real emission sequences directly. Instead, we justify in the paper that the chosen dataset can reasonably be modeled as an HMM, and we evaluate LLMs directly using real-world data.*
> > > >
> > > > **RC:** I respectfully disagree as well. If the goal of the paper is to learn the ICL ability as a predictor on HMM data, what is the point of using "almost-Markovian" data for evaluation? Does the fact that ICL is a better predictor than any other model on data which is assumed to be somewhat Markovian have meaning in the context of this paper?
> > > >
> > > > **W4:** The main paper doesn't contain enough details about the experimental setup, specifically regarding the emissions and their prior distributions.
> > > >
> > > > **W5:** Theorem 1 appears to be quite obvious for any learner, and it's unclear what makes an LLM particularly sensitive to these parameters.
> > > >
> > > > *Deriving sample complexity and error bounds for spectral learning on a single trajectory is non-trivial due to output correlations. Standard mixing arguments are challenging because we observe only one realization rather than i.i.d. samples. We address this by proving that the single trajectory estimates in Algorithm 6 (Appendix F) converge to the corresponding multi-trajectory (i.i.d.) estimates.*
> > > >
> > > > **RC:** Addressed above.
> > > >
> > > > *Regarding the LLM's sensitivity to Theorem 1 parameters, we refer the reviewer to our responses to Q4 (reviewer 1stE) and W1&W2 (reviewer oWyV).*
> > > >
> > > > **W6:** In the context of Markov models... rather than always providing the full context.
> > > >
> > > > *For HMMs, evaluating on a single new observation is different from a sequence of observations. It's necessary to infer the hidden state sequence using the entire observation sequence.*
> > > >
> > > > **RC:** As addressed before, I believe the paper is first and foremost missing the ability to assess P(xₜ|xₜ₋₁) before any full sequence, and this is the rationale behind the comment.

---

> > > ### Author Response · Authors · 2025-08-07
> > > **Response to Reviewer EN5o (Part 1)**
> > >
> > > We thank the reviewer for their follow-up responses, and openness to discussion. We will first address the two main points the reviewer raises.
> > >
> > > **1. Emission Probabilities and Hidden States**
> > >
> > > > In rows 301-302… Do the authors agree that the "gap" closed by this work lies in the difference between fully observed and hidden states?
> > >
> > > We appreciate the reviewer for citing the claims in the paper. Indeed, compared to related works on fully observed Markov chains, our paper focuses on a hidden Markov model, which is underexplored.
> > >
> > > > In this case, how come the emission probabilities are not specified across the paper and there is no ablation study performed on them?
> > >
> > > As we answered in the previous response (**W4**), **the emission matrix $\textbf B$ is an HMM parameter that we have done ablation on**. More specifically, we vary the entropy of $\textbf B$, $H(\textbf B,\boldsymbol \mu) = −\sum_{j,l}\mu_jb_{jl} \log b_{jl}$ where $\boldsymbol \mu$ is the stationary distribution of the underlying Markov transitions (main paper line 90-92, 101-103; Appendix A line 38-43). The entropy of $\textbf B$ quantifies the average uncertainty in observations given the underlying states.
> > >
> > > The details of constructing $\textbf B$ is included in the code, in `hmm.py` line 311-326. It starts with a deterministic matrix (zero entropy), then, for each target entropy up to the maximum, computes a row distribution with one dominant entry and the rest uniform, repeating it across rows to form a homogeneous matrix. These matrices are guaranteed to be stochastic and span a range of transition uncertainties, enabling controlled experiments or analyses involving varying degrees of randomness.
> > >
> > > > Do the authors agree... from one distributed by N(1,10^-6), N(2,10^-6), N(3,10^-6), which is in practice non-hidden?
> > >
> > > For HMMs with **continuous observations**, we agree with the reviewer that varying the emission variance (e.g., $\mathcal N(1,1)$ vs. $\mathcal N(1,10^{−6})$) effectively changes the “hiddenness” of the model. A very small variance yields nearly deterministic emissions, making the model practically non‑hidden.
> > >
> > > Our paper, however, **focuses on discrete HMMs** (main paper line 72–73), as LLMs operate on discrete tokenizations (main paper line 335–339). To explore the analogous effect of varying emission uncertainty, we ablate the entropy of $\textbf B$ from zero (deterministic, effectively non‑hidden) to the maximum (fully random). This serves the same purpose as varying emission variance in continuous HMMs but is aligned with the discrete token‑based setting of LLMs.
> > >
> > > > This is more pronounced... not explain the missing address to the emission distributions.
> > >
> > > We did not highlight the emission matrix $\textbf B$ in the main paper because it influences sequence complexity in a simpler way. Specifically, the emission process can be viewed as a memoryless invariant channel attached to the latent Markov source. If the underlying Markov chain is stationary, irreducible, and aperiodic, then the hidden Markov process is also stationary and ergodic [1, pg. 1531]. Consequently, varying $\textbf B$ mainly adjusts the observation entropy (from nearly deterministic to nearly uniform), but does not introduce additional temporal dependencies beyond those already determined by the transition matrix $\textbf A$.
> > >
> > > **2. Definition of "Learning an HMM"**
> > >
> > > > But, to my understanding, the paper did not show any of these… and I am not aware of any work describing this ability as “learning an HMM.”
> > >
> > > We agree that for an LLM to fully “learn” an HMM in the classical statistical sense, it would need to internally recover the hidden structure—such as the transition and emission matrices. Explicit parameter recovery, as in Baum–Welch, additionally requires assumptions on the model structure (e.g., that the hidden‑state transition matrix is $M\times M$; Table 1 in the main paper). We don’t claim that LLMs perform such recovery. However, we **do** claim their in‑context learning behavior is consistent with **implicit statistical learning of latent dynamics**, specifically in the sense of **improper learning**.
> > >
> > > As we note in Section 3.3 (line 188):
> > >
> > > > “…one can predict the next observation directly using these parameters along with hidden‑state belief updates, without explicitly learning the matrices A and B. This is therefore an example of *improper learning*, which has been extensively studied in related areas like linear dynamical systems [2].”
> > >
> > > To achieve the observed performance—approaching the theoretical optimal predictive distribution—a pre‑trained LLM must internally approximate $P(o_{t+1}|o_{1:t})$, whose optimal form is given in Eq. 1 (main paper line 183). From this perspective, pre‑trained LLMs act as an effective input‑output predictors of structured sequences.

---

> ### Author Response · Authors · 2025-08-07
> **Response to Reviewer EN5o (Part 2)**
>
> We will continue to address the specific RCs.
>
> > **RC to Sum1**: Sorry if I wasn't clear, but I believe this answer is not fair and escapes the need to address the point. My question was about the ICL “addressing probabilities,” while your reference is about the Viterbi algorithm. I hope this is an honest mistake.
>
> From the context in **Sum1**, “...through the lens of a prediction task, rather than by directly addressing probabilities,” we initially interpreted the distinction between “prediction” and “probability” as **argmax prediction** versus **predicting the full next‑emission probability distribution**.
>
> If this is not the intended meaning, we apologize for the misunderstanding. Based on your follow‑up, we now believe that “addressing probabilities” refers to recovering the underlying transition and emission matrices—that is, classical parameter recovery.
>
> If our interpretation is now correct, we would like to clarify:
>
> Our synthetic experiments show that LLM ICL approaches the theoretical optimal predictive distribution. To do so, the model must implicitly approximate $P(o_{t+1}|o_{1:t})$, whose optimal form is given in Eq. 1 (main paper line 183). From this perspective, pre‑trained LLMs act as effective input‑output predictors of structured sequences, similar to spectral learning algorithms. This differs from Baum–Welch, which tackles a non‑convex parameter‑recovery problem and may fail to converge to the theoretical optimum.
>
> > **RC to W1.2**: The point is not about the emissions. Looking at the transition alone, the ICL is not limited by the need to optimize Σ₁ᴺ p(xₜ|xₜ₋₁) given that it can decode p(xₜ|xₜ₋₁, xₜ₋₂...).
>
> We would like to point the reviewer to the body of classical methods such as Viterbi and Baum-Welch (Appendix C, Algorithm 1 and 3). Note that they both utilize the entire observed sequence for decoding via the forward-backward algorithm (Appendix C line 82-83 and 98-104).
>
> > **RC to W2.1**: I believe the comment comes from a misunderstanding. As addressed above, the emission probabilities that are missing are those that created the data, and which define to the greatest extent the "misclassification" of states to any model (even when the prior is known, hence the classification is optimal from a Bayesian perspective).
>
> As we explained in main point 1, the emission matrix $\textbf B$ is an HMM parameter that we explicitly model during the synthetic experiments.
>
> > **RC to W3**: I respectfully disagree as well. If the goal of the paper is to learn the ICL ability as a predictor on HMM data, what is the point of using "almost-Markovian" data for evaluation? Does the fact that ICL is a better predictor than any other model on data which is assumed to be somewhat Markovian have meaning in the context of this paper?
>
> First, we would like to clarify that we never suggest the real‑world data is “almost‑Markovian.” Instead, the real‑world datasets we chose can be interpreted as observation sequences from HMMs (main paper, lines 257‑259).
>
> We originally interpreted **W3** as a suggestion to change our real‑world experiments into a two‑step procedure: learn $\textbf A$, $\textbf B$, $\boldsymbol\pi$ from real‑world data, then evaluate LLM ICL on sequences sampled from these learned parameters.
>
> If this is not the intended meaning, we can reinterpret **W3** as: the reviewer would like us to change the synthetic experiments to learn $\textbf A$, $\textbf B$, $\boldsymbol\pi$ from data rather than procedurally generate them. Under this interpretation, our response is:
>
> - **Data scarcity and uniqueness**. In each scientific setting, the latent dynamics are unique, and typically only a small set of finite trajectories is available. Learning HMMs from such limited data would likely produce a **small and fragmented set of parameters**. Moreover, parameter recovery is a non‑convex optimization problem, and Baum‑Welch often fails to converge to the true parameters. As a result, the learned HMM parameters would likely be **inaccurate and unreliable**.
> - **Need for simulation coverage**. Simulation allows us to systematically explore a wide range of HMM properties (e.g., entropy, mixing rate, emission uncertainty) that are interesting to capture the complexity in latent dynamics and emission randomness. This broader coverage is important for evaluating pre‑trained LLM ICL performance.
>
> ---
>
> [1] Y. Ephraim and N. Merhav. *Hidden markov processes*. IEEE Transactions on Information Theory, 48(6):1518–1569, 2002. doi: 10.1109/TIT.2002.1003838.
>
> [2] Max Simchowitz, Karan Singh, and Elad Hazan. *Improper learning for non-stochastic control*. In Conference on Learning Theory, pages 3320–3436. PMLR, 2020.

---

> > ### Comment · Reviewer_EN5o · 2025-08-08
> >
> > Thank you for the detailed response.
> >
> > I stand corrected and apologize for my oversight regarding the emission matrix ablation study. You are right that the entropy of **B** is indeed varied in your experiments. I will update the score accordingly.
> >
> > However, I would encourage the authors to give more prominence to the emission analysis in the main paper, which i believe is still much missing given that this represents the core gap your work aims to address. The emission matrix **B** and its properties deserve more detailed discussion in the main text rather than being relegated primarily to the appendix and code.
> >
> > Moreover, I find the claim that emissions "influence sequence complexity in a simpler way" and that the emission process is merely "a memoryless invariant channel" to be somewhat in contrast with the paper's stated aim. If the primary contribution is bridging the gap between fully observed and hidden Markov models, then the emission mechanism should be central to the analysis, not treated as a secondary consideration.
> >
> > The characterization that varying **B** "mainly adjusts the observation entropy... but does not introduce additional temporal dependencies" seems to downplay the very complexity that distinguishes HMMs from fully observed Markov chains. This temporal dependency through hidden states, mediated by the emission process, is arguably the core challenge your work seeks to address.
> >
> > I would suggest that future revisions place greater emphasis on the emission analysis in the main paper to better align with the stated contribution of exploring hidden versus fully observed models.
> >
> > >
> > I agree that improper learning is indeed what is happening here, this is in fact the claim in the review, and the main concern about the paper positioning.
> >
> > >
> > Almost any TS model will give a probability on the entire event (that it, sequence), this does not change the fundamental fact that if the model operates under the Markovian assumption, the sequence probability factorizes as Σ p(x_t|x_{t-1}), whereas ICL can naturally leverage higher-order dependencies such as p(x_t|x_{t-1}, x_{t-2}, ...) $\forall t$ without theoretical restrictions. After all, this why we address them as Markovien.  It is counterintuitive that an analysis dedicated to Markov models would treat its core assumption as equivalent to unrestricted sequence modeling.

---

> ### Author Response · Authors · 2025-08-08
> **Response to Reviewer EN5o**
>
> We thank the reviewer for updating their score and acknowledging the emission matrix $\mathbf{B}$ ablation in our experiments. We will discuss the properties of $\mathbf{B}$ in more detail and bring additional experimental setup details into the new revision to avoid such confusions in the future.
>
> Regarding the comment on “simpler influence” and the “memoryless invariant channel” framing, our intention was not to downplay the role of $\mathbf{B}$ in shaping the observation process. Rather, this description reflects the standard factorization in HMM theory: under the Markovian assumption on the hidden states, temporal dependencies in the observations are mediated entirely by the latent chain. The complexity introduced by $\mathbf{B}$ lies in how it transforms hidden dynamics into observed dynamics, which can indeed be substantial in practical modeling. We agree that emphasizing this transformation’s role—especially in the context of the scientific modeling gap we address—would better align with the stated contributions.
>
> We also wish to clarify a potential misunderstanding: while the latent state process in an HMM is **first-order Markov**, the observation process is generally **not** first-order Markov unless emissions are deterministic (0 entropy). In general,
>
> $$p(o_t \mid o_{1:t-1}) = \sum_{x_t} p(o_t \mid x_t) p(x_t \mid o_{1:t-1}),$$
>
> where $\mathcal{X}$ is the latent state space and $\mathcal{O}$ is the observation space. This conditional probability depends on the **entire observation history** via the posterior over latent states.
>
> This differs from the reviewer’s stated factorization for a first-order Markov time series model:
>
> $$p(o_{1:T}) = \prod_{t=2}^T p(o_t \mid o_{t-1}),$$
>
> which holds only if the **observations themselves** are Markov (i.e., $o_t \perp o_{1:t-2} \mid o_{t-1}$). In an HMM, the emission mapping $\mathbf{B}$ breaks this property: two observation sequences with the same last symbol $o_{t-1}$ can have different predictive distributions for $o_t$ depending on the earlier history $o_{1:t-2}$, because that history changes the posterior belief over hidden states.
>
> In other words, $\mathbf{B}$ is not just a static “channel” but the mechanism that converts hidden-state dynamics into **history-dependent observation dynamics**, creating long-range dependencies that fully observed Markov models cannot capture.

---

### Official Review · Reviewer_aPua · 2025-07-01

**Clarity:** 3
**Significance:** 3
**Originality:** 3
**Rating:** 5
**Confidence:** 4

**Summary:**

This paper investigates whether pre-trained large language models (LLMs) can implicitly learn and predict the behavior of Hidden Markov Models (HMMs) through in-context learning (ICL), that is, by simply being provided with sequences generated by HMMs without explicit parameter updates or task-specific training. The authors systematically evaluate LLMs on a range of synthetic HMMs with varying properties, as well as on real-world animal behavior datasets.

The authors show that LLMs can converge to predict the next observation in HMM-generated sequences compared to an oracle defined with the Viterbi algorithm applying on the HMMs that were used to generate the sequences, given sufficient context. They also show the LLMs’ performance and convergence trends are closely linked to intrinsic HMM properties such as entropy and mixing rate. And they have also conducted additional experiments with real neuroscience datasets, and provide practical guidance for using LLM in-context learning as a diagnostic tool for exploring sequential data.

In summary, the study demonstrates that general-purpose LLMs possess an interesting capacity for modeling complex, latent-state-driven sequences and opens new avenues for applying ICL in scientific discovery and data analysis.

**Questions:**

- How are the initial HMM parameters (initial state, transition, emission probabilities) initialized for Baum-Welch training?
- What stopping criterion or tolerance is used for Baum-Welch training, and how many iterations does it typically require?
- How are HMM-generated sequences presented to the LLMs? Are there any special formatting or tokenization strategies used to maximize in-context learning?
- What decoding strategy and parameters (e.g., temperature, top-k, greedy) are used for LLM next-token prediction during evaluation?

**Ethical Concerns:**

["NO or VERY MINOR ethics concerns only"]

**Final Justification:**

I acknowledge that I have read the rebuttal and reviews and will raise my final score to "accept". I think this a good, interesting paper

**Limitations:**

yes

**Quality:**

3

**Strengths And Weaknesses:**

Quality
This paper presents a comprehensive and careful empirical study, systematically evaluating large language models (LLMs) on both synthetic HMMs and real-world datasets, and comparing their in-context learning performance against strong statistical and neural baselines.
While the empirical work is thorough and the main results are well-supported, one main downfall in my opinion is that ICL does not generalize HMMs for usuall applications for speech and handwriting recognition where HMMs are trained are generative models with multiple sequences a time.

Clarity
The paper is clearly written and well-organized, with a logical flow from motivation and background to synthetic experiments, scaling law analysis, and real-world application. The main contributions and findings are explicitly stated, and well-designed figures make the experimental results easy to interpret.
However, some important implementation details, such as prompt formatting and LLM configuration, are only available in the appendix, making it difficult to understing the methodology without reading all the material

Significance
The paper brings to the table an interesting discussion on the easiness of modeling sequencial data with LLMs.
On the other hand, LLMs are generally more complex than usual architectures for HMMs, which can make this an utterly more complex approach to solve problems that could be easily solved with some tweking in HMM topology.

Originality
The paper introduces a novel experimental paradigm, systematically probing whether LLMs can perform HMM inference purely through in-context learning.
However, the paper does not introduce new learning algorithms or theoretical frameworks, and the most original contributions are in empirical observation and framing rather than methodological invention.

---

> ### Author Rebuttal · Authors · 2025-07-31
>
> Thank you for your valuable review of our paper. We address each of your points below.
>
> > **Q1**: How are the initial HMM parameters (initial state, transition, emission probabilities) initialized for Baum-Welch training?
>
> We initialize the Baum-Welch (BW) HMM parameters with random seeds using the actual number of states. This already gives BW more information than the LLM, which does not know the number of states.
>
> Each setting is averaged over 4,096 sample sequences to reduce variance in performance (line 106). We do not initialize BW near the ground‑truth parameters, as this would artificially favor BW in low‑sample regimes by effectively giving it prior knowledge of the HMM model.
>
> > **Q2**: What stopping criterion or tolerance is used for Baum-Welch training, and how many iterations does it typically require?
>
> The Baum-Welch algorithm in this implementation uses the change in log-likelihood between iterations as its stopping criterion, halting when the absolute difference falls below a specified tolerance of 1e-6. This ensures the algorithm stops once the parameter updates no longer yield meaningful improvement in fitting the observed sequence. The number of iterations required for convergence typically ranges from 10 to 100, depending on factors such as the number of hidden states, the length and complexity of the observation sequence, and the random initialization of parameters. A maximum iteration cap of 100 is also set to prevent excessive computation in cases where convergence is slow or unattainable.
>
> > **Q3**: How are HMM-generated sequences presented to the LLMs? Are there any special formatting or tokenization strategies used to maximize in-context learning?
>
> All results in the main paper are using upper case letters like ‘ABC’ to represent emissions (Appendix E.2). For each unique emission, we represent it by an upper unique letter and a space. We ensure that each emission would only get tokenized into one token. We performed an ablation on different tokenization strategies for LLMs (Appendix E.2), and found no significant differences on the prediction task performance.
>
> > **Q4**: What decoding strategy and parameters (e.g., temperature, top-k, greedy) are used for LLM next-token prediction during evaluation?
>
> Decoding strategy does not apply in our setting since we directly use the normalized logits for prediction. For example, given a sequence “A B A” with 2 possible emissions (A/B) the LLM would predict 0.3 probability for A, 0.4 probability for B, and 0.3 probability for the rest of the tokens in the vocabulary. We normalize the probability for the possible emissions such that they add up to 1 (3/7 for A and 4/7 for B). For accuracy, we directly take the argmax of the probability and compare it with the ground-truth emission. For Hellinger distance, we compare the probability of all possible emissions (in this case, [3/7, 4/7]) with the ground-truth emission probabilities based on the current hidden state (Appendix B).
>
> > **W1** Quality: ICL does not generalize HMMs for usual applications for speech and handwriting recognition where HMMs are trained as generative models with multiple sequences at a time.
>
> First, we clarify that our goal is not to replace HMMs as a model architecture in classical applications like speech or handwriting recognition. Instead, this work explores pre-trained LLMs ICL as a complementary statistical tool for analyzing discrete sequential data, particularly in scientific settings where sequences are often unlabeled.
>
> Second, we acknowledge an inherent limitation: pre-trained LLMs operate on discrete tokenizations. As discussed in Section 6 (“Existing Gaps”), extending our approach from discrete observations with underlying discrete hidden dynamics to continuous, real-valued, or high-dimensional observations remains an open and promising research direction.
>
> Finally, our contributions are orthogonal to traditional HMM applications—we focus on demonstrating that there exists a lightweight, forward-pass-based tool for exploratory sequence analysis rather than a replacement for established generative modeling pipelines.
>
> > **W2** Clarity: Some important implementation details, such as prompt formatting and LLM configuration, are only available in the appendix, making it difficult to understand the methodology without reading all the material.
>
> Due to the main paper space limits, we had to push the implementation details to Appendices for finishing our story. We will work on a better content organization in the revised version. Specifically, we plan to add more details of the experimental setup (Appendix B), a detailed discussion about when LLMs fail to converge (Appendix D.4), and results of ablations on LLMs sizes and tokenizations (Appendix E) to the main paper contents.
>
> > **W3** Significance: LLMs are generally more complex than usual architectures for HMMs, which can make this an utterly more complex approach to solve problems that could be easily solved with some tweaking in HMM topology.
>
> We appreciate the reviewer’s concern and address it in three parts:
>
> First, beyond using LLMs as a predictive tool on HMM sequences, a key contribution of our paper is demonstrating that LLMs with ICL can achieve near-optimal prediction on HMM sequences. As the community continues to explore the mechanism of ICL, we believe this new empirical observation provides a valuable step toward understanding how LLMs acquire and leverage latent structure without explicit training.
>
> Second, the strength of LLM ICL is not solely due to the transformer architecture but also to large-scale pretraining. In contrast, “tweaking HMM topology” is not straightforward in practice. Our synthetic experiments show that Baum-Welch often fails to match LLM ICL’s robust predictive performance without significant parameter fitting or prior knowledge of the true dynamics.
>
> Third, pre-trained LLMs require no additional training and perform ICL via simple forward passes. In contrast, classical methods require either strong initialization for Baum-Welch or a large dataset to train RNNs effectively. In our experiments, using LLM ICL on average takes 0.14 seconds for a sequence with 4096 emissions on a H100 while it takes 10.15 seconds for a Baum-Welch algorithm to converge a single time on an Intel(R) Xeon(R) Gold 6426Y CPU, and 13.21 seconds for an LSTM model to converge on a H100. This efficiency gap comes from the ability of pre-trained LLMs to perform parallel inference over previous tokens without training, while Baum-Welch and RNNs depend on iterative expectation-maximization or sequence-level training pipelines. From a practitioner’s perspective, this makes LLM ICL a fast, convenient, and low-overhead tool for exploratory sequence analysis.
>
> > **W4** Originality: The paper does not introduce new learning algorithms or theoretical frameworks, and the most original contributions are in empirical observation and framing rather than methodological invention.
>
> Our work is motivated by the needs of scientists analyzing real‑world dynamical systems. By assessing pre‑trained LLMs as statistical tools, we demonstrate that LLMs performing in‑context learning can achieve near‑optimal performance on HMM prediction, a highly challenging task. These findings provide a foundation for studying how LLMs internally approximate latent probabilistic structures, potentially shedding light on the mechanisms behind in‑context learning. Similar contributions have appeared at NeurIPS, including works such as [1, 2].
>
>  ---
>
> [1] Garg, Shivam, et al. "What can transformers learn in-context? a case study of simple function classes." *Advances in neural information processing systems* 35 (2022): 30583-30598.
>
> [2] Gruver, Nate, et al. "Large language models are zero-shot time series forecasters." *Advances in Neural Information Processing Systems* 36 (2023): 19622-19635.

---

> ### Comment · Reviewer_aPua · 2025-08-05
> **Acknowledgment**
>
> I acknowledge that I have read the rebuttal and reviews and will raise my final score to "accept". I think this a good, interesting paper

---

### Official Review · Reviewer_oWyV · 2025-07-03

**Clarity:** 4
**Significance:** 3
**Originality:** 2
**Rating:** 5
**Confidence:** 4

**Summary:**

This paper shows that pre-trained LLMs can learn Hidden Markov Models given in-context. Interestingly, they achieve near-optimal prediction accuracy that converges to nearly the Viterbi algorithm optimum with true parameters.. The authors systematically vary HMM properties (entropy, mixing rate, state dimensions) and show that convergence speed depends critically on these characteristics: faster mixing and lower entropy lead, for faster convergence. Comparing against baselines (Baum-Welch, LSTM, n-gram), LLMs consistently outperform traditional statistical approaches. The authors provide theoretical conjectures linking ICL behavior to spectral learning algorithms. Finally, the authors test on real-world animal decision-making datasets, where ICL achieves competitive performance with expert-designed models on mouse choice behavior and meaningful performance on complex rat reward-learning tasks. The work then establishes practical guidelines positioning LLMs as "zero-shot statistical tools" for scientific discovery.

**Questions:**

Q1: In Fig 4 caption, isn't smaller lambda 2 fast mixing and large lambda 2 slow mixing?
Q2: There must be limits, right? Are there extreme cases where LLMs cannot figure out the hidden state well?
Q3: Have you observed any sudden transitions? i.e. the LLM "grokking" that there exists a hidden state?

---

Q4: This is a pure curiosity driven question and will not affect the score at all negatively if ignored: What is you gave the context then natural language prompted the model (+CoT) to predict the next state?

**Ethical Concerns:**

["NO or VERY MINOR ethics concerns only"]

**Final Justification:**

The authors have answered the questions i raised adequately.

This paper is a thorough and complete investigation on using LLMs to "fit" hidden Markov models.

I maintain my positive score.

**Limitations:**

Yes.

**Paper Formatting Concerns:**

None.

**Quality:**

3

**Strengths And Weaknesses:**

S1: The paper is very clear and well written. All figures, terms and definitions are easy to follow and well motivated.

S2: The paper spans theoretical grounding, synthetic experiments and real data experiments. The findings connect well.

S3: The paper validates their finding using multiple models(appendices) and using multiple metric and multiple HMMs.

S4: The paper goes further and provides guidelines for using LLMs for HMM identification.

---

W1: While the paper demonstrates some connection to spectral learning theory, the predicted learning curves are not explicitly fitted/compared to the empirical in-context learning curves.

W2: The connection of the theoretical section to the rest is relatively weak

---

> ### Author Rebuttal · Authors · 2025-07-31
>
> Thank you for your insightful and helpful feedback. We respond to your individual questions below.
>
> > **Q1**: In Fig 4 caption, isn't smaller lambda 2 fast mixing and large lambda 2 slow mixing?
>
> Yes. Thanks for catching this, we will update in the next version.
>
> > **Q2**: There must be limits, right? Are there extreme cases where LLMs cannot figure out the hidden state well?
>
> Yes. During the synthetic experiments, we did observe cases (e.g., Appendix D, Fig. 2: 8 states, 4 observations, A entropy ≈ 2.5) where LLMs ICL failed to converge to the Viterbi algorithm’s performance. The detailed discussions are in Appendix D.4.
>
> > **Q3**: Have you observed any sudden transitions? i.e. the LLM "grokking" that there exists a hidden state?
>
> This is an insightful question. Indeed, in-context learning is an emergent property, which appears during later stages of pretraining. A related theoretical work [1] studies how ICL emerges during pretraining. However, a key difference between this and our experiments is training versus inference.
>
> From our experiments, we do not observe any sudden transitions. Instead, the performance improvements of LLMs ICL are gradual and steady (as shown in Main paper Figure 3, Appendix D multiple Figures). Such behavior is one key reason that we hypothesize LLMs ICL operate as input–output predictors, effectively performing a form of smart filtering over observation sequences.
>
> > **W1** & **W2**: While the paper demonstrates some connection to spectral learning theory, the predicted learning curves are not explicitly fitted/compared to the empirical in-context learning curves.
>
> Following the reviewer’s suggestion, we added an additional experiment by implementing Algorithm 6 (Appendix F), a variant of spectral learning with theoretical performance guarantees in Theorem 1, and evaluating it on the same synthetic HMM setting used in Figure 5 (8 states, 8 observations). The comparative results are shown in the table below:
>
> | seq_len | 4 | 8 | 16 | 32 | 64 | 128 | 256 | 512 | 1024 | 2048 | 4096 *(new)* |
> |---|---|---|---|---|---|---|---|---|---|---|---|
> | LLM_acc | 0.39624 | 0.4436 | 0.48999 | 0.50952 | 0.55688 | 0.552 | 0.54199 | 0.54565 | 0.56054 | 0.56396 | **0.58154** |
> | Viterbi_acc | 0.55762 | 0.54932 | 0.56348 | 0.54614 | 0.56567 | 0.55859 | 0.544189 | 0.54785 | 0.56006 | 0.56323 | **0.57983** |
> | BW_acc | 0.49609 | 0.42407 | 0.41406 | 0.40576 | 0.43262 | 0.44556 | 0.43823 | 0.44775 | 0.47363 | 0.49097 | **0.50171** |
> | Spectral_acc *(new)* | N/A | 0.20728 | 0.22974 | 0.28393 | 0.34253 | 0.41064 | 0.49072 | 0.51318 | 0.53296 | 0.54248 | **0.56494** |
> | LLM_hellinger | 0.51198 | 0.4673 | 0.44358 | 0.43160 | 0.41619 | 0.41653 | 0.41573 | 0.40733 | 0.40264 | 0.40261 | **0.39427** |
> | Viterbi_hellinger | 0.36416 | 0.36221 | 0.36225 | 0.36751 | 0.35555 | 0.36485 | 0.37032 | 0.36448 | 0.36376 | 0.36312 | **0.35438** |
> | BW_hellinger | 0.54587 | 0.57499 | 0.59736 | 0.58332 | 0.53185 | 0.4899 | 0.46357 | 0.43798 | 0.42707 | 0.41467 | **0.41107** |
> | Spectral_hellinger *(new)* | N/A | 0.51126 | 0.5459 | 0.55643 | 0.51654 | 0.4723 | 0.40133 | 0.37927 | 0.36355 | 0.35434 | **0.35608** |
>
> When the sequence length is short (≤ 64), the hidden‑state belief update b_\hat​ becomes numerically unstable because the sample complexity of spectral learning scales as $\mathcal O (M^2 L)$, which is 512 in our setting. Once the context length reaches ≥ 512, spectral‑learning predictions converge distributionally to the ground truth, and their performance aligns with that of the LLMs, unlike the BW‑based predictions. At very large context lengths, spectral learning yields smaller Hellinger distances to the true output distribution, whereas the LLM achieves slightly higher accuracy. This occurs because LLMs tend to over‑concentrate probability mass on the top choice—boosting accuracy but increasing divergence for the rest of the distribution.
>
> > **Q4**: What if you gave the context then natural language prompted the model (+CoT) to predict the next state?
>
> We appreciate the thinking engagements of the reviewer! Although we didn’t include any experiments using natural language and reasoning models in this paper, we did experiment with this setup directly on neuroscience real-world data. What we found was that the LLMs struggle to do reasoning in the meaningful way that we hope. More importantly, when natural language prompts (e.g., the mice's experimental contexts) are mixed into the dynamical sequences, the prediction accuracy actually has decreased significantly. It is an interesting future work to understand what happens to the HMM sequential prediction task when natural language prompts are introduced.
>
> ---
>
> [1] Edelman, Ezra, et al. "The evolution of statistical induction heads: In-context learning markov chains." *Advances in neural information processing systems* 37 (2024): 64273-64311.

---

> > ### Comment · Reviewer_oWyV · 2025-08-05
> >
> > Thank you for your responses.
> >
> > This is a solid paper, an I will maintain my already positive score!

---

### Official Review · Reviewer_1stE · 2025-07-03

**Clarity:** 3
**Significance:** 3
**Originality:** 3
**Rating:** 4
**Confidence:** 4

**Summary:**

This work evaluates the ability of pre-trained LLMs to model data generated by Hidden Markov Models (HMMs). The work evaluates LLMs' performance on diverse synthetic HMMs, varying parameters including mixing rate, entropy level, and number of hidden states and observations. They find by leveraging in-context learning (ICL) abilities, LLMs are able to converge on theoretically optimal prediction performance in certain settings (i.e., low entropy and high mixing rate), and suffer in settings where optimal prediction is poor as well. The experiments also show that pre-trained LLMs are able to outperform state-of-the-art across various HMM settings. The work provides theoretical conjectures to help understand the performance of these pre-trained LLMs and provides practical suggestions for how to use pre-trained LLMs with ICL as diagnostic tools for complex data. Finally, they show ICL achieves competitive performance for understanding model dynamics in real-world animal decision-making datasets in simple settings, but struggles in settings similar to when it struggled in synthetic experiments (i.e., complex settings).

**Questions:**

1. Are LLMs able to discover the underlying HMM mechanisms well (i.e., stationary distribution, transition/emission matrices, etc.), or is their strong performance constrained to prediction?
2. How can we guarantee findings on using LLMs for HMM prediction can translate to inputs/fields with respect to the guidelines on using LLMs as diagnostic tools for data structure and learnability?
3. How do we expect transformer models to perform a a baseline for this task (using the sample training set-up as an LSTM)? Are the LLMs strong because of their training process, or is it simply the transformer architecture which is good for this prediction task?
4. What is the relationship between ICLs with LLMs and spectral learning that underlies the theory in Section 3.3?

**Ethical Concerns:**

["NO or VERY MINOR ethics concerns only"]

**Final Justification:**

New experiments and justification for the impact of the paper through the author discussion. Score is contingent on authors clarifying the terminology of "learning HMMs" in the updated version of the work.

**Limitations:**

yes

**Quality:**

3

**Strengths And Weaknesses:**

**Strengths**

1. The work provides a nice background of HMMs and sets up a clear story that is followed in the rest of the work.
2. Overall, the empirical results for this paper are strong and clearly answer the research questions laid out by the work: Can LLMs predict next states in HMMs, what properties of HMMs affect the LLMs ability to do so, and how does this translate to real world data.
3. The work accurately isolates entropy and mixing rates as interesting axes to vary in the HMMs, and do a good job of characterizing when LLMs fail in these different settings.
4.  The work does a good job of comparing to a theoretically optimal model to show both 1) LLMs are able to approach this model in some settings, and 2) the places where LLMs suffer are also where optimality is not guaranteed.


**Weaknesses**

1. It is difficult to read the different graphs and result plots due to their small size and small legend text. Please consider larger images or tables for displaying results, especially those in Figure 3 and Figure 5.
2. The relationship between using ICL with pre-trained LLMs and spectral learning remains unclear. The work provides a theoretical characterization of spectral learning for HMMs, but does not describe why we can view these results as a potential explanation for the ICL results due to a lack of connection between spectral learning and ICL using LLMs. Accordingly, Section 3.3 seems quite disconnected, and the theoretical results do not seem meaningful to the rest of the paper.
3. Related to the point above, there is limited understanding of why LLMs are good at this task. Is it the training process? The transformer architecture? Size of the model? What inherent to LLMs is resulting in strong performance. More experiments ablating different parts of LLMs could be useful for answering this question.
4. The findings from the reward-learning rats dataset are underwhelming. The work finds that LLMs using ICL are unable to perform well on this task compared to a baseline, and hypothesize that this is due to the complexity of the underlying dynamics. But in these complex settings, even optimal algorithms should suffer (as seen in the synthetic results). However, the baseline model performs noticeably better than the LLM, so I am not convinced it's simply too complex of a setting for the LLMs.
5. The work generalizes beyond the findings provided in the paper often. For example, Section 4 provides guidelines for how scientists can use their findings to use LLMs as data-discovery tools in new use-cases outside of synthetic HMMs, without discussing whether their results may not hold in new settings or for new data-discovery problems. I would caution the authors from making such grand statements given their findings.

**Comments**

1. The caption of Figure 4 seems incorrect, as the slower mixing rates are with the higher $\lambda_2$ values.
2. I would caution the use of "zero-shot" for these models. Using in-context learning is inherently few-shot as examples of predictions/sequences are given to the model.

---

> ### Author Rebuttal · Authors · 2025-07-31
>
> Thank you for your thorough review and thoughtful suggestions. We address your questions and concerns below.
>
> > **Q1**: Are LLMs... constrained to prediction?
>
> Our findings show that LLMs ICL performance extends beyond simple “argmax” prediction. Using distributional metrics such as Hellinger distance (Main Paper Fig. 5, right; App. D Fig. 3), we observe that LLM‑predicted distributions converge toward the ground‑truth next‑emission distributions of the underlying HMMs.
>
> Motivated by the needs of scientists analyzing real‑world dynamical systems, our goal is to assess pre‑trained LLMs as statistical tools for structured sequential data rather than to explain their internal mechanisms. Future work will investigate how LLMs internally approximate latent probabilistic structures, which may illuminate the mechanisms behind in‑context learning.
>
> > **Q2**: How can we guarantee... learnability?
>
> We interpret this question as aligned with the reviewer’s concern in **W5** regarding whether the guidelines in Section 4 generalize beyond the specific findings in this paper.
>
> Our decision to use synthetic experiments on HMMs is directly motivated by the scientific use case we stated above. We deliberately chose HMMs for two reasons:
> 1. HMMs are an expressive class of generative models that are broadly used in scientific modeling—many real-world discrete-time dynamical systems [1-3] can be represented as or reduced to HMMs.
> 2. HMMs capture how real-world scientific data is often structured: scientists often start with hypotheses about latent dynamics and choose observables accordingly [4,5]. This results in the fact that we often only can obtain partially observable data—a setting for which HMMs are a natural formalism.
>
> We also want to emphasize that the guidelines in Section 4 are intentionally cautious and observational, not prescriptive. For example, we state: *“If you observe that an LLM’s ICL prediction accuracy on your data sequence steadily improves and saturates with increased context length (like in Figure 3), this strongly indicates a learnable, non-random underlying structure.”* We do not claim this is a necessary and sufficient condition for learnability. Rather, the goal is to provide a useful heuristic for scientists who may not be experts in LLMs but are trying to assess whether their sequential data contains structure that can be exploited.
>
> Lastly, regarding the possibility of guaranteeing the utility of LLMs for real data: in scientific domains like neuroscience, practitioners commonly use models such as LSTMs [6,7] to fit behavioral or neural sequences—despite limited theoretical guarantees. LLMs ICL plays a similar practical role, offering the additional benefit of being usable out-of-the-box without task-specific training.
>
> > **Q3**: How do we expect transformer... prediction task?
>
> This is an excellent question that will be helpful for us to understand the internal mechanisms of LLMs ICL’s strong performance on HMMs. We have included an additional experiment here as a first step in response to this suggestion, and will add the full set of results to the final version of the paper. Specifically, we train a 2-layer transformer model using the same training setup as in our LSTM experiments (App. D) with the same hidden/feed forward dimension and the same embedding dimension. For each evaluation, we average performance over 16 training samples, as we do with LSTMs. We provide a comparative table below under the same setting as Figure 5 (8 states, 8 observations):
>
> |seq_len|4|8|16|32|64|128|256|512|1024|2048|
> |---|---|---|---|---|---|---|---|---|---|---|
> |LLM_acc|0.3962|0.4436|0.485|0.5095|0.5569|0.552|0.542|0.5457|0.5605|**0.564**|
> |LSTM_acc|0.1875|0.3125|0.3125|0.0625|0.375|0.5|0.375|0.375|0.3125|**0.5**|
> |TF_acc *(new)*|0.375|0.4375|0.5625|0.5|0.5|0.5625|0.4375|0.6875|0.375|**0.563**|
> |LLM_hellinger|0.512|0.4673|0.4436|0.4316|0.4162|0.4165|0.4157|0.4073|0.4026|0.4026|
> |LSTM_hellinger|0.5598|0.526|0.5146|0.5856|0.4804|0.5179|0.4846|0.4582|0.4811|0.4294|
> |TF_hellinger *(new)*|0.5391|0.5099|0.4786|0.5237|0.4721|0.4425|0.4503|0.4608|0.4435|0.4251|
>
> As shown, the trained small transformer model performs more similar to the pre-trained LLM (Qwen2.5 7B) than the LSTM baseline does. While the trained models exhibit higher variance due to limited sample sizes, these findings suggest that the transformer architecture itself plays a meaningful role in performance. In parallel, large-scale pretraining likely enables LLMs to leverage in-context learning to approximate the benefits of task-specific training.
>
> We appreciate the review’s suggestion and will incorporate the trained transformer baseline into the synthetic experiments section of the paper.
>
> Relatedly, in **W3**, the reviewer requested an ablation study on LLM size. We have conducted ablations on both model size and tokenization (App. E) from the perspective of tool usability. However, we did not emphasize these results in the main paper, as they did not reveal useful scaling trends.
>
> > **Q4**: What is the relationship... theory in Section 3.3?
>
> We interpret this question associated with the reviewer’s concern in **W2**—that the paper does not clearly explain why the theoretical characterization of spectral learning for HMMs may provide insight into the ICL results.
>
> Our primary motivation for introducing spectral learning is the hypothesis that LLM ICL behaves as an input–output predictor, effectively performing a form of smart filtering over observation sequences. This contrasts with explicit parameter‑estimation methods like the non‑convex Baum–Welch (BW) algorithm, which may fail to converge to the global optimum. This perspective is supported by the distinct behavioral patterns we observed between LLMs and BW across experiments (Sec. 3.2, Fig. 5; App. D).
>
> The second motivation is that we desire a notion of sample complexity relating to accurate prediction and complexity of the underlying dynamics. Spectral methods provide sample‑complexity guarantees for HMM prediction tasks in terms of the structural properties of the underlying process (e.g., mixing rate and emission rank). This gives a principled way to interpret the LLM ICL results (Sec. 3.1, Fig. 4), and motivates our derivation of a sample complexity bound for spectral learning on HMMs.
>
> To further support this connection, we added an additional experiment by implementing Algorithm 6 (App. F), a variant of spectral learning with theoretical performance guarantees in Theorem 1, and evaluating it on the same synthetic HMM setting used in Figure 5 (8 states, 8 observations). The results are discussed in our response to reviewer oWyV (W1 & W2) and are consistent with our hypothesis: LLMs behave more like spectral learners than EM‑based methods like BW.
>
> We acknowledge that this conceptual link between ICL and spectral learning could be emphasized more clearly in the main paper and plan to highlight it in a revision.
>
> > **W4**: The findings from the reward-learning rats...
>
> This is a very thoughtful assessment.
>
> The CogFunSearch baseline (App. H.2, “Rat FullModel” in [8]) represents a cognitive model with latent states reflecting mice’s internal strategies and emission probabilities corresponding to their decisions. This naturally maps to an HMM framework.
>
> Importantly, CogFunSearch achieves its strong performance through extremely computationally expensive search over model structures, fitting each mouse individually—taking weeks to complete—so its results are close to an “optimal algorithm” benchmark.
>
> We want to further address the reviewer’s concern about the numerical performance gap:
> 1. Neuroscientists typically evaluate models by normalized likelihood over full sequences, aggregated per animal. By this metric (used in [8]), LLM ICL achieves competitive performance for 3 out of the 20 animals (CogFunSearch 0.672±0.012). In our paper, we chose to visualize prediction accuracy versus sequence length to highlight that LLMs ICL curve is flat, which mirrors our synthetic experiments when the normalized entropy of transitions and emissions is high (e.g., App. D, Fig. 2: 8 states, 4 observations, A entropy ≈ 2.5).
>
> |Animal|M14|M23|M25|M29|M27|M46|M15|M17|M43|M37|M42|M22|M18|M40|M16|M31|M41|M47|M44|M39|
> |---|---|---|---|---|---|---|---|---|---|---|---|---|---|---|---|---|---|---|---|---|
> |LLM_nl|0.6773|0.6721|0.6671|0.6411|0.6378|0.6179|0.617|0.613|0.611|0.607|0.6041|0.6036|0.5913|0.5893|0.5882|0.5783|0.5726|0.5651|0.5542|0.5494|
>
> 2. The dataset has only two possible observations, so even modest numerical gaps can appear visually large. The Figure 7 gap corresponds to a (0.864 - 0.758)/0.5 ≈ 0.21 over chance, which is comparable to our synthetic results comparable to our synthetic high‑entropy HMM results (e.g., above setting 0.05/0.25=0.2).
>
> > **W1** and Comments
>
> Thanks for the suggestions on style improvements and catching the typo. We will surely update those, incorporating the additional experiments.
>
> ---
>
> [1] Mor B et al. (2021). *A systematic review of hidden Markov models and their applications*. Arch Comput Methods Eng 28:1429–1448.
>
> [2] McClintock BT et al. (2020). *Uncovering ecological state dynamics with hidden Markov models*. Ecol Lett 23:1878–1903.
>
> [3] Noé F et al. (2013). *Projected and hidden Markov models for calculating kinetics and metastable states of complex molecules*. J Chem Phys 139:18.
>
> [4] Gershman SJ & Niv Y (2010). *Learning latent structure: carving nature at its joints*. Curr Opin Neurobiol 20:251–256.
>
> [5] Gershman SJ et al. (2010). *Context, learning, and extinction*. Psychol Rev 117:197.
>
> [6] Jordan MI. (1986). *Attractor dynamics and parallelism in a connectionist sequential machine*. Proc CogSci 8.
>
> [7] Wills TJ et al. (2005). *Attractor dynamics in the hippocampal representation of the local environment*. Science 308:873–876.
>
> [8] Castro PS et al. (2025). *Discovering symbolic cognitive models from human and animal behavior*. bioRxiv.

---

> > ### Comment · Reviewer_1stE · 2025-08-04
> >
> > I appreciate the author's response to my questions and concerns. The response helped clarify a few points, but some concerns still remain.
> >
> > First, I still worry that to satisfy the claim that LLMs are learning hidden Markov models requires the ability for an LLM to understand/estimate the internal mechanisms/parameters of the HMM, rather than only doing next token/probability-distribution prediction. I do appreciate the Hellinger distance metric, however, and note that these results show more than only next-token prediction, which is a good step.
> >
> > Moreover, I appreciate the transformer results from the authors. However, this actually leaves me with more questions/concerns. Specifically, a very small transformer is able to achieve similar performance trends to much larger LLMs, which makes me believe that LLMs themselves are not the solution to this task, simply the transformer architecture is.  Larger transformers that are not the size of LLMs might even outperform these pre-trained LLMs, and though they are costly to train, their inference requirements would be much smaller. If the finding is simply that transformers can do well on HMM tasks, then this limits the novelty of the findings/results as it is already well-studied that LLMs are very good at pattern matching.
> >
> > Finally, the bridge between spectral learning and LLMs is still a bit difficult to understand. How are these theoretical results unique to LLMs compared to other "input-output" predictors like LSTMs, Transformers, etc.? Are these theoretical results attempting to explain only the LLMs performance, or performance trends for ML prediction models as a whole? If the latter, I still don't immediately see why this connection is necessary.
> >
> > Once again, I appreciate the authors hard work on the paper and response and am open to more discussion. However, due to these concerns, I currently leave my score unchanged.

---

> > > ### Author Response · Authors · 2025-08-06
> > > **Response to Reviewer 1stE (Part 3)**
> > >
> > > **3. On bridging spectral learning and LLMs**
> > >
> > > To achieve the observed performance—steadily approaching the theoretical optimal predictive distribution—a pre‑trained LLM must internally approximate $P(o_{t+1}|o_{1:t})$, whose optimal form is given in Eq. 1 (main paper, line 183). Our analysis in Section 3.3 formalizes why this form of in-context prediction is possible, drawing a connection to the spectral learning framework.
> > >
> > > We can view spectral learning as a theory-backed algorithm that is solving the above objective, and has empirical behavior similar to ICL. In contrast, the performance curves of trained small models, such as LSTMs or even small transformers, are often non-monotonic with respect to context length.
> > >
> > > The sample complexity (lower) bound derived from spectral learning is practically useful: it serves as a reference for practitioners, offering guidance on how long a sequence needs to be, given the complexity of the underlying dynamics, to ensure reliable learning. In real-world settings—especially in scientific domains—it is often unclear whether poor model performance arises from insufficient data or from a lack of structure in the sequence itself. Theoretical tools like these help resolve such ambiguities.
> > >
> > > We appreciate the opportunity to clarify the theoretical connection between spectral learning and LLM ICL. In response, we are considering explicitly including spectral learning as a baseline in Table 1, evaluated alongside Viterbi, Baum–Welch and trained models. We will also clarify the role of Eq. 1 and the theoretical results with more detailed and explicit explanation to bridge the connection.
> > >
> > > ---
> > >
> > > [1] Max Simchowitz, Karan Singh, and Elad Hazan. *Improper learning for non-stochastic control*. In Conference on Learning Theory, pages 3320–3436. PMLR, 2020.
> > >
> > > [2] Edelman E. et al. *The evolution of statistical induction heads: In‑context learning Markov chains*. NeurIPS 37, 64273‑64311 (2024).
> > >
> > > [3] Rajaraman N. et al. *Transformers on Markov data: Constant depth suffices*. NeurIPS 37, 137521‑137556 (2024).
> > >
> > > [4] Rajaraman N., Jiao J., & Ramchandran K. *An analysis of tokenization: Transformers under Markov data*. NeurIPS 37, 62503‑62556 (2024).
> > >
> > > [5] Mirchandani, Suvir, et al. *Large language models as general pattern machines.* arXiv preprint arXiv:2307.04721 (2023).
> > >
> > > [6] Li Y. et al. *Transformers as algorithms: Generalization and stability in in‑context learning*. ICML (2023).
> > >
> > > [7] Garg S. et al. *What can transformers learn in‑context? A case study of simple function classes.* NeurIPS 35, 30583‑30598 (2022).

---

> > > > ### Comment · Reviewer_1stE · 2025-08-06
> > > >
> > > > I appreciate the author's response to each point made.
> > > >
> > > > In response to each of the three parts:
> > > >
> > > > 1. My concern is that, with respect to past work, the novelty of the work is rather limited given the goal is learning to predict vs. actually learning underlying dynamics of HMMs. This is not to discount the existing work in the field, but rather, I believe the findings of this work are limited in the context of existing work mentioned in those fields. I do appreciate the new application to HMMs, though I worry prediction-only provides a limited understanding of the capabilities of these models above what already exists. Specifically, the authors mention the field of "Empirical demonstrations of pattern matching on complex sequences without theoretical justification", which I believe is roughly where the contribution lies as I remain unconvinced about the theoretical justification using spectral learning (see point 3). However, the field is more than just the single reference above, as this is a phenomena explored in many applications that people are viewing from many theoretical lenses (i.e., see [1]).
> > > >
> > > > 2. I agree with the authors that, as the model reaches theoretical optimal performance, models may not be able to outperform the LLM. However, the instability of the transformer might be due to the limited sizes (i.e., 2-layer), making this not a fair comparison. That being said, I do agree that there is a use in seeing how LLMs without any training perform over Transformers, which is a valid point.
> > > >
> > > > 3. I appreciate the authors description of spectral learning and the importance of these findings. However, I still remain unconvinced that this is really a theoretical justification for the performance of LLMs and ICL. I still worry that this relationship between the goals of spectral learning and LLMs is too broad to really say that spectral learning theory can help describe the performance trends of LLMs. Though the empirical trends see somewhat similar, they still differ pretty substantially and are only similar at somewhat high context lengths.
> > > >
> > > > Because of these points, I still remain with my score.
> > > >
> > > > [1] Large Language Models Are Latent Variable Models: Explaining and Finding Good Demonstrations for In-Context Learning.

---

> ### Author Response · Authors · 2025-08-06
> **Response to Reviewer 1stE (Part 1)**
>
> We sincerely thank the reviewer for the thoughtful follow‑up and for acknowledging that our initial rebuttal clarified several points, including how LLM ICL performance on real‑world HMM data informs our understanding of the underlying dynamics. We also appreciate the opportunity to address the remaining concerns.
>
> **1. On LLMs “learning” HMMs vs. next‑token prediction**
>
> We agree that for an LLM to fully “learn” an HMM in the classical statistical sense, it would need to internally recover the hidden structure—such as the transition and emission matrices. Explicit parameter recovery, as in Baum–Welch, additionally requires assumptions on the model structure (e.g., that the hidden‑state transition matrix is $M\times M$; Table 1 in the main paper). Our claim is **not** that LLMs perform such recovery. Rather, their in‑context learning behavior is consistent with **implicit statistical learning of latent dynamics**, specifically in the sense of **improper learning**.
>
> As we note in Section 3.3 (line 188-191):
>
> > “…one can predict the next observation directly using these parameters along with hidden‑state belief updates, without explicitly learning the matrices $\textbf A$ and $\textbf B$. This is therefore an example of *improper learning*, which has been extensively studied in related areas like linear dynamical systems [1].”
>
> To achieve the observed performance—approaching the theoretical optimal predictive distribution—a pre‑trained LLM must internally approximate $P(o_{t+1}|o_{1:t})$, whose optimal form is given in Eq. 1 (main paper, line 183-186). From this perspective, pre‑trained LLMs act as an effective input‑output predictors of structured sequences, much like spectral learning algorithms, and unlike BW which solves a non-convex optimization problem and does not always converge to the theoretical optimum.
>
> To avoid confusion, we are open to renaming our contribution as “learn to predict” rather than “learn” in the title, to emphasize that LLM ICL focuses on effective in‑context prediction rather than classical parameter recovery. We also invite the reviewer to clarify whether their concern is primarily terminological—i.e., about our use of “learn”—or whether they believe contributions based on implicit predictive learning are insufficient without explicit parameter recovery. We believe the former is a naming issue, whereas the latter would discount a large body of influential work on predictive modeling (e.g., spectral and meta‑learning approaches) despite their strong empirical utility.

---

> ### Author Response · Authors · 2025-08-06
> **Response to Reviewer 1stE (Part 2)**
>
> **2. On the role of transformer architecture vs. pre‑trained LLMs**
>
> We appreciate the reviewer’s observation that the transformer architecture is a key factor in the performance trends we report. As shown in the table from our previous response, a 2‑layer transformer trained from scratch on synthetic HMM sequences achieves comparable performance to a 7B pre‑trained LLM in ICL, whereas LSTMs perform substantially worse. This validates that **transformer architecture provides an inductive bias** for learning latent dynamics.
>
> However, we want to continue discuss several points where our perspective differs:
>
> > LLMs themselves are not the solution to this task… Larger transformers that are not the size of LLMs might even outperform these pre-trained LLMs…
>
> First, in many of our HMM settings, pre‑trained LLMs already converge to the theoretical optimal predictive distribution, leaving little headroom for other models to “outperform.”
>
> Our goal in this work is **not** to find the single highest‑performing HMM predictor, but rather to examine whether in-context learning (without any gradient updates) can do well on this hard statistical task. And based on our current findings in the experiments, the Hellinger distance of ICL shows very stable distributional convergence, which is impressive, whereas the trained small transformers show instability.
>
> > If the finding is simply that transformers can do well on HMM tasks, then this limits the novelty of the findings/results as it is already well-studied that LLMs are very good at pattern matching.
>
> Pre‑trained LLMs achieve this near-optimal performance on HMMs without any gradient updates. In contrast, small transformers require direct supervised training on the task data, for which having a good performance is more expected.
>
> The fact that ICL alone can solve HMM prediction is non‑trivial and not directly implied by prior ICL work. Prior work on ICL spans a spectrum:
> 1. Theoretically grounded explanations for simple tasks such as fully observed Markov models [2–4], and
> 2. Empirical demonstrations of pattern matching on complex sequences without theoretical justification [5].
>
> Our work benchmarks LLMs on HMMs with clear theoretical baselines, including Viterbi, Baum–Welch, and spectral learning, and shows that LLMs can match or surpass these classical methods. This goes beyond rote pattern‑matching and aligns ICL behavior with classical statistical learning in a principled way. Benchmarking LLMs on tasks with clear theoretical baselines—even relatively simple ones like linear regression [6,7]—is considered valuable for understanding ICL.
>
> Moreover, we demonstrate the practical usage of this capability by applying LLM ICL to real animal behavioral data, which to our knowledge is the **first demonstration of LLM ICL performing HMM‑style inference in a real‑world setting**. For these reasons, we believe our study makes a novel and meaningful contribution.

---

> ### Author Response · Authors · 2025-08-08
> **Our Contributions, and Response to Reviewer 1stE**
>
> We thank the reviewer for their time in delving deeper with us and for their openness to discussion.
>
> We respectfully disagree with the characterization of our contribution as “limited in the context of existing work mentioned in those fields.” Our core contribution is the quantitative demonstration that LLM ICL achieves near-optimal prediction for HMMs—a complex yet expressive class of generative models widely used in scientific modeling. This is significant for two reasons:
> - **For Science**: In many scientific applications, obtaining an optimal predictive model without complex, non-convex parameter fitting or additional training is highly valuable. Our synthetic experiments reveal ICL scaling trends linked to key HMM properties like mixing rate and emission entropy (Section 3). We complement this with practical guidelines for scientists who may not be LLM experts, and a real-data demonstration of the tool’s effectiveness (Section 4). This combination of empirical characterization and real-world validation makes this “off-the-shelf statistical tool” truly usable for scientists. We believe that works demonstrating how to apply ICL to a specific field or use case [20] offer significant value.
> - **For ICL**: Prior work does not imply that pre-trained LLMs can steadily converge to optimal HMM prediction. Existing works either focus on how ICL emerged [1-3,14,19], demonstrate ICL on simpler or non-sequential tasks (e.g., fully observed Markov or i.i.d. samples) [4-12,15], or examine correlations between ICL performance and pre-training data [16–19]. A detailed discussion is provided in our follow-up comment, including analysis of the “ICL as implicit Bayesian inference” line of work [14,15], which covers the paper cited by the reviewer. Achieving near-optimal predictive distributions here requires implicitly tracking latent states and performing belief-state updates over long contexts—capabilities beyond rote pattern-matching. Benchmarking against Viterbi, Baum–Welch, and spectral learning shows that LLM ICL matches or surpasses classical methods on a classically hard sequential problem without gradient updates, pushing the boundaries of current ICL understanding.
>
> ---
>
> For point 2, we thank the reviewer for noting our results with the transformer. We would like to emphasize the following (independent of model size):
> - **The small, trained transformer is a specialist**. Its strong performance is expected, as it was trained explicitly and exclusively on HMM data. Regarding the reviewer’s suggestion to explore larger transformers, we agree this could yield useful insights and will conduct necessary ablations in future work to better understand the internal mechanisms of ICL on HMMs. Our current experimental design follows prior ICL studies, which typically use 1–3-layer transformers to obtain theoretically tractable results [1–12].
> - **The pre-trained LLM is a generalist**. It has never been fine-tuned on HMMs, yet with zero gradient updates it can infer the task from context and perform on par with a specialist—approaching the theoretical optimum. This ability is both surprising and, in our view, novel. Moreover, our ablations on LLM size and tokenization (Appendix E) show that this level of performance emerges once the model exceeds a certain size threshold, and that it remains robust across different tokenization schemes.
>
> ---
>
> On point 3, we apologize for the lack of clarity here. The connection to spectral learning is not intended to provide a complete theory of how pre-trained LLMs work. Rather, it serves two specific, pragmatic purposes in our paper:
> - **Spectral learning theory offers a principled framework for interpreting our findings**. Given an observation sequence, the optimal form of $P(o_{t+1}|o_{1:t})$ (Eq. 1, main paper line 183) can be directly derived from the ground-truth HMM parameters. Spectral learning is designed to optimize this exact objective, making its sample-complexity bounds true lower limits for any learner without prior knowledge of the sequence’s latent structure. These bounds—explicitly dependent on HMM properties such as mixing rate and emission entropy—provide a plausible explanation for the shapes of the ICL performance curves we observe. They clarify how data properties influence model performance, and in practice, they can guide the choice of context length needed for reliable next-emission prediction.
> - **Spectral learning theory helps situate the ICL relative to classical methods**. Unlike EM-based algorithms (e.g., Baum-Welch), which can become trapped in local optima, both spectral learning and LLM ICL behave as input-output predictors that can bypass this non-convexity. The similar performance curves suggest they might operate in a similar paradigm, distinct from explicit parameter fitting.

---

> > ### Author Response · Authors · 2025-08-08
> > **Supportive Materials for Our Contributions (Part I)**
> >
> > **Details of ICL Related Works**
> >
> > Based on existing empirical and theoretical understanding of ICL, the capacity to steadily converge to the optimal prediction of HMM sequences is not derivable. The most relevant prior works fall into the following categories:
> > - Explain how ICL *induction heads* emergence theoretically using first-order Markov models [1-3].
> > - Explain how ICL can predict optimally on $k^{th}$-order Markov processes [4,5]. This branch of work has inspired us to add $n$-gram as one of our comparison baselines.
> > - Show how ICL can do gradient descent using linear regression task (with i.i.d. samples), with theoretical understanding on linear attention [6-9]. Show how ICL can do nonlinear tasks like classification [10-12].
> > - Compare ICL to fine-tuning [13], in which they find “in data-matched settings, in-context learning can generalize more flexibly than fine-tuning”, and propose a technique to improve finetuning.
> > - Understand ICL as implicit Bayesian inference [14,15].
> >     - [14] first connects the emergence of ICL with the next-token pretraining on data with latent document-level concepts (denoted $\Theta$). Assuming the prompt concept $\theta^* \in \Theta$, they show that “the error of the in-context predictor is optimal if a distinguishability condition holds” (e.g., $\Theta$ discrete), and that when distinguishability does not hold (e.g., $\Theta$ continuous-valued), the expected error still decreases with the length of each example. Their experiments generate synthetic datasets as mixtures of sequences from 5 HMMs (each $\theta = (\mathbf{A}, \mathbf{B}, \pi)$), then at test time sample one HMM ($\theta^* \in \Theta$) and generate $n \in \{0,\dots,64\}$ prompt sequences of length $k \in \{3,5,8,10\}$. They report that increasing $k$ significantly improves accuracy, and increasing $n$ slightly improves accuracy when $n < 20$ [14, Fig. 3]. An ablation where the test HMM is not in $\Theta$ (“unseen concept”) shows ICL “fails to extrapolate” [14, Fig. 4-right].
> >         - Their finding that accuracy increases with $k$ (under $\theta^* \in \Theta$) is explained using asymptotic statistical theory for model misspecification [14, Theorem 2]. However, under our framework—grounded in properties of HMMs—this simply reflects that ICL achieves higher accuracy with longer context lengths for an HMM sequence.
> >         - Their claim from the ablation that “ICL fails to extrapolate unseen concepts” is, in our view, likely not supported by our results. We directly evaluate an out-of-the-box pre-trained LLM on a large set of synthetic HMM parameters (234 novel $\theta^*$’s, generated through a constructive procedure unlikely to be present in any existing dataset, though we cannot assert this absolutely) and observe consistent ICL behaviors—such as near-optimality and scaling trends—as reported in Sections 2.3 and 3.1 of the main paper. One possible explanation is that the scale and diversity of pretraining data lead pre-trained LLMs to acquire higher-level “latent concepts” that confer some invariance across different synthetic HMM parameters.
> >     - [15] formulates ICL as a latent variable model, motivated from topic models (similar to [14]). However, their formulation is that ICL “learned” latent \theta is from **i.i.d. set of $k$ demonstration examples** [15, Section 2.1] (similar to the $n$ in [14]). This is a huge difference from our framework, **static vs. dynamic** latent, entirely different difficulty. So we can only view their experimental results as a preliminary potential that ICL might behave well on ICL.
> > - Under the Bayesian framework, differentiate memorization with generalization (i.e., induction heads vs. task vector) of ICL [16-18], or connect PAC sample complexity between pre-training and ICL [19]. However, our synthetic HMMs and real-world dataset are both newly generated, and are unlikely to be used in pre-training.
> > - “Report phenomena without theoretical justification”, and demonstrate how to apply ICL to a field or use case [20]. We want to suggest to the reviewer, such works are valuable, as this work is very influential in the robotics community.

---

> > ### Author Response · Authors · 2025-08-08
> > **Supportive Materials for Our Contributions (Part II)**
> >
> > [1] Bietti, Alberto, et al. *Birth of a transformer: A memory viewpoint.* Advances in Neural Information Processing Systems 36 (2023): 1560–1588.
> >
> > [2] Edelman, Ezra, et al. *The evolution of statistical induction heads: In-context learning Markov chains.* Advances in Neural Information Processing Systems 37 (2024): 64273–64311.
> >
> > [3] Makkuva, Ashok Vardhan, et al. *Attention with Markov: A curious case of single-layer transformers.* The Thirteenth International Conference on Learning Representations. 2025.
> >
> > [4] Rajaraman, Nived, et al. *Transformers on Markov data: Constant depth suffices.* Advances in Neural Information Processing Systems 37 (2024): 137521–137556.
> >
> > [5] Rajaraman, Nived, Jiantao Jiao, and Kannan Ramchandran. *An analysis of tokenization: Transformers under Markov data.* Advances in Neural Information Processing Systems 37 (2024): 62503–62556.
> >
> > [6] Akyürek, Ekin, et al. *What learning algorithm is in-context learning? Investigations with linear models.* The Eleventh International Conference on Learning Representations. 2023.
> >
> > [7] Von Oswald, Johannes, et al. *Transformers learn in-context by gradient descent.* International Conference on Machine Learning. PMLR, 2023.
> >
> > [8] Ahn, Kwangjun, et al. *Transformers learn to implement preconditioned gradient descent for in-context learning.* Advances in Neural Information Processing Systems 36 (2023): 45614–45650.
> >
> > [9] Zhang, Yedi, et al. *Training dynamics of in-context learning in linear attention.* Proceedings of the 42nd International Conference on Machine Learning. Vol. 267. PMLR, 2025.
> >
> > [10] Li, Hongkang, et al. *Transformers as multi-task feature selectors: Generalization analysis of in-context learning.* NeurIPS 2023 Workshop on Mathematics of Modern Machine Learning. 2023.
> >
> > [11] Li, Hongkang, et al. *How do nonlinear transformers learn and generalize in in-context learning?.* arXiv preprint arXiv:2402.15607 (2024).
> >
> > [12] Oko, Kazusato, et al. *Pretrained transformer efficiently learns low-dimensional target functions in-context.* Advances in Neural Information Processing Systems 37 (2024): 77316–77365.
> >
> > [13] Lampinen, Andrew K., et al. *On the generalization of language models from in-context learning and finetuning: A controlled study.* arXiv preprint arXiv:2505.00661 (2025).
> >
> > [14] Xie, Sang Michael, et al. *An explanation of in-context learning as implicit Bayesian inference.* arXiv preprint arXiv:2111.02080 (2021).
> >
> > [15] Wang, Xinyi, et al. *Large language models are latent variable models: Explaining and finding good demonstrations for in-context learning.* Advances in Neural Information Processing Systems 36 (2023): 15614–15638.
> >
> > [16] Yin, Kayo, and Jacob Steinhardt. *Which attention heads matter for in-context learning?.* Forty-second International Conference on Machine Learning. 2025.
> >
> > [17] Gupta, Ritwik, et al. *Enough coin flips can make LLMs act Bayesian.* arXiv preprint arXiv:2503.04722 (2025).
> >
> > [18] Wurgaft, Daniel, et al. *In-context learning strategies emerge rationally.* arXiv preprint arXiv:2506.17859 (2025).
> >
> > [19] Wies, Noam, Yoav Levine, and Amnon Shashua. *The learnability of in-context learning.* Advances in Neural Information Processing Systems 36 (2023): 36637–36651.
> >
> > [20] Mirchandani, Suvir, et al. *Large language models as general pattern machines.* arXiv preprint arXiv:2307.04721 (2023).

---

### Note · Authors · 2025-08-12

We thank all reviewers and the AC for their time and valuable feedback, which have strengthened our work.

This work provides the first quantitative demonstration that pre-trained LLMs, via in-context learning (ICL), can achieve near-optimal prediction on Hidden Markov Models (HMMs)—a foundational class of sequential generative models with broad scientific applications. Through synthetic experiments varying key HMM parameters (e.g., mixing rate, emission entropy), we map performance scaling trends and benchmark LLM ICL against established methods like Viterbi, Baum–Welch, and LSTM. Our results suggest that LLM ICL functions as an input–output predictor without explicit parameter recovery, paralleling spectral learning.

For science applications, this approach is practical: in real-world, often the ground truth hidden parameters are never observable, and near-optimal prediction is the best we can do. We provide scaling insights tied to HMM properties and guidelines for when ICL can be reliably applied. We further validate our findings on real-world neuroscience data, demonstrating that LLM ICL can serve as an “off-the-shelf” statistical tool for scientists without LLM expertise. This combination of systematic characterization, theoretical grounding, and real-world validation makes ICL truly usable by scientists.

In response to reviewers feedback, we have clarified:
- Novelty: LLMs’ ICL exceeds prior understanding of sequential prediction (1stE).
- Framing: the near-optimal performance on predicting HMMs corresponds to improper learning in the classical sense (1stE).
- Theory connection: empirical LLM performance aligns with spectral learning theory, which provides a principled framework for understanding sample complexity and performance trends based on HMM properties (1stE, oWyV).
- Real-world context: expanded details of the neuroscience experiments (1stE).

We also conducted additional experiments in direct response to reviewer suggestions:
1. Transformer baseline: A trained 2-layer transformer, compared to an LSTM baseline, suggests that architectural bias contributes to ICL performance. We acknowledge that fully characterizing ICL's generalization mechanisms remains important future work.
2. Spectral learning evaluation: Direct comparison supports our conjecture that LLM ICL functions like spectral learning—predicting observations without recovering explicit parameters.

We sincerely thank the reviewers and AC again for their valuable feedback.

---

### Decision · Program_Chairs · 2025-09-17

**Decision:**

Accept (poster)

**Comment:**

This paper evaluates whether LLMs can model data generated by hidden Markov models using their in-context learning abilities. The authors performed thorough empirical investigations where they varied HMM properties and showed the differences in predictive performance of LLMs. These experiments showed that HMMs can perform comparably to traditional HMM algorithms (which were also evaluated). Based on these experiments, the authors suggest that LLMs could be used as scientific discover tools for this type of sequential data and they demonstrated this approach on some animal behavior data.

The reviewers pointed a few strengths:
* The empirical validation was strong and answered the proposed question and tested with multiple models, multiple metrics, and multiple HMMs to build confidence in the results.
* The work accurately isolates entropy and mixing rate and characterizes when the method fails.
* The authors compare the approach to the theoretical optimum and show LLMs approach it in some settings and points out when optimality isn't guaranteed.
* The authors proposed guidelines for how to use the technique in the real world based on their experiments.

There were a couple weakness that the authors identified:
* The connection between ICL and spectral learning is unclear.
* What actually leads to the observed performance? Training? Transformer architecture? ...
* The authors don't distinguish between their prediction problem and learning of an HMM – i.e. estimating the underlying parameters.

The author-reviewer discussion addressed most of the weaknesses in the paper. There were a few weaknesses by a reviewer that upon closer inspection revealed a mis-understanding by the reviewer and which should not penalize the paper.

As pointed out by two reviewers, the authors should make a clear distinction between learning an HMM and the prediction problem they actually tackle in the paper in their final version.